# Ultra-conformal skin electrodes with synergistically enhanced conductivity for long-time and low-motion artifact epidermal electrophysiology

Yan Zhao[1], Song Zhang [2], Tianhao Yu [3], Yan Zhang[1], Guo Ye[1], Han Cui[4,5], Chengzhi He [6], Wenchao Jiang[3], Yu Zhai[7], Chunming Lu [7], Xiaodan Gu [2] & Nan Liu [1,3✉]

Accurate and imperceptible monitoring of electrophysiological signals is of primary importance for wearable healthcare. Stiff and bulky pregelled electrodes are now commonly used in clinical diagnosis, causing severe discomfort to users for long-time using as well as artifact signals in motion. Here, we report a ~100 nm ultra-thin dry epidermal electrode that is able to conformably adhere to skin and accurately measure electrophysiological signals. It showed low sheet resistance (~24 Ω/sq, 4142 S/cm), high transparency, and mechano-electrical stability. The enhanced optoelectronic performance was due to the synergistic effect between graphene and poly (3,4-ethylenedioxythiophene) polystyrene sulfonate (PEDOT:PSS), which induced a high degree of molecular ordering on PEDOT and charge transfer on graphene by strong π-π interaction. Together with ultra-thin nature, this dry epidermal electrode is able to accurately monitor electrophysiological signals such as facial skin and brain activity with low-motion artifact, enabling human-machine interfacing and long-time mental/physical health monitoring.

[1] Beijing Key Laboratory of Energy Conversion and Storage Materials, College of Chemistry, Beijing Normal University, Beijing, China. [2] School of Polymer Science and Engineering, The University of Southern Mississippi, Center for Optoelectronic Materials and Device, Hattiesburg, MS, USA. [3] Beijing Graphene Institute, Beijing, China. [4] Department of Acupuncture and Moxibustion, Shenzhen Traditional Chinese Medicine Hospital, Shenzhen, China. [5] CAS Key Laboratory of Human-Machine Intelligence-Synergy Systems, Shenzhen Institutes of Advanced Technology, Chinese Academy of Sciences, Shenzhen, China. [6] Beijing Advanced Innovation Center for Soft Matter Science and Engineering, Beijing University of Chemical Technology, Beijing, China. [7] State Key Laboratory of Cognitive Neuroscience and Learning & IDG/McGovern Institute for Brain Research, Beijing Normal University, Beijing, China. ✉email: nanliu@bnu.edu.cn

Epidermal electronics refer to electronic devices that are mounted on the epidermis and able to acquire information about the body precisely or to provide therapy to the body. It can open new pathways for wearable healthcare[1–3], athletic training, drug delivery systems[4], human-machine interaction[5–7], etc. Skin-contact electrophysiology, such as electromyography (EMG), electrocardiography (ECG), electrooculogram (EOG), and electroencephalogram (EEG), relying on the recording of biopotential, are important clinical measurements to assess health status. The ability to accurately and imperceptibly monitor electrophysiological signals is the most basic requirement for wearable electronics[8]. Skin-contact electrodes in clinical use are mainly pregelled Ag/AgCl[9–11]. Although they provide good signal quality, the electrolytic gel may irritate the skin and will dry in a certain period, making it impossible for long-term monitoring and diagnosis. Moreover, with the gel interfacing on skin, electrode displacement with respect to skin will cause severe motion artifact[12,13]. Specifically, when detecting facial electrophysiological signals, gel electrodes are easily detached from skin, resulting in poor electrophysiological signal resolution. This is because superficial portions of the face muscles are uneven, full of wrinkles, and inevitably large-shape changed. To achieve high signal quality with low-motion artifact for long-time monitoring, dry electrodes being ultra-conformably adhesive to skin and sufficiently mechano-electrical stable are highly desirable for biopotential monitoring.

Conducting polymer[14–18] and gold film[13,17] are two frequently studied dry electrodes for electrophysiological recordings. As a representative conducting polymer, PEDOT:PSS blended with waterborne polyurethane and D-sorbitol (~20 μm) was reported to be adhesive on skin and electrically stable upon strain, being able to measure electrophysiological signals[19]. But the detected signal quality (e.g.,: sEMG) is not good, because thick material is difficult to be ultra-conformal on skin and blending with polymer results in poor conductivity (545 S/cm) (Supplementary Table 1). PEDOT:PSS inkjet printed on a commercial decal transfer paper (~1 μm) to form a thin and intrinsic film can conquer this challenge. Such PEDOT:PSS is conformal to skin and highly conductive[20], however pure PEDOT:PSS is too brittle to maintain enough conductivity in particular when measuring electrophysiological signals on largely deformed skin. Thin Au film has also emerged as a promising alternative because they are conductive and biocompatible. For example, Au film sandwiched between parylene film (sub-300 nm) was proved to be self-adhesive to the arbitrary 3D biological surface and able to monitor surface biopotentials with less motion artifact[2]. However, Au-film dry electrodes are expensive and visible on the skin, which may bring light-induced photoelectric artifact when applied in certain scenarios. To make dry electrodes more suitable for epidermal electronics, low-cost, optical transparency, ultra-conformability to skin and mechano-electrical stability are essential characteristics to be introduced into dry electrodes for biopotential monitoring.

As the thinnest electrode material, graphene demonstrates superior optical transparency, high electrical conductivity, excellent mechanical property, and low electrochemical reactivity[21–32] (Supplementary Table 2). It is also regarded as a promising skin-electrode for electrophysiological signal detection[8]. Poly(methyl methacrylate) (PMMA)—assisted monolayer CVD graphene electronic tattoo, with a total thickness of 463 ± 30 nm, the optical transparency of ~85%, the sheet resistance of 1994.33 ± 264 Ω/sq and stretchability of ~40%, was applied on skin to detect electrophysiological signals. To fabricate dry electrodes that are easy to handle, polymer thin film (herein PMMA) was utilized. However, the PMMA polymer carrier layer would introduce contamination, cracks, or defects into the CVD-grown graphene during transfer, leading to a significant decrease in the conductivity of final electrodes[33,34]. Therefore, to improve the quality of electrophysiological recordings for graphene-based ultra-conformal dry electrodes, a modified PEDOT:PSS layer, not only capable of handling graphene as well as enhancing its superior electronic performances, was proposed in this work.

Here, we elaborated a strategy for the fabrication of transparent, highly conductive, and ultra-thin dry electrodes by using PEDOT:PSS thin film as a polymer carrier layer to handle and enhance the electrical performance of CVD-grown graphene for biopotential monitoring. To enable a thin, uniform, continuous and highly conductive PEDOT:PSS layer covered on top of graphene, surfactants and ionic compounds were added since pristine PEDOT:PSS solution did not wet graphene and exhibited limited conductivity. Optimized dry electrodes of PEDOT:PSS transferred CVD graphene film (PTG) demonstrated a total thickness of ~100 nm, a sheet resistance of ~24 Ω/sq (4142 S/cm), high optical transparency, and a sufficient mechano-electrical stability for skin electronics. The dramatic increase in electrical conductivity is not merely a consequence of a parallel connection of two conductive units, but a synergistic effect of improved molecular packing of PEDOT and charge transfer on underlying graphene. This observation has been recognized by Grazing-incidence wide-angle X-ray scattering (GIWAXS), Raman, electron spin resonance (ESR) spectroscopy, and so on. PTG exhibited outstanding capability for electrophysiological signal recording such as facial skin and brain activity in motion, enabling human-machine interfacing and long-time mental/physical health monitoring. Our synergistic strategy towards ultra-conformal, transparent, and dry conductive electrodes will pave a new way for epidermal electronics.

## Results

**Fabrication of PTG**. To achieve dry electrodes with superior electronic performance for biopotential monitoring, we synthesized graphene via CVD[35] method on copper foil and used conducting polymer PEDOT:PSS (Fig. 1a, b) as the carrier layer to transfer graphene. The fabrication process of PTG is presented in Fig. 1c. When PEDOT:PSS solution was spin-coated on graphene/Cu foil, it quickly de-wet, shrank into a droplet, and could not form a continuous transfer layer. This is because graphene is highly hydrophobic with a high contact angle (96°) due to the lack of oxygen-rich functional groups, which impedes PEDOT:PSS from wetting on the graphene (Supplementary Fig. 1). Adding surfactants[36,37] is an efficient way to enhance the wettability of PEDOT:PSS solution as well as its electrical conductivity. Here, we chose sodium dodecyl sulfate (SDS)[38], as it is a safe additive that has been widely used in detergents for laundry (Fig. 1b). In addition to SDS, an appropriate weight ratio of ionic compound[39] (herein bis(trifluoromethane) sulfonimide lithium salt as an example, BSL in short), was also introduced into PEDOT:PSS solution to further improve the electrical conductivity and mechanical stretchability of PTG as skin electrodes. As-obtained PEDOT:PSS (with SDS and BSL)/graphene/Cu foil was annealed at 120 °C for several minutes to enhance the interaction between PEDOT:PSS and graphene, and subsequently immersed in $(NH_4)_2S_2O_8$ solution to etch the underlying Cu foil, followed by rinsing in deionized water to remove possible residues (See Methods, Supplementary Figs. 2, 3). The resultant PTG thin film was then floated on water and scooped onto arbitrary substrates, such as a thin elastomer, glass, decal transfer tattoo paper and even skin, working as a dry electrode immediately.

**Optoelectronic performances of PTG**. Due to the amphiphilic nature of these additives, the addition of SDS and BSL to PEDOT:

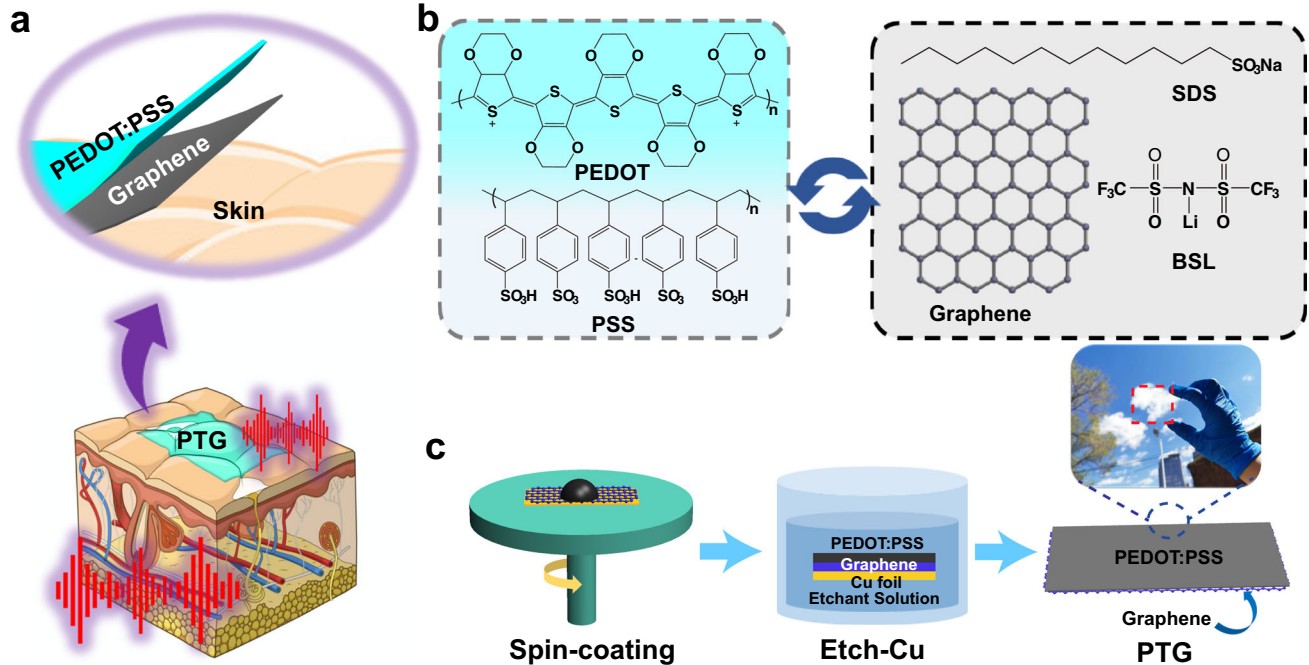

**Fig. 1 Fabrication of PTG as a skin electrode. a** Schematic illustration of PTG as a skin electrode for electrophysiology. **b** Chemical structure of PEDOT:PSS, graphene, SDS, and BSL. **c** Schematic illustration of the fabrication process of the double-layer stacked structure of PEDOT:PSS/graphene. Inset is a transparent PTG held on a plastic substrate.

PSS solution improves its wettability on the hydrophobic surface and helps to form a continuous and uniform PEDOT:PSS conductive layer on graphene. To verify the improvement in electrical conductivity of PTG with the addition of SDS and BSL, we measured 4-probe sheet resistance and transmittance on a variety of PTGs as a function of SDS and BSL. The sheet resistance of PTG thin film was clearly reduced with increased SDS concentrations in the PEDOT:PSS solution (Fig. 2a). This is because the addition of SDS may weaken the electrostatic interaction between PEDOT and PSS, leading to the decoupling of PEDOT from PSS and increased crystallinity. In the meanwhile, transmittance of PTG increased with the addition of SDS, as SDS induced a less-viscous solution and a thinner PEDOT:PSS film was formed at the same spin-coating condition. When the concentration of the surfactant exceeded a critical micelle concentration, superfluous SDS may cause flocculation, even precipitation, which would generate rough surfaces and reduce the transmittance inevitably. We indeed observed that when the concentration of SDS was more than 1.0 wt%, the PEDOT:PSS began to precipitate, resulting in big aggregates on PEDOT:PSS layer (Supplementary Fig. 4). Hence, the optimized concentration of SDS additive in PEDOT:PSS solution is chosen to be 1.0 wt%. To achieve both high stretchability and conductivity, a small molecule ionic compound BSL was added as a second additive in PEDOT:PSS solution (1.0 wt% SDS). As shown in Fig. 2b, with the increase of BSL concentration, the sheet resistance of PTG decreased, while the transmittance of PTG film initially increased at 1.0 wt% BSL and then decreased due to the gelation of the PEDOT:PSS solution. At 5.0 wt% BSL concentration, the conductivity of PTG can reach ~4142 S/cm, which exceeds the record high value for PEDOT:PSS only film (Supplementary Table 3). However, the gelation of PEDOT:PSS solution at high BSL concentration makes obstacles for the formation of uniform PEDOT:PSS carrier layers by spin-coating, leading to a thick PTG film. We also compared our PTG with most approaches recently reported for enhancing the conductivity of PEDOT:PSS in terms of film thickness (nm), conductivity (S/cm), and sheet resistance

($\Omega$/sq) (Table 1). Considering both conductivity and sheet resistance, our PTG is superior to PEDOT:PSS as transparent and conductive electrodes, particularly as ultra-conformal electrodes.

Next, the mechano-electrical stability of PTG was evaluated (see "Methods"). We placed PTG onto poly(ethylene terephthalate) (PET) and bent it at various angles. Almost no resistance change was observed when PTG was at a bending radius from 12 to 2 mm (Fig. 2c). To evaluate its electrical durability at strain, PTG was then transferred onto an elastomeric styrene-ethylene-butadiene-styrene (SEBS) substrate, and the 2-probe resistance change was monitored under strain. It is noted that upon increasing the number of the underlying graphene layer, the resistance change becomes slower. For the SEBS-supported PEDOT:PSS-transferred tri-layer graphene, the resistance was kept as low as 1680 $\Omega$ at 40% strain (Fig. 2d), suggesting that it is capable of working as a stretchable electrode up to 40% strain. The underlying graphene of PTG results in the better electrical stability upon tensile strain. To understand this mechanism, we performed a systematic morphology study on pure PEDOT:PSS and PTGs (1G+PEDOT:PSS, 2G+PEDOT:PSS) on SEBS elastomer (50 μm thick) before and after mechanical stretching at 20% strain by AFM. In the high magnification images, the granule-to-nanofibril transition was clearly observed on PTGs (Supplementary Fig. 5). Applied 20% strain on SEBS, cracks appeared on all the films, which are mostly perpendicular to the strain direction. However, arising from the underlying graphene, there are obviously less cracks and vaguer parallel wrinkles on PTGs than those on pure PEDOT:PSS film (Fig. 2e). It indicates that a smaller tensile strain was transferred onto PTG due to the large sliding between graphene and elastomer. With the increase of graphene layer number, this factor is strengthened accompanied with the formation of more percolation pathways, leading to better electrical durability at strain. This is consistent with the report for multilayer graphene as stretchable electrodes[40,41]. Thus, to compromise the transparency, conductivity and mechano-electrical stability of PTG, our optimal condition for

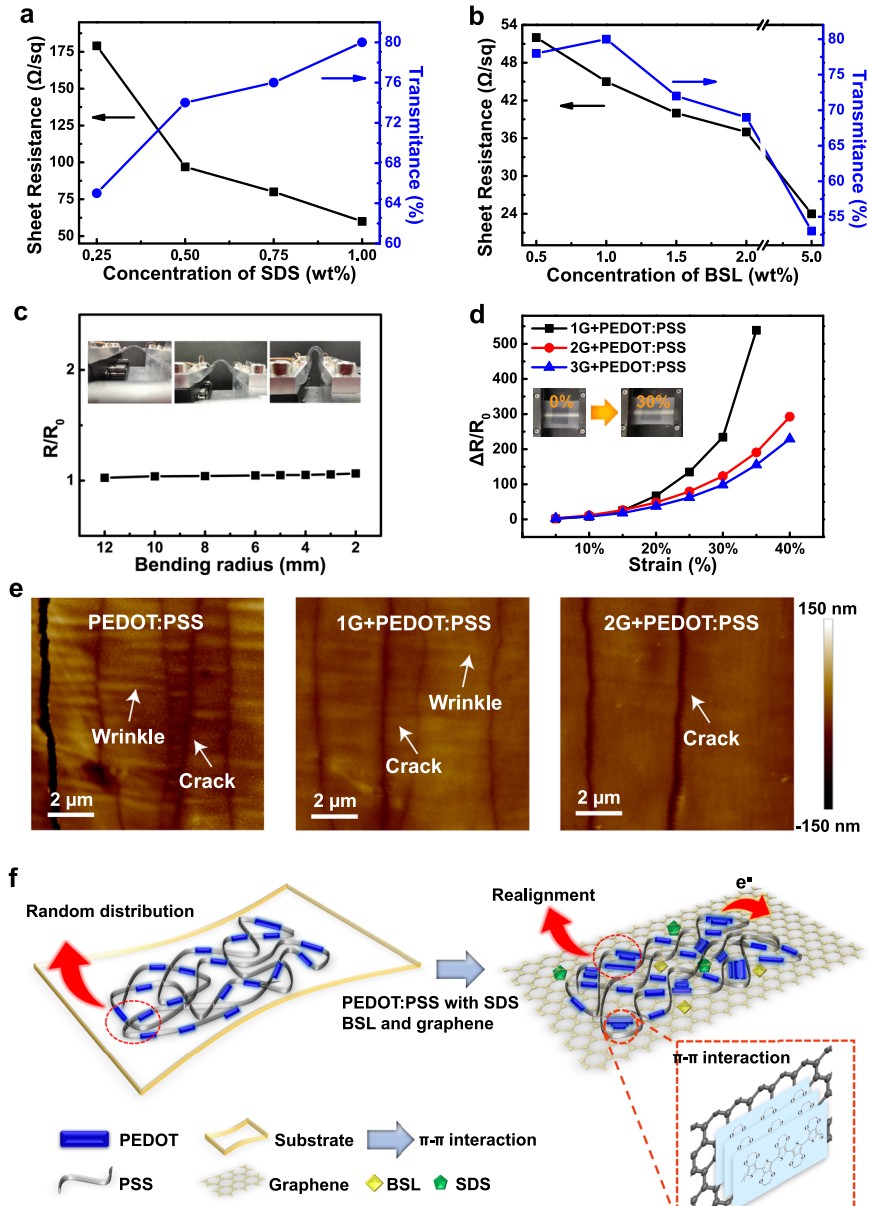

**Fig. 2 Optoelectronic performances of PTG. a**, **b** Sheet resistance and transmittance (at λ = 550 nm) of PTG films at various SDS (**a**) and BSL (**b**, 1.0 wt% SDS) concentrations when forming PEDOT:PSS carrier layer. **c** Normalized resistance changes of PTGs on PET at bending radius from 12 to 2 mm, and inset photographs showed PTGs on PET at different bending degrees. **d** Normalized resistance changes of PTGs on elastic SEBS at various strains. PTGs were made by PEDOT:PSS transferred mono-, bi-, and tri-layer graphene. **e** Morphological comparison of pure PEDOT:PSS and PTGs (mono- and bi-layer graphene) on SEBS upon 20% strain by AFM. The strain direction is perpendicular to the cracks and parallel to the wrinkles which are generated by Poisson effect. **f** Schematic illustration of the synergistic enhancement in electrical conductivity by π-π interaction and charge delocalization between graphene and PEDOT:PSS.

following epidermal electrophysiology is bi-layer graphene transferred by PEDOT:PSS at both SDS and BSL concentrations of 1.0 wt%.

It is noteworthy that at the same spin-coating condition with same concentration of additives (SDS+BSL) in PEDOT:PSS solution, PEDOT:PSS only (~185 nm) film is ~2.3 times thickness of PTG (~80 nm, 1G) film (Supplementary Table 3). Because contact angles of PEDOT:PSS solution on graphene and $SiO_2$/Si substrates are similar (74.6° and 74.5°, respectively, Supplementary Fig. 6), the obviously decreased thickness of PTG is highly likely due to the strong π–π interaction between PEDOT:PSS and graphene, leading to reordering of PEDOT on graphene. On the other hand, at the same thickness (~80 nm) of PTG and pure

PEDOT:PSS films, the conductivity of PTG (~2850 S/cm, 1G) is ~2 times high of PEDOT:PSS (1620 S/cm). These indicate that the synergistic enhancement between PEDOT:PSS and graphene is the main contributor to the high electrical conductivity of PTG. Detailed preparation condition and electrical property of the PTG and pure PEDOT:PSS films were summarized in Supplementary Table 3. We hypothesized that by interfacing PEDOT:PSS with graphene, the molecular packing of PEDOT will be rearranged, with a highly ordered thin-film morphology as induced by the electron coupling from graphene. In the meantime, graphene will be charge doped by PEDOT:PSS while maintaining a clean interface without the removal of the polymer carrier layer (Fig. 2f).

**Table 1 Sheet resistance ($R_{sh}$), Thickness and Conductivity of different PEDOT:PSS film.**

| Methods | $R_{sh}$ (Ω/sq) | Thickness (nm) | Conductivity (S/cm) | Application | Reference |
|---|---|---|---|---|---|
| Ionic liquid additive | 59 | – | 3100 | Stretchable circuit | *Sci. Adv.* **3**, e1602076 (2017) |
| H$_2$SO$_4$ soaking | 46 | – | 4000 | Solar cell | *Adv. Mater.* **27**, 2317–2323 (2015) |
| HNO$_3$ soaking | – | 27 | 4100 | Solar cell | *Adv. Electron. Mater.* **1**, 1500121 (2015) |
| Zn(TFSI)$_2$ doping | – | – | 4115 | Solar cell | *Joule* **3**, 2205–2218 (2019) |
| MSA soaking | 50 | 79 | 2540 | Solar cell | *ACS Appl. Mater. Interfaces* **7**, 15314 −15320 (2015) |
| Methanol/MSA soaking | 43 | 65 | 3560 | Solar cell | *ACS Appl. Mater. Interfaces* **7**, 16287 −16295 (2015) |
| PTG | 45 | 80 | 2850 | Electrophysiology detection | This work |
| | 24 | 100 | 4142 | | |

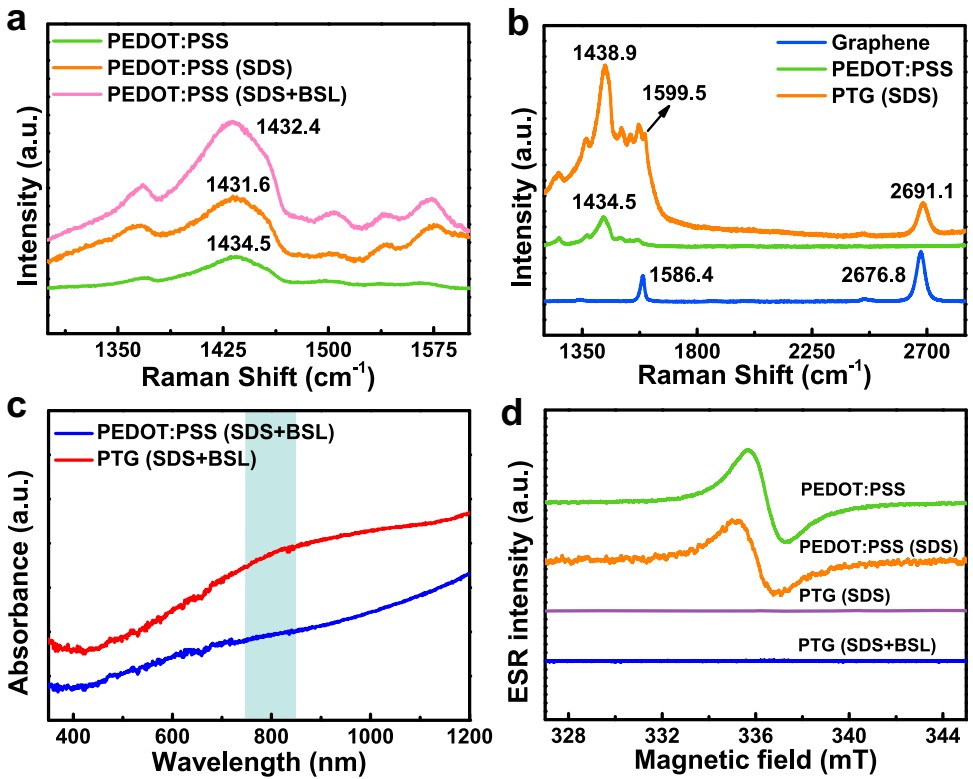

**Fig. 3 Synergistic enhancement between PEDOT:PSS and graphene. a** Raman spectra of PEDOT:PSS films with different additives. **b** Comparison of Raman spectra of graphene, pristine PEDOT:PSS and PTG films. **c** UV–vis-NIR spectra of pure PEDOT:PSS and PTG films. **d** ESR spectra of various PEDOT: PSS and PTG films.

**Synergistic enhancement between PEDOT:PSS and graphene.** To Figure out the synergistic effect between PEDOT:PSS and graphene, we performed a comprehensive study on the structure of PTG. Raman spectroscopy is a powerful tool to characterize carbon-based materials. For the PEDOT:PSS film only, there was a vibration peak at 1434.5 cm$^{-1}$ originated from C=C symmetrical stretching[39] of PEDOT. After the addition of SDS and BSL, the peak position red-shifted to 1431.6 cm$^{-1}$ and 1432.4 cm$^{-1}$, respectively. The PEDOT chain in the pristine PEDOT:PSS aqueous solution is coiled due to the Coulombic interaction. With the addition of SDS and BSL, the coiled PEDOT chain becomes linear. The red-shift confirmed the structural transition of the thiophene ring on PEDOT chains, from a coil-like benzoid structure in pristine to a linear-like quinoid structure[42,43] (Supplementary Figs. 7, 8). This structural transition led to an effective interchain coupling of PEDOT chains. After interfacing PEDOT: PSS with graphene (Fig. 3b), the C=C symmetrical vibration peak

of PTG further shifted to 1438.9 cm$^{-1}$, indicating a more thorough structural transformation of PEDOT. Moreover, the G band, a characteristic peak of CVD-grown graphene (1586.4 cm$^{-1}$) was shifted to 1599.5 cm$^{-1}$ in PTG. Such blue-shifts indicated that there is strong π–π interaction between PEDOT:PSS layer and graphene, which will engender the delocalization of π electrons, thus increasing the charge carrier mobility of PTG thin film.

The charge delocalization of π electrons was also confirmed by UV–vis-NIR spectroscopy. Compared with pristine PEDOT:PSS film, a strong absorbance occurred in PTG over a broad range, both at the ~800-nm peak and the free-carrier tail of the near-infrared region. This observation proved the charge delocalization between graphene and PEDOT:PSS, leading to the formation of polaron pairs, that is, bipolarons. The π-π interaction effect was then studied by electron spin resonance (ESR), which probed the presence of unpaired electrons (Fig. 3d). The strong ESR signals of pristine PEDOT:PSS films indicated the localized charge carrier

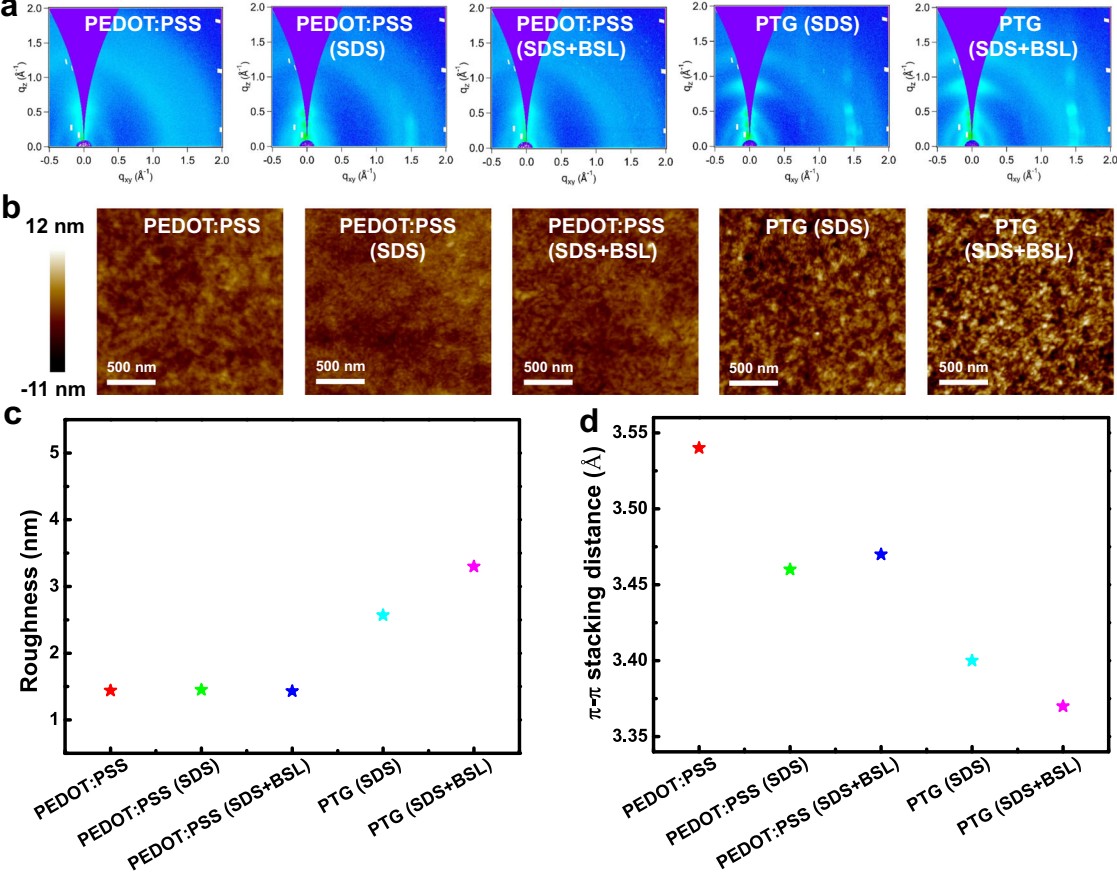

**Fig. 4 Highly ordered PEDOT in PTGs. a** 2D GIWAXS patterns of pristine PEDOT:PSS, PEDOT:PSS (SDS), PEDOT:PSS (SDS+BSL), PTG (SDS), and PTG (SDS+BSL). **b** AFM height images of various films. **c, d** Roughness (**c**) and π–π stacking distances (**d**) of above films.

in PEDOT chains. In contrast, at the presence of graphene, PTGs, either with SDS or BSL, showed significantly decreased ESR signals. This result implied that the charge carrier was converted from localized polaron states to delocalized bipolaron states upon interfacing PEDOT:PSS with graphene. Due to the strong π–π interaction between PEDOT:PSS and the underlying graphene, a more ordered semicrystalline structure of PEDOT can be achieved, thus an enhanced electrical property of PTG can be obtained.

The effect of additives (SDS, BSL) and graphene on the molecular packing of PEDOT was further investigated by Grazing-incidence wide-angle X-ray scattering (GIWAXS), an advanced technique to analyze thin-film morphology. Fig. 4a compared 2D GIWAXS patterns of PEDOT:PSS films and PTGs with and without additives, respectively. The 1D scattering profiles along $q_z$ and $q_{xy}$ directions were obtained from the 2D GIWAXS patterns (Supplementary Fig. 9). The (010) peak at around $q_z = 1.8\ Å^{-1}$ is attributed to the π–π stacking of PEDOT, and their lattice spacing can be calculated from Bragg's law and shown as follows: 3.54 Å (pristine PEDOT:PSS), 3.46 Å (PEDOT: PSS with SDS), 3.47 Å (PEDOT:PSS with SDS and BSL) 3.40 Å (PTG with SDS) and 3.37 Å (PTG with SDS and BSL). With the addition of SDS, BSL, and mainly the underlying graphene, the π–π stacking distance of PEDOT decreases from 3.54 to 3.37 Å (Fig. 4d), which ranked among the smallest intermolecular stacking distance of solution-processed PEDOT observed so far[44]. The closer packing of PEDOT chains in PTG again confirmed the strong π–π interaction between graphene and PEDOT, which could contribute to an enhanced electrical conductivity.

The highly ordered molecular packing in PTGs can also be reflected in the surface morphology. We used atomic force microscopy (AFM) to observe the morphological evolution of pure PEDOT:PSS and PTGs (Fig. 4b). With the addition of SDS and BSL, limited change in the surface roughness and morphology can be observed when compared with pristine PEDOT:PSS films. While interfacing with graphene, PTGs exhibited a distinct morphological transition from granules to nanofibrils, and a 2–3 times increase in roughness values (Fig. 4c), which is consistent with a previous study[39,44]. The more ordered PEDOT packing in PTGs can be explained by the strong π–π interaction between graphene and PEDOT:PSS. Overall, the synergistic enhancement between PEDOT:PSS and graphene will induce a higher crystallinity in PEDOT, so as to achieve a high conductivity for PTG to accurately detect electrophysiological signals.

**Low-motion artifact electrophysiological monitoring**. Interfacial impedance is a critical parameter in epidermal electrophysiological measurement. Without additional gel, dry electrodes would reduce the contact impedance by forming conformal interface with skin. According to the Young's modulus map under AFM, average Young's modulus of PTG is about 640 kPa (Fig. 5c, Supplementary Fig. 10), which matches the modulus of human stratum corneum (~150 kPa). Bending stiffness ($\bar{E}I$) refers to the amount of stiffness that the subject will deflect:

$$\bar{E}I = \frac{1}{12(1-v^2)}\bar{E}h^3$$

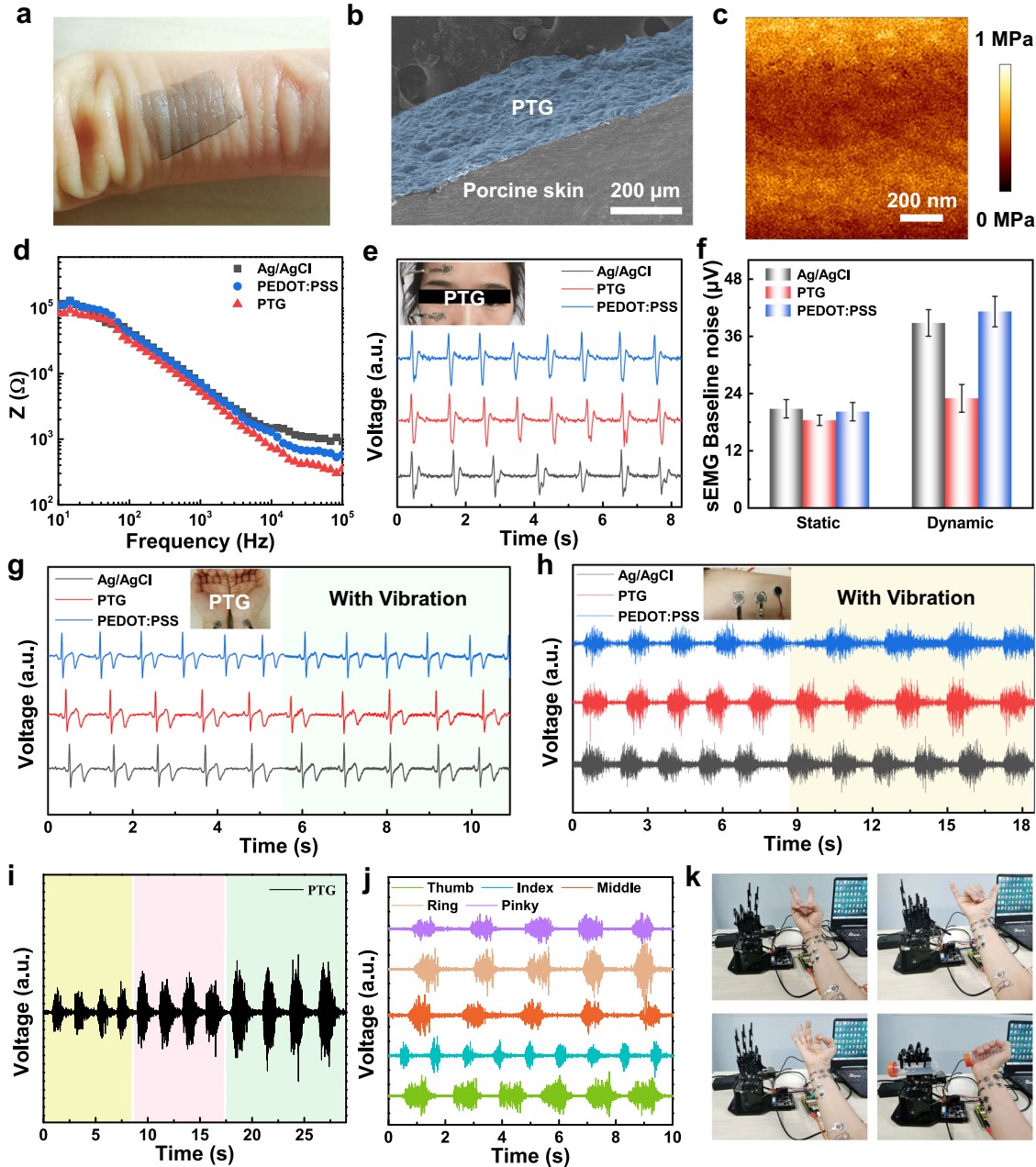

**Fig. 5 Electrophysiological signal detection of PTG dry electrodes. a** A photograph of PTG conformed on a human ring finger. **b** A SEM image of cross-sectional view of PTG (top) on porcine skin (bottom). **c** AFM Young's modulus map of PTG. **d** The impedance analysis of PTG (red), pure PEDOT:PSS (blue) and Ag/AgCl (black) electrodes. **e** EOG measured by PTG (red), pure PEDOT:PSS (blue) and Ag/AgCl (black) electrodes, showing peaks and valleys corresponding to the eyelid movement. **f** sEMG baseline noise comparison of the three types of electrodes in static and dynamic status. **g** ECG measured by PTG (red), pure PEDOT:PSS (blue) and Ag/AgCl (black) electrodes without and with vibration. Characteristic P, Q, R, S, and T waves are clearly identified. **h** sEMG measured by PTG (red), pure PEDOT:PSS (blue) and Ag/AgCl (black) electrodes without and with vibration, showing lower-motion artifact detected by PTG. Insets of (e,g., **h**) are photos of PTG electrodes for EOG, ECG, sEMG measurements. **i** sEMG measured by PTG at different griping force, showing that sEMG intensity becomes higher at larger gripping force. **j** sEMG acquired by PTG from the finger (thumb, index, middle, ring, and little fingers) motion. **k** Driving the robotic hand to show various gestures (victory, Arabic numerals of six and three, and holding a centrifuge tube) based on multi-channel sEMG signals extracted by PTG electrodes. Five PTG electrode pairs were adhered on designated locations for acquiring sEMG signals of thumb, index, middle, ring and little fingers, and remotely driving the movement of robotic hand.

where $\bar{E}$ represents Young's modulus, $h$ represents thickness, $\upsilon$ represents Poisson ratio. The ultra-thin nature and low Young's modulus of PTG lead to a similar bending stiffness with that of skin, indicating conformal contact with wrinkled skin (see Discussion in Supplementary information)[13,45]. Transferred onto a human ring finger full of wrinkles, PTG indeed exhibited intimate contact on it (Fig. 5a). The conformability is also confirmed by

SEM imaging of PTG on porcine skin (Fig. 5b, Supplementary Fig. 11). The interfacial impedance analysis between electrodes and skin was conducted on the human arm with electrodes placed on the surface of it (Fig. 5d). PTG showed comparable interfacial impedance with PEDOT:PSS dry electrodes and commercial Ag/AgCl gel electrodes, and at 100 Hz, the impedance (~32 kΩ) is lower than PEDOT:PSS (~42 kΩ) and the Ag/AgCl (~45 kΩ).

With low interfacial impedance, PTG was next applied in electrophysiological detection, with pure PEDOT:PSS and Ag/AgCl as control. EOG is to record eye potential and helps noninvasively assess the oculomotor function. Electrodes were adhered on the forehead and lower eyelids. Three types of electrodes showed comparable peaks and valleys with clear eye health information (Fig. 5e). ECG is another important electro-physiological signal for detecting heart activity. Dry electrodes were attached on the forearm for measuring ECG signals (Fig. 5g). All of the rhythmics-relevant parameters, such as P, Q, R, S, and T waves (Supplementary Fig. 12), can be clearly identified, which is crucial in clinical diagnosis. To record the muscle bioelectricity, sEMG was detected by adhering electrodes on the forearm (Fig. 5h, i). In static state, the sEMG signal amplitude obtained by three kinds of electrodes were comparable, while signal-to-noise ratio (SNR) values obtained by PTG ($23 \pm 0.7$ dB) are higher than those of Ag/AgCl electrodes ($19 \pm 0.5$ dB) and PEDOT:PSS electrodes ($18 \pm 0.5$ dB). (Supplementary Fig. 14). As electrolytic gel will dry (Supplementary Fig. 13) and induce electrode displacement, the major challenge of Ag/AgCl electrodes in electrophysiological detection is the motion artifact. Hence, the motion artifact was evaluated here by baseline noise. We used electromechanical vibrator to mimic skin vibration in arm movement. The electromechanical vibrator was placed near the working electrodes (Supplementary Fig. 15). The average baseline noise of PTG in sEMG recording was just $23 \pm 2.9$ μV, 1.7 times smaller than $38.8 \pm 2.8$ μV of Ag/AgCl and 1.8 times smaller than $41.2 \pm 3.2$ μV of PEDOT:PSS in the same condition (Fig. 5f). Thus, in dynamic status, PTG electrodes demonstrated relatively stable SNR ($21 \pm 0.3$ dB) than those of Ag/AgCl electrodes ($14 \pm 0.4$ dB) and pure PEDOT:PSS ($13 \pm 0.4$ dB) (Supplementary Fig. 14). The improved sEMG signal quality in both static and dynamic status can be attributed to the better conformability and mechano-electrical stability of PTG. Motion artifact was also evaluated in ECG (Fig. 5e, f). Transited from the static to dynamic status, the baseline noise detected by PTG is relatively stable than Ag/AgCl and PEDOT:PSS electrodes. sEMG is more sensitive to the vibration compared with ECG, because the detected bandwidth (20–2000 Hz vs. 0.05– 100 Hz) is higher, and muscle contraction comes along with large skin deformation. Although pure PEDOT:PSS was measured in above tests, it is noted that without the additional graphene, pure PEDOT:PSS film is very brittle and easily break out upon strain. In addition, the infrared image (Supplementary Fig. 16) showed that a minimal level of heat was accumulated upon mounting the PTG electrodes on a volunteer's forearm. Overall, our ultra-conformal PTG exhibited less motion artifact in electrophysiological signal detection, suggesting PTG is potential for accurate human-machine interfacing.

The sEMG signals are generated by muscular excitation-contraction, and converting the sEMG signals to control artificial limbs is urgently expected clinically. Firstly the sEMG signals were extracted by our electrodes and then converted to Pulse Width Modulation (PWM) pulses to drive the servo motor (Supplementary Figs. 17–21). Even with external interference (Supplementary Fig. 21), sEMG signal extracted by PTG can still precisely control the mechanical claw due to its low-motion artifact detection. Five electrode pairs were adhered on designated locations to respectively detect sEMG signals of thumb, index, middle, ring and little finger of the volunteer (Fig. 5j), showing high detected quality with SNR values around 20 dB (Supplementary Fig. 23). These sEMG signals were further converted to control the movement of corresponding fingers of the robotic hand (Supplementary Fig. 22). As shown in Fig. 5k and Supplementary Movie 1, various gestures of the robotic hand, such as victory, Arabic numerals of six and three, and holding a

centrifuge tube, were demonstrated. Equipped with complicated digital processing and learning algorithms, our ultra-conformal and dry PTG skin electrodes could be applicable in controlling artificial limbs with more complexity and accuracy.

**Facial sEMG acquisition and facial expression-robot interaction.** EMG is an important way to evaluate facial muscle function and analyze facial muscle expression (Fig. 6a). Compared to needle EMG (nEMG), sEMG is superior in painlessness, invasiveness, large facial muscle area characterization, and monitoring over time. Thus, sEMG is more practical in wearable healthcare to self-evaluate facial nerve disorder for both pre- and post-diagnosis. Considering that the superficial portions of the face muscles are uneven, full of wrinkles, and inevitably large-shape changed, facial sEMG electrodes being highly conductive, ultra-conformal as well as mechano-electrical stable are necessary. It is still a big challenge to detect facial sEMG stably and accurately both in laboratory and in clinic. To verify the capability of PTG as facial sEMG electrodes, we applied a pair of them on levator labii superioris alaeque nasi on one side of the face with Ag/AgCl as control on the other side (Fig. 6d). PTG showed more stable signal quality than Ag/AgCl in facial sEMG detection (Fig. 6b). This phenomenon can be attributed to the low-motion artifact electrophysiology of PTG on skin. In contrast, Ag/AgCl gel electrodes are easily detached from the facial skin resulted from the displacement of gel, and the PEDOT:PSS is too brittle and completely insulative when the facial skin is stretched in dynamic state (Supplementary Fig. 24). With the addition of graphene underlying PEDOT:PSS, the electrical stability of PTG upon stretching is greatly enhanced (Fig. 2d), facilitating the accurate sEMG signal acquisition in particular in motion of the face. Laser speckle contrast imaging technique (LSCI) is another important technique for facial nerve diseases diagnosis. It can evaluate the function of facial microcirculation, such as blood flow velocity and blood perfusion. Leveraging laser speckle contrast imaging into sEMG measurements, it will be beneficial to the deep understanding of pathological process and mechanism (Fig. 6c). Commercial Ag/AgCl gel electrodes are opaque. Utilizing the relatively transparent PTG dry electrodes, simultaneously monitoring sEMG and speckle imaging at the same location can be realized, which will provide more information to the diagnosis and treatment of facial nerve palsy (Fig. 6d, e). The blood perfusion under PTG electrodes can be measured by LSCI whereas the blood perfusion under Ag/AgCl electrodes cannot be measured by LSCI. Moreover, with the accurate monitoring of sEMG by PTG, we applied it to control the robotic hand by facial muscle expression. Three pairs of PTG were attached on zygomaticus, risorius, and corrugator to control the index, middle finger, and wrist of the robotic hand respectively (Supplementary Movie 2). When the volunteer smiles, the robotic hand shows the gesture of victory (Fig. 6f). When the volunteer extends and knits her brows, the robotic hand is able to rotate its twist correspondingly (Fig. 6g). Taken together, the high conductivity, ultra-conformability, certain mechano-electrical stability and transparency contributes PTG as an ideal electrode for the acquisition of facial sEMG without detriment to laser speckle contrast imaging, as well as being able to control artificial limbs.

**Long-time monitoring of EEG signals.** Electroencephalogram (EEG) recording is to measure the electrical activity of the brain, providing helpful information for the diagnosis or treatment of the brain disorders. Due to the very weak signal strength (μV scale), long-time monitoring of EEG is extremely challenging, in particular for the status of exercising. Normally, for EEG monitoring, the volunteer sits in a comfortable position peacefully and experiences

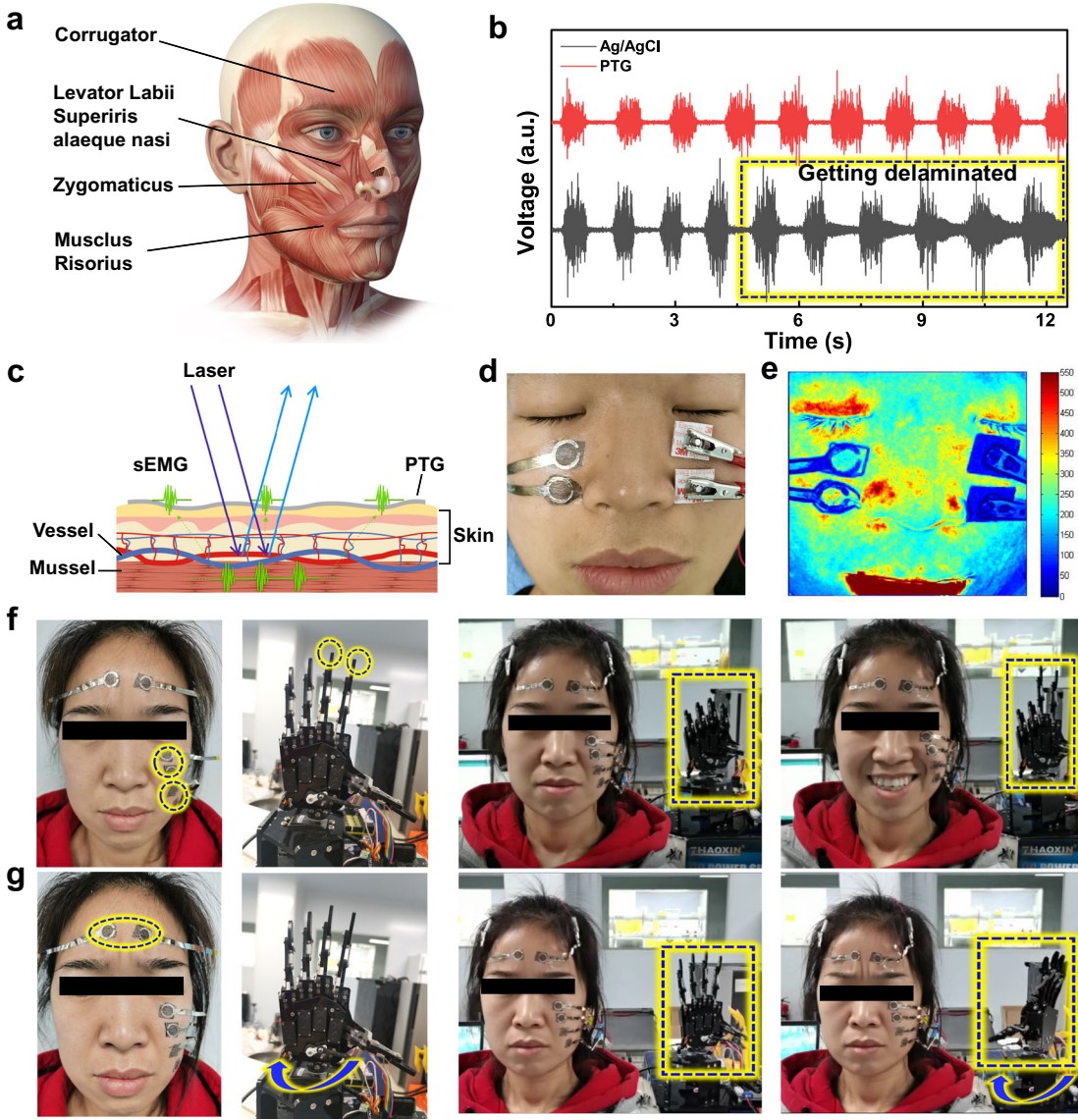

**Fig. 6 Facial sEMG acquisition and facial expression-robot interaction. a** A photo of facial expression muscles. **b** Facial sEMG measured by PTG (red) and Ag/AgCl (black) electrodes. PTG showed lower-motion artifact. **c** Schematic illustration of simultaneously monitoring sEMG and speckle imaging at the same location. **d** A photograph of PTG electrodes connected by metal for facial sEMG detection on levator labii superioris alaeque nasi on one side of the face with Ag/AgCl as control on the other side. Photo credit: Yan Zhao, State Key Laboratory of Digital Manufacturing Equipment and Technology, Beijing Normal University. **e** A laser speckle contrast imaging corresponding to (d), The blood perfusion under PTG electrodes can be measured by laser speckle contrast imaging technique (LSCI) whereas the blood perfusion under Ag/AgCl electrodes cannot be measured by LSCI. Photo credit: Yan Zhao, State Key Laboratory of Digital Manufacturing Equipment and Technology, Beijing Normal University. **f** Driving the robotic hand based on multi-channel facial sEMG signals extracted by PTG electrodes. Two PTG electrode pairs were adhered on zygomaticus and risorius to control the index and middle finger, respectively. Right photos showing that the robotic hand at the gesture of victory when the volunteer smiles. Photo credit: Yan Zhao, State Key Laboratory of Digital Manufacturing Equipment and Technology, Beijing Normal University. **g** PTG electrodes were adhered on corrugator to control the wrist of the robotic hand. Photos showing that the robotic hand rotates its twist when the volunteer extends and knits her brows Photo credit: Yan Zhao, State Key Laboratory of Digital Manufacturing Equipment and Technology, Beijing Normal University.

brain stimulus, such as doing math quiz, watching videos or listening to music. As PTG is ultra-thin and the attachment is comfortable, we, therefore, applied PTG on a volunteer for long-time monitoring of EEG without taking off the electrode pairs. During the monitoring of 12 h, the volunteer slept, exercised, and recovered to calm with two PTG electrodes mounted in the frontal region (Fp1 and Fp2) of the brain and one reference electrode behind the ear[46]. The EEG signal quality is decent, despite that the volunteer is at various statuses and even soaked in sweat after exercising. Fig. 7a–c showed three typical EEG signals corresponding to the three statuses of brain. After fast Fourier transformation (FFT) treatment (Fig. 7d–f), the brain waves

can be divided into five parts: delta wave (0–4 Hz), theta wave (4–8 Hz), alpha wave (8–13 Hz), beta wave (13–30 Hz), and gamma wave (40 Hz or higher)[46,47]. To differentiate various statuses of the brain, alpha and beta waves were extracted and compared separately (Fig. 7g–l). The intensity of alpha and beta waves at exercising is obviously stronger than that at sleeping and calm status. This is in accord with the cognition that exercise improves our concentration. The stable recording of EEG signals can be explained by the intimate contact of ultra-thin PTG on skin regardless the extension/contraction and sweat secretion of skin. This can be verified by the stable contact impedance at 10 Hz as a function of time (Supplementary

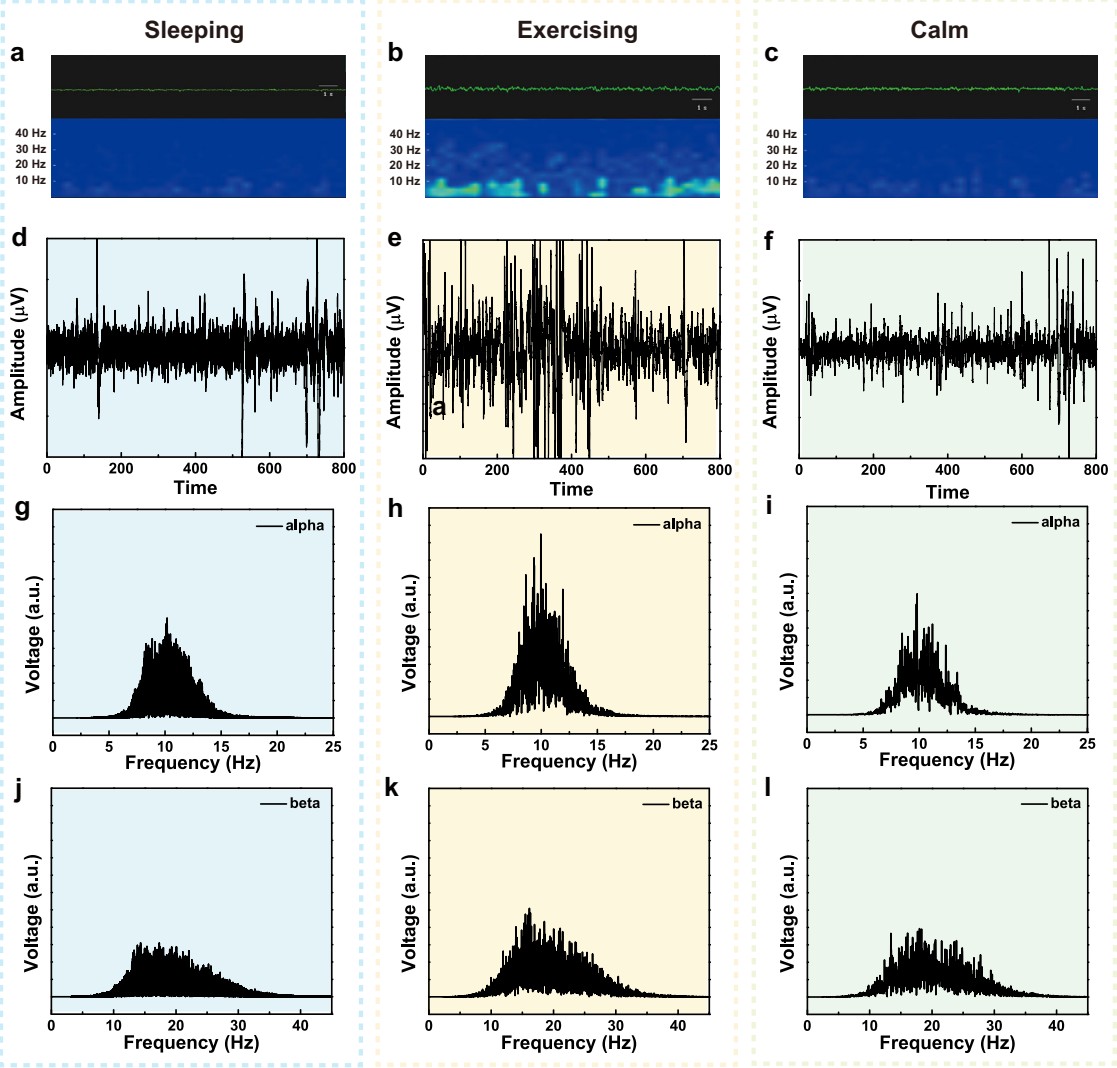

**Fig. 7 Long-time monitoring of EEG. a–c** Original EEG signals, **d–f** fast Fourier Transformed EEG signals, **g–i** the extracted alpha waves and **j–l** beta waves when the volunteer was at status of sleeping, exercising (skipping rope) and recovering to calm state after ~1 h.

Fig. 25). The capability of long-time monitoring of EEG suggests that PTG is an ideal skin-electrode candidate, facilitating the future healthcare and human-machine interfacing based on accurate EEG acquisition.

## Discussion

In conclusion, by leveraging two well-known skin electrodes together, we have demonstrated a transparent, highly conductive, and ultra-conformal dry electrode, which is PEDOT:PSS thin film-transferred CVD-grown graphene (PTG). Graphene transferred by PEDOT:PSS without the step of removing the carrier layer was confirmed to be electron doped with higher conductivity; on the other hand, the PEDOT backbones showed improved molecular packing due to the underlying graphene with strong π–π interaction. Thus, the synergistic enhancement effect between graphene and PEDOT:PSS resulted PTG in a very high conductivity (4142 S/cm). Together with the ultra-thin feature, PTG dry electrode exhibited excellent capability of long-time and low-motion artifact electrophysiological signal recording. For facial muscle evaluation, it showed superior advances in lower-motion artifact electrophysiology and less disturbance to laser speckle contrast imaging than commercial Ag/AgCl gel electrodes, and being able to stably and accurately control the robotic

hand. For the very weak EEG signals, the user can continuously wear the PTG for 12 h monitoring regardless the extension/contraction and sweat secretion of skin. Our proposed ultra-thin, transparent, and conductive dry electrodes will pave a new way for epidermal electronics.

## Methods

**Materials**. PEDOT:PSS solution (Clevios PH 1000) was purchased from Han Feng, the sodium dodecyl sulfate (99%) was obtained from Innochem, the SEBS and BSL were obtained from Aladdin Industrial Corporation. The decal transfer paper was purchased from The Magic Touch Ltd., UK.

**Fabrication of PTG dry electrode**. Cu foil was put in the quartz tube under a constant pressure of 20 sccm (standard cubic centimeter per minute) $H_2$ and 35 sccm $CH_4$. After 40 min growth at 1000 °C, both sides of Cu foil were covered with monolayer graphene. The SDS of 0.25 wt%–1.5 wt% was added to PEDOT:PSS solution and stirred vigorously for 2 h. The BSL was then added to PEDOT:PSS/SDS solution and stirred for another 2 h. Before spin-coating, the prepared PEDOT:PSS solution was filtered by 0.45 μm PES syringe filter. And the PEDOT:PSS solution was spin-coated on the Cu foil with graphene at 2000 rpm for 60 s, forming a PEDOT:PSS/graphene/Cu structure. The PEDOT:PSS coated graphene on Cu foil was annealed at 120 °C. Subsequently, Cu foil was etched by ammonium persulfate [$(NH_4)_2S_2O_8$] solution about 2 h, then the PEDOT:PSS/graphene film was washed in deionized water for 3 times, The film can be transferred to arbitrary substrates, such as silicon wafer, glass, decal transfer tattoo paper, and flexible substrates.

**PTG Characterization**. The sheet resistances were measured using a four-probe resistance measuring meter, with a Keithley 2400 Source Measure Unit. The transparency of the thin film was tested by an ultraviolet-visible spectrometer (UV-2450). The AFM images were obtained in tapping mode (Nanoscope III, Digital Instrument). The Grazing-incidence wide-angle X-ray scattering (GIWAXS) measurement was performed on a laboratory beamline system (Xenocs Inc. Xeuss 2.0) with an X-ray wavelength of 1.54 Å and sample to detector distance of 17.1 cm. An incidence angle of 0.2° was used. Diffraction images were recorded on a Pilatus 1 M detector (Dectris Inc.) and processed using the Nika software package, in combination with WAXS Tools. The Raman spectra were obtained at 532 nm excitation (LabRAM HR Evolution). The electrochemical impedance spectroscopy (EIS) measurements were conducted using a CHI600e electrochemical workstation (CH Instruments).

**Electrophysiological measurements**. Two working electrodes were placed on forearm, and the reference electrode was placed on the back of wrist for sEMG. The electrodes were connected with commercial signal recording equipment (DueLite) and sEMG signals were digitally filtered in MATLAB by a six-order Butterworth filter. And the low cut-off frequency is 10 Hz while the high cut-off frequency is 450 Hz. For ECG, two working electrodes and reference electrodes were placed on arm by using the Einthoven triangle principle. During EOG measurement, the working electrodes were placing on the forehead and lower eyelids, respectively. For easy handling and stable connection to electrophysiological recording instrument, SEBS or decal transfer tattoo paper were sometimes assisted as an additional substrate, and evaporated Ag films were aided in the connection between soft-hard interfaces.

**Motion artifact characterization**. The electromechanical vibrator was placed near the working electrodes (about 2 cm) to induce skin vibration, which is similar to vibration in arm movement.

**Facial sEMG measurements and controlling the robotic hand**. Two working electrodes of PTG were placed on levator labii superioris, and one reference electrode was placed behind the ear. Ag/AgCl and PEDOT:PSS electrodes were placed on the same position on the other side of face as control. Laser speckle contrast imaging was recorded in SIM BFI-WF System (Wuhan SIM Opto-Technology Co. Ltd., China). The monitor detector was set 25–30 cm above the volunteer' face, with a maximum field of view of $14.8 \times 14.8$ cm so that the entire facial area could be illuminated. The frequency used for the blood perfusion image was 1 s, and the size of the image was $512 \times 512$ pixels. To control the robotic hand, three pairs of PTG were attached on zygomaticus, risorius and corrugator to extract sEMG signals of the index, middle finger, and wrist of the robotic hand, respectively.

**Long-time monitoring of EEG**. Brain activity was evaluated by EEG signals using a commercial signal recording equipment (DueLite). The EEG signal was measured by mounting two working electrodes (PTG on tattoo paper with Ag as connections) in the frontal region of the brain and one reference electrode behind the ear. During the monitoring of 12 hours, the volunteer slept for 1 h, and exercised for 0.5 h, finally recovered to calm without taking off PTG electrodes. The EEG signal was analyzed by Matlab and Fast Fourier Transformation.

## Data availability

The authors declare that all the data that support the findings of this study are available within the article and its Supplementary information files.

## Code availability

The codes used in this study are available from the corresponding author upon request.

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

## Acknowledgements

This work is supported by National Natural Science Foundation of China (22072006, 21903007, and 22005036), Young Thousand Talents Program (110532103), Beijing Normal University Startup funding (312232102), Beijing Municipal Science & Technology Commission (No. Z191100000819002), and the Fundamental Research Funds for the Central Universities (310421109).

## Author contributions

N.L. designed and supervised the research. Y.Z. performed the experiments. S.Z. and X.G. did GIWAXS characterization. Y.Z., T.Y., Y.Z., W.J., and G.Y. helped in the data collection. H.C. and Y.Z. did the laser speckle contrast imaging. C.H. did the Young's modulus experiments. Y.Z. and C.L. helped analyze the EEG data. N.L. and Y.Z. co-write the manuscript. All authors discussed and revised the manuscript.

## Competing interests

The authors declare no competing interests.
