## [Peer Review File · Nature Communications]

REVIEWER COMMENTS

Reviewer #1 (Remarks to the Author):

The main innovation in this work is the use of CVD-grown graphene as the substrate for the spin-coating of PEDOT:PSS so as to enhance the morphological ordering and packing of PEDOT:PSS, and therefore its conductance. A set of systematic characterizations have been carried out to understand the changes of the PEDOT:PSS's molecular and packing ordering. Also, the transferred graphene and PEDOT:PSS bilayer exhibits stretchability up to 40% strain, which is another interesting point. The overall application direction proposed and demonstrated for this material design is for the on-body measurement of electrophysiological signals, to achieve a higher signal-to-noise ratio. Although the presented material design in this work carries some new phenomenon and achieved some interesting results, the technological advancement in reference to the state-of-the-art alternatives has been convincingly illustrated. Moreover, some results also deserve deeper understanding. Here are the comments:

1. There has already been a number of different approaches established for enhancing the conductivity of PEDOT:PSS. Compared to these approaches, it seems like the achieved conductance is on the similar level. For example, in the work of "Wang et al., Sci. Adv. 2017;3: e1602076", the sheet resistance of the PEDOT:PSS with some ionic liquid additives reached the lowest value of 59 ohms/sq, which is not much higher than the lowest value of 45 ohms/sq in this work. As such, to illustrate the technological value of this work, the authors need to make a comprehensive comparison in all the major aspects to the reported approaches for enhancing the conductivity of PEDOT:PSS. For doing so, the sheet resistances reported in this paper needs to be converted to conductivity in the unit of S/cm.
2. For the claim high transconductance as an advantage from this bilayer film, it seems like this is not correlated to the interactions between PEDOT:PSS and graphene. So if it only comes from the reduced thickness of PEDOT:PSS, then there is not much uniqueness here in achieving the high transconductance.
3. What is the mechanism underlying the stretchability of this bilayer film? Why the multi-layer graphene can lead to even better stretchability?
4. For the demonstration of using the bilayer conducting film to measure the electrophysiological signals from the skin surface, the comparison was made with Ag/AgCl gel. However, this is not a really fair or meaningful comparison, as the Ag/AgCl gel is a very different material system with very different charge conducting mechanism. Since PEDOT:PSS with different designs has already been utilized in a lot of works for measuring the electrophysiological signals, the comparison should be made with a dry electrode based on another type of dry-state PEDOT.
5. One important reason for using gel as the electrode on skin is to achieve the stable adhesion to the skin. Replacing it with the bilayer film as the dry electrode, the authors didn't explain about how such stable adhesion is achieved.
6. The last demonstration of using the measured EMG signal to control the robotic arm doesn't really show the advantage of the improved signal-to-noise ratio obtained from the bilayer dry electrodes.

Overall, this paper needs to be carefully revised to show its technological advancement, before it can be possibly further considered for Nat Comm.

Reviewer #2 (Remarks to the Author):

The manuscript by Zhao et al. reports an interesting approach to employ PEDOT:PSS as a transfer medium for CVD graphene, which produces a bilayer composite material termed as PTG. The process essentially eliminates the need to remove the sacrificial transfer medium of PMMA in the conventional technique. As a transparent electrode, the excellent optoelectronic performances of PTG comes from the synergistic charge transfer effects between graphene and PEDOT:PSS supported by Raman spectroscopy and GIWAXS. In addition, PTG also show decent mechanical deformability to function as conformal skin electrodes to acquire electrophysiological signals with low motion artifacts. I recommend the manuscript to be considered for publication after minor revisions regarding the following aspects:

(1) The origin of the stretchability of PTG is less clear. I wonder whether it comes from certain intrinsic mechanisms or structural reasons (e.g. surface wrinkles formed during the transfer process). Additional discussions/experimental evidences should be provided. In addition, the reason for improved stretchability upon increasing the number of underlying graphene layers should be explained.

(2) The authors claim the addition of bis(trifluoromethane) sulfonimide lithium salt into PEDOT:PSS improves the electrical conductivity and the mechanical stretchability. Although it is intuitive that ionic compound may boost the conductivity of PEDOT:PSS, the benefit of BSL salt on mechanical deformability requires additional clarifications and explanations.

(3) In order to put the optoelectronic performances of PTG into the research context, a table summarizing the state-of-the-art of graphene transparent electrodes should be provided.

REVIEWER COMMENTS

Reviewer #1 (Remarks to the Author):

The main innovation in this work is the use of CVD-grown graphene as the substrate for the spin-coating of PEDOT:PSS so as to enhance the morphological ordering and packing of PEDOT:PSS, and therefore its conductance. A set of systematic characterizations have been carried out to understand the changes of the PEDOT:PSS's molecular and packing ordering. Also, the transferred graphene and PEDOT:PSS bilayer exhibits stretchability up to 40% strain, which is another interesting point. The overall application direction proposed and demonstrated for this material design is for the on-body measurement of electrophysiological signals, to achieve a higher signal-to-noise ratio. Although the presented material design in this work carries some new phenomenon and achieved some interesting results, the technological advancement in reference to the state-of-the-art alternatives has been convincingly illustrated. Moreover, some results also deserve deeper understanding. Here are the comments:

We thank the reviewer for the positive comments of our work and welcome the opportunity to clarify the questions. Below we address the reviewer's comments point-by-point.

1. There has already been a number of different approaches established for enhancing the conductivity of PEDOT:PSS. Compared to these approaches, it seems like the achieved conductance is on the similar level. For example, in the work of "Wang et al., Sci. Adv. 2017;3: e1602076", the sheet resistance of the PEDOT:PSS with some ionic liquid additives reached the lowest value of 59 ohms/sq, which is not much higher than the lowest value of 45 ohms/sq in this work. As such, to illustrate the technological value of this work, the authors need to make a comprehensive comparison in all the major aspects to the reported approaches for enhancing the conductivity of PEDOT:PSS. For doing so, the sheet resistances reported in this paper needs to be converted to conductivity in the unit of S/cm.

Response: We thank the reviewer for raising the concern. In this manuscript, we aim at preparing ultra-conformal skin electrodes for achieving high-quality electrophysiological signal detection. CVD graphene is the thinnest electrode material, and PEDOT:PSS with SDS and ionic liquid is able to form a thin and continuous film (~ 100 nm) on it through spin-coating. These two conductive materials form a synergistic film, named PTG (~ 100 nm). When evaluating the conductivity of a thin film, the unit of Ω/sq is often used. As an alternative, the unit of S/cm is to express as bulk conductivity. To convert the sheet resistance (R_{sh} in Ω/sq) to conductivity (S/cm), the thickness of films (t) was carefully measured, and the conductivity can be equal to $\frac{1}{R_{\text{sh}} \times t}$. Following your suggestion, we compared our PTG with most approaches for enhancing the conductivity of PEDOT:PSS reported in recent 5 years in terms of film thickness (nm), conductivity (S/cm), and sheet resistance (Ω/sq) (new Table 1). Considering both conductivity and transparency, our PTG is superior to PEDOT:PSS as a transparent and conductive electrode particularly ultra-conformal electrode.

On the other hand, referring to the work of "Wang et al., Sci. Adv. 2017;3: e1602076", in which the

ionic liquid was as high as 50 wt%, we further increased the concentration of ionic liquid. With the addition of a much lower concentration of ionic liquid (5.0 wt%), the conductivity can reach $\sim 24 \Omega/\text{sq}$ (4142 S/cm), which is comparable to 4100 S/cm of the referred work. Such high conductivity confirms not only the enhancement effect of ionic liquid, but also the synergistic effect of graphene. We could hypothesize that in PTG system, at an equal concentration of ionic liquid to the referred work, the conductivity could be extremely high. This has been preliminary verified by $\sim 15 \Omega/\text{sq}$ for PTG at 7.0 wt% ionic liquid. However, higher concentration of ionic liquid results in non-uniform PTG film due to gelation of PEDOT:PSS solution. To balance the transparency and achieve ultrathin electrode, our optimal PTG condition is with the addition of 1.0 wt% ionic liquid.

The new experimental data for PTG at 5.0 and 7.0 wt% ionic liquid, the new comparison table of figure of merits of PEDOT:PSS, graphene and PTG films (Figure 2b, Table 1, Table S2), and corresponding discussion has been added in the revised manuscript. Detailed revision is as below.

In the revised manuscript:

“At 5.0 wt% BSL concentration, the conductivity of PTG can reach $\sim 4142 \text{ S/cm}$, which exceeds the record high value (4100 S/cm) for PEDOT:PSS only film (Table S2). However, the gelation of PEDOT:PSS solution at high BSL concentration makes obstacles for the formation of uniform PEDOT:PSS carrier layers by spin-coating, leading to a thick PTG film. We also compared our PTG with most approaches recently reported for enhancing the conductivity of PEDOT:PSS in terms of film thickness (nm), conductivity (S/cm), and sheet resistance (Ω/sq) (Table 1). Considering both conductivity and sheet resistance, our PTG is superior to PEDOT:PSS as transparent and conductive electrodes, particularly as ultra-conformal electrodes.”

Figure 2. Optoelectronic performances of PTG. (b) Sheet resistance and transmittance (at $\lambda = 550 \text{ nm}$) of PTG films at various BSL (1.0 wt% SDS) concentrations when forming PEDOT:PSS carrier layer.

Table 1. Sheet resistance (R_{sh}), Thickness and Conductivity of different PEDOT:PSS film.

Methods	R_{sh} (Ω/sq)	Thickness (nm)	Conductivity (S/cm)	Application	Reference
Ionic liquid additive	59	-	3100	Stretchable circuit	Sci. Adv. 3 , e1602076 (2017)
H ₂ SO ₄ soaking	46	-	4000	Solar cell	Adv. Mater. 27 , 2317–2323 (2015)
HNO ₃ soaking	-	27	4100	Solar cell	Adv. Electron. Mater. 1 , 1500121 (2015)
Zn(TFSI) ₂ doping	-	-	4115	Solar cell	Joule 3 , 2205–2218 (2019)
MSA soaking	50	79	2540	Solar cell	ACS Appl. Mater. Interfaces 7 , 15314–15320 (2015)
Methanol/MSA soaking	43	65	3560	Solar cell	ACS Appl. Mater. Interfaces 7 , 16287–16295 (2015)
PTG	45	80	2850	Electrophysiology detection	This work
	24	100	4142		

Table S2. Comparison of the thickness and electrical conductivity of PTG and pure PEDOT:PSS films at various preparation conditions.

Sample	Preparation condition	R_{sh} (Ω/sq)	Thickness (nm)	Conductivity (S/cm)
Pure PEDOT:PSS	SDS 1.0 wt%+BSL 1.0 wt%, 4000 rpm	70.1±0.2	83.5±2.2	1620
Pure PEDOT:PSS	SDS 1.0 wt%+BSL 1.0 wt%, 3000 rpm	80.2±0.4	185.0±4.1	674
PTG (1G+PEDOT:PSS)	SDS 1.0 wt%+BSL 1.0 wt%, 3000 rpm	43.8±0.9	79.7±3.7	2850
PTG (1G+PEDOT:PSS)	SDS 1.0 wt%+BSL 5.0 wt%, 3000 rpm	24.1±0.3	100±4.8	4142
PTG (1G+PEDOT:PSS)	SDS 1.0 wt%+BSL 7.0 wt%, 3000 rpm	14.6±0.7	-	-
Graphene		600±2	~1	

2. For the claim high transconductance as an advantage from this bilayer film, it seems like this is not correlated to the interactions between PEDOT:PSS and graphene. So if it only comes from the reduced thickness of PEDOT:PSS, then there is not much uniqueness here in achieving the high transconductance.

Response: We thank the reviewer for the comments. We first compared the electrical conductivity of pure PEDOT:PSS and PTG films at the same thickness and concentration of additives (SDS+BSL). At both thicknesses of ~ 80 nm, the conductivity of PTG (~ 2850 S/cm) is ~ 2 times higher of PEDOT:PSS (1620 S/cm, new Table S2), indicating that the interaction between PEDOT:PSS and graphene plays an important role in increasing the conductivity of PEDOT:PSS.

We also noted that at the same spin-coating condition with same concentration of additives (SDS+BSL) in PEDOT:PSS solution, thickness of pure PEDOT:PSS (~ 185 nm) film is ~ 2.3 times thicker of PTG (~ 80 nm) film. The interference of wetting can be excluded, because contact angles of PEDOT:PSS solution on graphene and SiO₂/Si substrates are 74.6° and 74.5° respectively. The obviously decreased thickness of PTG is possibly due to the strong π - π interaction between PEDOT:PSS and graphene, leading to reordering of PEDOT on graphene. The reduced thickness

resulted from reordering of PEDOT was also observed in the literature (*ACS Appl. Mater. Interfaces* **10**, 29115-29126 (2018)).

Above new experimental data at various thicknesses and spin-coating condition were compared in a table (Table S2), and corresponding discussion has been added in the revised manuscript. Detailed revision is as below.

In the revised manuscript:

“It is noteworthy that at the same spin-coating condition with same concentration of additives (SDS+BSL) in PEDOT:PSS solution, PEDOT:PSS only (~ 185 nm) film is ~ 2.3 times thickness of PTG (~ 80 nm, 1G) film (Table S2). Because contact angles of PEDOT:PSS solution on graphene and SiO₂/Si substrates are similar (74.6° and 74.5°, respectively, Figure S6), the obviously decreased thickness of PTG is highly likely due to the strong π - π interaction between PEDOT:PSS and graphene, leading to reordering of PEDOT on graphene. On the other hand, at the same thickness (~ 80 nm) of PTG and pure PEDOT:PSS films, the conductivity of PTG (~ 2850 S/cm, 1G) is ~ 2 times high of PEDOT:PSS (1620 S/cm). These indicate that the synergistic enhancement between PEDOT: PSS and graphene is the main contributor to the high electrical conductivity of PTG. Detailed preparation condition and electrical property of the PTG and pure PEDOT:PSS films were summarized in Table S2.”

Table S2. Comparison of the thickness and electrical conductivity of PTG and pure PEDOT:PSS films at various preparation conditions.

Sample	Preparation condition	R _{sh} (Ω /sq)	Thickness (nm)	Conductivity (S/cm)
Pure PEDOT:PSS	SDS 1.0 wt%+BSL 1.0 wt%, 4000 rpm	70.1±0.2	83.5±2.2	1620
Pure PEDOT:PSS	SDS 1.0 wt%+BSL 1.0 wt%, 3000 rpm	80.2±0.4	185.0±4.1	674
PTG (1G+PEDOT:PSS)	SDS 1.0 wt%+BSL 1.0 wt%, 3000 rpm	43.8±0.9	79.7±3.7	2850
PTG (1G+PEDOT:PSS)	SDS 1.0 wt%+BSL 5.0 wt%, 3000 rpm	24.1±0.3	100±4.8	4142
PTG (1G+PEDOT:PSS)	SDS 1.0 wt%+BSL 7.0 wt%, 3000 rpm	14.6±0.7	-	-
Graphene		600±2	~1	

3. What is the mechanism underlying the stretchability of this bilayer film? Why the multi-layer graphene can lead to even better stretchability?

Response: We thank the reviewer for raising this concern. The underlying graphene of PTG can lead to better electrical stability upon tensile strain. To understand this mechanism, we newly performed a systematic morphology study on graphene, pure PEDOT:PSS and PTGs (1G+PEDOT:PSS, 2G+PEDOT:PSS) on SEBS elastomer (50 μ m thick) before and after mechanical stretching at 20% strain by AFM. In the high magnification images, the granule-to-nanofibril transition was clearly observed on PTGs (Figure S5). Applied 20% strain on SEBS, cracks appeared on all the films, which are mostly perpendicular to the strain direction. However, arising from the underlying graphene, there are obviously less cracks and vaguer parallel wrinkles on PTGs than those on pure PEDOT:PSS (new Figure 2e). It indicates that a smaller strain was transferred onto

above PEDOT:PSS in PTG due to the larger sliding between graphene and substrates. With the increase of graphene layer number, this factor is strengthened accompanied with the formation of more percolation pathways. Multilayer graphene for stretchable electrodes has been reported in the literature of “Bao et al., *Sci. Adv.* 2017; 3: e1700159”, which was also explained as the maintenance of percolation pathway and sliding over each other.

New morphological data of graphene, pure PEDOT:PSS and PTGs before and after tensile strain were added in the revised manuscript (Figure 2e) and supporting information (Figure S5) with corresponding mechanism understanding. Detailed revision is as below.

In the revised manuscript:

“The underlying graphene of PTG results in the better electrical stability upon tensile strain. To understand this mechanism, we performed a systematic morphology study on pure PEDOT:PSS and PTGs (1G+PEDOT:PSS, 2G+PEDOT:PSS) on SEBS elastomer (50 μm thick) before and after mechanical stretching at 20% strain by AFM. In the high magnification images, the granule-to-nanofibril transition was clearly observed on PTGs (Figure S5). Applied 20% strain on SEBS, cracks appeared on all the films, which are mostly perpendicular to the strain direction. However, arising from the underlying graphene, there are obviously less cracks and vaguer parallel wrinkles on PTGs than those on pure PEDOT:PSS film (Figure 2e). It indicates that a smaller tensile strain was transferred onto PTG due to the large sliding between graphene and elastomer. With the increase of graphene layer number, this factor is strengthened accompanied with the formation of more percolation pathways, leading to better electrical durability at strain. This is consistent with the report for multilayer graphene as stretchable electrodes^{40,41}. Thus, to compromise the transparency, conductivity and mechano-electrical stability of PTG, our optimal condition for following epidermal electrophysiology is bi-layer graphene transferred by PEDOT:PSS at both SDS and BSL concentrations of 1.0 wt%.”

Figure 2. (e) Morphological comparison of pure PEDOT:PSS and PTGs (mono- and bi-layer graphene) upon 20% strain by AFM. The strain direction is perpendicular to the cracks and parallel to the wrinkles which are generated by Poisson effect.

Figure S5. Morphological comparison of pure PEDOT:PSS and PTGs (mono- and bi-layer graphene) on SEBS by AFM.

4. For the demonstration of using the bilayer conducting film to measure the electrophysiological signals from the skin surface, the comparison was made with Ag/AgCl gel. However, this is not a really fair or meaningful comparison, as the Ag/AgCl gel is a very different material system with very different charge conducting mechanism. Since PEDOT:PSS with different designs has already been utilized in a lot of works for measuring the electrophysiological signals, the comparison should be made with a dry electrode based on another type of dry-state PEDOT.

Response: We thank the reviewer for raising this point. One motivation to develop dry electrodes is to replace commercial Ag/AgCl gel electrodes. Because the electrolytic gel may irritate the skin and will dry in a certain period, making it impossible for long-term monitoring and diagnosis. Moreover, with the gel interfacing on skin, electrode displacement with respect to skin will cause severe motion artifact. Therefore, in this work we aim at ultra-conformal dry electrodes for low-motion artifact epidermal electrophysiology, and compared the sEMG signal detection with Ag/AgCl gel electrodes.

Following your suggestion, we additionally fabricated pure PEDOT:PSS film at the same preparation condition (eg.: additives in solution and spin-coating), and compared the sEMG signal detection with our PTG. As they are both ultrathin and conformal to skin, their interfacial impedances are comparable to commercial Ag/AgCl, and at 100 Hz, the impedance of pure PEDOT:PSS ($\sim 42 \text{ k}\Omega$) is higher than PTG ($\sim 32 \text{ k}\Omega$) (Figure 5d). In sEMG detection in static and dynamic status (Figure 5h), the amplitude of baseline noise of pure PEDOT:PSS increased significantly from $20.2 \pm 1.9 \mu\text{V}$ to $41.2 \pm 3.2 \mu\text{V}$ (new Figure 5f), and the SNR (Figure S14) decreased from $18 \pm 0.5 \text{ dB}$ to $13 \pm 0.4 \text{ dB}$. On the other hand, PTG electrodes demonstrated a relatively stable baseline noise ($18.4 \pm 1.1 \mu\text{V}$ vs. $23 \pm 2.9 \mu\text{V}$) and SNR ($23 \pm 0.7 \text{ dB}$ vs. $21 \pm 0.3 \text{ dB}$). This is because without the additional graphene, pure PEDOT:PSS film is very brittle and its conductivity would be easily affected by strain. In dynamic status, pure PEDOT:PSS film would severely crack with an obviously alleviated conductivity, leading to unstable electrophysiological signal detection. In particular the situation on largely deformed skin, pure PEDOT:PSS may even fail in sEMG detection. For instance in facial sEMG detection (newly added Figure 6), because the superficial portions of face muscle are uneven, full of wrinkles, and large-shape changed, pure PEDOT:PSS and Ag/AgCl gel electrodes are not able to record sEMG or over a period.

We also compared our work with other PEDOT:PSS electrodes for electrophysiology (Table S3). One is to directly inkjet PEDOT:PSS on commercial decal transfer tattoo paper as a conformal dry

electrode for electrophysiology (*Adv. Sci.* **5**, 1700771 (2018)). The other is to blend PEDOT:PSS with water soluble PU to form adhesive film (*Nat. Commun.* **11**, 4683 (2020)). The former one has limitation in mechano-electrical stability, and the latter one is thick, opaque and not very conductive. In contrast, our PTG is highly conductive, ultrathin, relatively transparent and mechano-electrically stable, showing low-motion artifact electrophysiology with high SNR both in static and dynamic status. To strengthen the application, we optimized the measurements of ECG, EOG, and sEMG by PTG, and improved human-machine interfacing by multi-channel extraction of sEMG.

New comparisons of impedance analysis, and EOG, ECG, sEMG measurements of PTG, pure PEDOT:PSS and Ag/AgCl both in static and dynamic status (Figure 5d-h, Figure S14-15), an improved human-machine interfacing by multi-channel extraction of sEMG (Figure 5i, Figure S17-22, Video S1), a new table compared the capability of electrophysiological detection of PTG and various PEDOT:PSS (Table S3), and corresponding discussion have been added in the revised manuscript. Detailed revision is as below.

In the revised manuscript:

“PTG showed comparable interfacial impedance with PEDOT:PSS dry electrodes and commercial Ag/AgCl gel electrodes, and at 100 Hz, the impedance ($\sim 32 \text{ k}\Omega$) is lower than PEDOT:PSS ($\sim 42 \text{ k}\Omega$) and the Ag/AgCl ($\sim 45 \text{ k}\Omega$).

With low interfacial impedance, PTG was next applied in electrophysiological detection, with pure PEDOT:PSS and Ag/AgCl as control. EOG is to record eye potential and helps noninvasively assess the oculomotor function. Electrodes were adhered on the forehead and lower eyelids. Three types of electrodes showed comparable peaks and valleys with clear eye health information (Figure 5e). ECG is another important electrophysiological signal for detecting heart activity. Dry electrodes were attached on the forearm for measuring ECG signals (Figure 5g). All of the rhythmic-relevant parameters, such as P, Q, R, S, and T waves (Figure S12), can be clearly identified, which is crucial in clinical diagnosis. To record the muscle bioelectricity, sEMG was detected by adhering electrodes on the forearm (Figure 5h). In static state, the sEMG signal amplitude obtained by three kinds of electrodes were comparable, while signal-to-noise ratio (SNR) values obtained by PTG ($23 \pm 0.7 \text{ dB}$) is higher than Ag/AgCl electrodes ($19 \pm 0.5 \text{ dB}$) and PEDOT:PSS electrodes ($18 \pm 0.5 \text{ dB}$). (Figure S14). As electrolytic gel will dry and induce electrode displacement, the major challenge of Ag/AgCl electrodes in electrophysiological detection is the motion artifact. Hence, the motion artifact was evaluated here by baseline noise. We used electromechanical vibrator to mimic skin vibration in arm movement. The electromechanical vibrator was placed near the working electrodes (Figure S15). The average baseline noise of PTG in sEMG recording was just $23 \pm 2.9 \mu\text{V}$, 1.7 times smaller than $38.8 \pm 2.8 \mu\text{V}$ of Ag/AgCl and 1.8 times smaller than $41.2 \pm 3.2 \mu\text{V}$ of PEDOT:PSS in the same condition (Figure 5f). Thus, in dynamic status, PTG electrodes demonstrated a relatively stable SNR ($21 \pm 0.3 \text{ dB}$) than those of Ag/AgCl electrodes ($14 \pm 0.4 \text{ dB}$) and pure PEDOT:PSS ($13 \pm 0.4 \text{ dB}$) (Figure S14). The improved sEMG signal quality in both static and dynamic status can be attributed to the better conformability and mechano-electrical stability of PTG. Motion artifact was also evaluated in ECG (Figure 5e, f). Transited from the static to dynamic status, the baseline noise detected by PTG is relatively stable than Ag/AgCl and PEDOT:PSS electrodes. sEMG is more sensitive to the vibration compared with ECG, because the detected bandwidth (20 – 2000 Hz vs. 0.05-100 Hz) is higher, and muscle contraction comes along with large skin deformation. Although pure PEDOT:PSS was measured in above tests, it is noted that without the additional graphene, pure

PEDOT:PSS film is very brittle and easily break out upon strain. In addition, the infrared image (Figure S16) showed that a minimal level of heat was accumulated upon mounting the PTG electrodes on a volunteer's forearm. Overall, our ultra-conformal PTG exhibited less motion artifact in electrophysiological signal detection, suggesting PTG is potential for accurate human-machine interfacing.

Figure 5. Electrophysiological signals detection of PTG dry electrodes. (d) The impedance analysis of PTG (red), pure PEDOT:PSS (blue) and Ag/AgCl (black) electrodes. (e) EOG measured by PTG (red), pure PEDOT:PSS (blue) and Ag/AgCl (black) electrodes, showing peaks and valleys corresponding to the eyelid movement. (f) sEMG baseline noise comparison of the three types of electrodes in static and dynamic status. (g) ECG measured by PTG (red), pure PEDOT:PSS (blue) and Ag/AgCl (black) electrodes without and with vibration. Characteristic P, Q, R, S and T waves are clearly identified. (h) sEMG measured by PTG (red), pure PEDOT:PSS (blue) and Ag/AgCl (black) electrodes without and with vibration, showing lower-motion artifact detected by PTG. Insets of (e, g, h) are photos of PTG electrodes for EOG, ECG, sEMG. (i) Driving the robotic hand to show various gestures (victory, Arabic numerals of six and three, and holding a centrifuge tube) based on multi-channel sEMG signals extracted by PTG electrodes. Five PTG electrode pairs were adhered on designated locations for acquiring sEMG signals of thumb, index, middle, ring and little fingers of the volunteer, and remotely driving the movement of robotic hand.

The sEMG signals are generated by muscular excitation-contraction, and converting the sEMG signals to control artificial limbs is urgently expected clinically. Firstly the sEMG signals were extracted by our electrodes and then converted to Pulse Width Modulation (PWM) pulses to drive the servo motor (Figure S17-S21). Even with external interference (Figure S21), sEMG signal

extracted by PTG can still precisely control the mechanical claw due to its low-motion artifact detection. Five electrode pairs were adhered on designated locations to respectively detect sEMG signals of thumb, index, middle, ring and little finger of the volunteer, and further control the movement of corresponding fingers of the robotic hand (Figure S22). As shown in Figure 5i and Video S1, various gestures of the robotic hand, such as victory, Arabic numerals of six and three, and holding a centrifuge tube, were demonstrated. Equipped with complicated digital processing and learning algorithms, our ultra-conformal and dry PTG skin electrodes could be applicable in controlling artifact limbs with more complexity and accuracy.”

Figure S14. The baseline noise of ECG (a) and SNR of sEMG (b) measured by Ag/AgCl, PEDOT:PSS and PTG electrodes without and with vibration respectively. The signal-to-noise ratio (SNR) compares the level of signal to noise power, and is calculated as following: $SNR_{dB} = 20\lg\left(\frac{A_{signal}}{A_{noise}}\right)$, where A is root mean square amplitude.

Figure S15. Evaluation of baseline noise in sEMG detection using an electromechanical vibrator to mimic skin vibration in arm movement. The electromechanical vibrator was placed near the working electrodes at an approximate distance of 2 cm. To stably connect PTG to the electrophysiological recording instrument, evaporated Ag films were aided in the connection between soft-hard interfaces.

Figure S17. The overall functional block diagram. It includes front-end analog circuit acquisition and back-end digital signal filtering processing.

$$\begin{aligned}
 f &= \frac{1}{2\pi RC} \\
 &= \frac{1}{2 \times 3.14 \times 4.7 \times 10^4 \times 10^{-9}} \\
 &= 3000\text{Hz} \\
 \text{Gain} &= 1 + \frac{100\text{k}\Omega}{33\text{k}\Omega} = 4
 \end{aligned}$$

Figure S18. Circuit diagram of a low-pass filter (LPF).

Figure S19. Circuit diagram of a filter, 55-2500 Hz with a gain of 177.

Figure S20. Circuit diagram of the envelope mode with a gain of 22.

Figure S21. (a) Schematic illustration of PTG electrodes on the arm for sEMG detection and robotic claw controlling process. Firstly the sEMG signals were extracted by PTG electrodes and then converted to Pulse Width Modulation (PWM) pulses to drive the servo motor. (b) Demonstration of controlling a robotic claw during vibration. We knocked our arm via a pen to mimic skin vibration in dynamic state, and did not observe any interference while controlling the robotic claw.

Figure S22. Five electrode pairs were adhered on designated locations to respectively control the movement of corresponding fingers of the robotic hand.

Table S3. Comparison of the capability of electrophysiological detection of PTG with various PEDOT:PSS.

Methods	Property	Detection	Control experiments	Application	Reference
PEDOT:PSS tattoo electrode	520 Ω /sq (150 nm thick, transparent), conductive at 5% strain	ECG, sEMG	Similar SNR and noise voltage vs. Ag/AgCl in static status	No human-machine interfacing	Adv. Sci. 5 , 1700771 (2018)
PEDOT:PSS/PU/D-sorbitol hybrid film	545 S/cm (20 μ m thick, opaque), stably conductive at 30% strain	ECG, sEMG	Lower noise voltage vs. Ag/AgCl for ECG in static status	One-channel human-machine interfacing	Nat. Commun. 11 , 4683 (2020)
PTG	4142 S/cm (25 Ω /sq, 100 nm thick, transparent), conductive at 40% strain	ECG, EOG, sEMG, facial sEMG at largely deformed position	Higher SNR and lower noise voltage vs. Ag/AgCl both in static and dynamic status	Multi-channel human-machine interfacing	This work

5. One important reason for using gel as the electrode on skin is to achieve the stable adhesion to the skin. Replacing it with the bilayer film as the dry electrode, the authors didn't explain about how such stable adhesion is achieved.

Response: We thank the reviewer for raising this concern.

In our opinion, gel in Ag/AgCl helps the electrode form conformal contact with skin, whereas double side tape around the gel assists the electrode stably adhere to the skin. When the skin is rough, wrinkled or easily deformed, the commercial electrode would be easily detached, thus requiring additional media to maintain stable contact. The main challenges for using gel electrodes are the severe skin discomfort for long-time use and the motion artifact caused by the displacement. The

contact impedances of gel electrodes were monitored to be increased with time going on (new Figure S13). The degradation indicates that the contact of gel electrodes becomes unstable in a long time, which has also been reported in *Adv. Mater.* 2020, 2001496. To conquer these challenges, dry electrodes without gel are boosting for wearable healthcare.

There are basically two ways to achieve dry electrodes with stable adhesion to skin. One is to utilize intrinsically adhesive material, such as adhesive polymer or adhesive tape. The other way is to prepare ultrathin film as electrodes. For example, ultrathin Au film (~ 300 nm) was reported to be self-adhesive and conformable on skin, and can be used in biopotential recording with less-motion artifact (*Adv. Funct. Mater.* 2018, 1803279). The thickness of our PTG is ~ 80 nm, forming intimate contact on skin (new Figure 5a). To exhibit the stable adhesion on skin, we transferred PTG on porcine skin, and observed if they form conformal interface in between by SEM. The cross-sectional view clearly showed that PTG is conformable to the uneven and curved surface of the skin (new Figure 5a-b and Figure S11). Besides, the Young's modulus of PTG measured by AFM is about 640 kPa (new Figure 5c), which matches the modulus of human skin. Taken together, the ultrathin and low Young's modulus of PTG lead to conformal contact on skin, thus facilitating accurate and stable electrophysiological signal detection without gel.

New photo image of PTG on the junction of a finger (Figure 5a), SEM image of PTG on porcine skin (Figure 5b and Figure S11), AFM Young's modulus map of PTG (Figure 5c, Figure S10), the dependence of contact impedance of gel electrodes on time (Figure S13), and corresponding discussion have been added in the revised manuscript.

In the revised manuscript:

“Low-motion artifact electrophysiological monitoring. Interfacial impedance is a critical parameter in epidermal electrophysiological measurement. Without additional gel, dry electrodes would reduce the contact impedance by forming conformal interface with skin. According to the Young's modulus map under AFM, average Young's modulus of PTG is about 640 kPa (Figure 5c, Figure S10), which matches the modulus of human stratum corneum. Transferred onto a human ring finger full of wrinkles, PTG exhibited intimate contact on it (Figure 5a). The conformability is also confirmed by SEM imaging of PTG on porcine skin (Figure 5b, Figure S11). The ultrathin PTG with low Young's modulus leads to conformal contact with wrinkled skin^{45, 46}. The interfacial impedance analysis between electrodes and skin was conducted on the human arm with electrodes placed on the surface of it (Figure 5d). PTG showed comparable interfacial impedance with PEDOT:PSS dry electrodes and commercial Ag/AgCl gel electrodes, and at 100 Hz, the impedance (~ 32 k Ω) is lower than PEDOT:PSS (~ 42 k Ω) and the Ag/AgCl (~ 45 k Ω).

With low interfacial impedance, PTG was next applied in electrophysiological detection, with pure PEDOT:PSS and Ag/AgCl as control. EOG is to record eye potential and helps noninvasively assess the oculomotor function. Electrodes were adhered on the forehead and lower eyelids. Three types of electrodes showed comparable peaks and valleys with clear eye health information (Figure 5e). ECG is another important electrophysiological signal for detecting heart activity. Dry electrodes were attached on the forearm for measuring ECG signals (Figure 5g). All of the rhythmic-relevant parameters, such as P, Q, R, S, and T waves (Figure S12), can be clearly identified, which is crucial in clinical diagnosis. To record the muscle bioelectricity, sEMG was detected by adhering electrodes on the forearm (Figure 5h). In static state, the sEMG signal amplitude obtained by three kinds of

electrodes were comparable, while signal-to-noise ratio (SNR) values obtained by PTG (23 ± 0.7 dB) is higher than Ag/AgCl electrodes (19 ± 0.5 dB) and PEDOT:PSS electrodes (18 ± 0.5 dB). (Figure S14). As electrolytic gel will dry and induce electrode displacement, the major challenge of Ag/AgCl electrodes in electrophysiological detection is the motion artifact. Hence, the motion artifact was evaluated here by baseline noise. We used electromechanical vibrator to mimic skin vibration in arm movement. The electromechanical vibrator was placed near the working electrodes (Figure S15). The average baseline noise of PTG in sEMG recording was just 23 ± 2.9 μ V, 1.7 times smaller than 38.8 ± 2.8 μ V of Ag/AgCl and 1.8 times smaller than 41.2 ± 3.2 μ V of PEDOT:PSS in the same condition (Figure 5f). Thus, in dynamic status, PTG electrodes demonstrated a relatively stable SNR (21 ± 0.3 dB) than those of Ag/AgCl electrodes (14 ± 0.4 dB) and pure PEDOT:PSS (13 ± 0.4 dB) (Figure S14). The improved sEMG signal quality in both static and dynamic status can be attributed to the better conformability and mechano-electrical stability of PTG. Motion artifact was also evaluated in ECG (Figure 5e, f). Transited from the static to dynamic status, the baseline noise detected by PTG is relatively stable than Ag/AgCl and PEDOT:PSS electrodes. sEMG is more sensitive to the vibration compared with ECG, because the detected bandwidth (20 – 2000 Hz vs. 0.05-100 Hz) is higher, and muscle contraction comes along with large skin deformation. Although pure PEDOT:PSS was measured in above tests, it is noted that without the additional graphene, pure PEDOT:PSS film is very brittle and easily break out upon strain.”

Figure 5. Electrophysiological signal detection of PTG dry electrodes. (a) A photograph of PTG conformed on a human ring finger. (b) A SEM image of cross-sectional view of PTG (top) on porcine skin (bottom). (c) AFM Young's modulus map of PTG. (d) The impedance analysis of PTG (red), pure PEDOT:PSS (blue) and Ag/AgCl (black) electrodes. (h) sEMG measured by PTG (red), pure PEDOT:PSS (blue) and Ag/AgCl (black) electrodes without and with vibration, showing lower-motion artifact detected by PTG. Inset of (h) is photo of PTG electrodes for sEMG measurements.

Figure S10. AFM height images and Young's modulus images of PTG with BSL (a, c) and without BSL (b, d).

Bending stiffness (\overline{EI}_{sum}) of our PTG electrode is calculated as following:

$$\overline{EI} = \frac{1}{12(1-\nu^2)} \overline{E}h^3$$

$$\overline{EI}_{sum} = \alpha \overline{EI}_{electrode} + (1-\alpha) \overline{EI}_{substrate}$$

Where \overline{E} represents Young's modulus, h represents thickness, ν represents Poisson ratio, and α represents the area ratio of electrode to substrate. Together with the small Young's modulus and ultrathin nature, bending stiffness of our PTG electrode (with tattoo substrate) is ~ 0.00236 pNm. Correspondingly, according to above formula, bending stiffness of skin is ~ 0.00156 pNm (Young's modulus of skin is ~ 150 kPa, and the thickness of stratum corneum is ~ 15 μm)¹. They match with each other very well.

Figure S11. SEM image of PTG (bottom) on porcine skin (top).

Figure S13. Dependence of contact impedance of Ag/AgCl gel electrodes on time.

6. The last demonstration of using the measured EMG signal to control the robotic arm doesn't really show the advantage of the improved signal-to-noise ratio obtained from the bilayer dry electrodes.

Response: We thank the reviewer for raising this concern. Following your advice, we extended PTG to the acquisition of facial sEMG and utilized the facial expression to control a robotic hand. Because face is uneven, full of wrinkles, and inevitably shape changed, it requires the facial sEMG electrodes to be highly conductive, ultra-conformal to skin as well as mechano-electrically stable. PTG indeed can accurately acquire facial sEMG signals. In contrast, gel electrodes are easily detached from the face, and the brittle PEDOT:PSS becomes completely insulative especially when the face is in motion. In addition, PTG electrodes do not affect the laser speckle contrast imaging due to the optical transparency. Thus, PTG is able to simultaneously monitor sEMG and laser speckle contrast imaging at the same location, providing more information for the diagnosis and treatment of facial nerve diseases. With the accurate monitoring of sEMG by PTG, we applied it to control the robotic hand by the multi-channel transformation of the facial muscle expression. Taken together, the high conductivity, ultra-conformability, certain mechano-electrical stability and transparency contribute PTG as an ideal electrode for the acquisition of facial sEMG without detriment to laser speckle contrast imaging, as well as being able to control artifact limbs.

A completely new Figure 6, Video S2 and corresponding description were added in the revised manuscript. Detailed revisions are listed as below.

In the revised manuscript:

“Facial sEMG acquisition and facial expression-robot interaction. EMG is an important way to evaluate facial muscle function and analyze facial muscle expression (Figure 6a). Compared to needle EMG (nEMG), sEMG is superior in painlessness, invasiveness, large facial muscle area characterization, and monitoring over time. Thus, sEMG is more practical in wearable healthcare to self-evaluate facial nerve disorder for both pre- and post-diagnosis. Considering that the superficial portions of the face muscles are uneven, full of wrinkles, and inevitably large-shape changed, facial sEMG electrodes being highly conductive, ultra-conformal as well as mechano-electrical stable are necessary. It is still a big challenge to detect facial sEMG stably and accurately both in laboratory and in clinic. To verify the capability of PTG as facial sEMG electrodes, we applied a pair of them on levator labii superioris alaeque nasi on one side of the face with Ag/AgCl as control on the other side (Figure 6d). PTG showed more stable signal quality than Ag/AgCl in facial sEMG detection (Figure 6b). This phenomenon can be attributed to the low-motion artifact electrophysiology of PTG

on skin. In contrast, Ag/AgCl gel electrodes are easily detached from the facial skin resulted from the displacement of gel, and the PEDOT:PSS is too brittle and completely insulative when the facial skin is stretched in dynamic state. With the addition of graphene underlying PEDOT:PSS, the electrical stability of PTG upon stretching is greatly enhanced (Figure 2d), facilitating the accurate sEMG signal acquisition in particular in motion of the face. Laser speckle contrast imaging technique (LSCI) is another important technique for facial nerve diseases diagnosis. It can evaluate the function of facial microcirculation, such as blood flow velocity and blood perfusion. Leveraging laser speckle contrast imaging into sEMG measurements, it will be beneficial to the deep understanding of pathological process and mechanism (Figure 6c). Commercial Ag/AgCl gel electrodes are opaque. Utilizing the relatively transparent PTG dry electrodes, simultaneous monitoring sEMG and speckle imaging at the same location can be realized, which will provide more information to the diagnosis and treatment of facial nerve palsy (Figure 6d and 6e). The blood perfusion under PTG electrodes can be measured by LSCI whereas the blood perfusion under Ag/AgCl electrodes cannot be measured by LSCI. Moreover, with the accurate monitoring of sEMG by PTG, we applied it to control the robotic hand by facial muscle expression. Three pairs of PTG were attached on zygomaticus, risorius and corrugator to control the index, middle finger, and wrist of the robotic hand respectively (Video S2). When the volunteer smiles, the robotic hand shows the gesture of victory (Figure 6f). When the volunteer extends and knits her brows, the robotic hand is able to rotate its twist correspondingly (Figure 6g). Taken together, the high conductivity, ultra-conformability, certain mechano-electrical stability and transparency contributes PTG as an ideal electrode for the acquisition of facial sEMG without detriment to laser speckle contrast imaging, as well as being able to controlling artifact limbs (Table S3).”

Figure 6. Facial sEMG acquisition and facial expression-robot interaction. (a) A photo of facial expression muscles. (b) Facial EMG measured by PTG (red) and Ag/AgCl (black) electrodes. PTG showed lower-motion artifact. (c) Schematic illustration of simultaneous monitoring sEMG and speckle imaging at the same location. (d) A photograph of PTG electrodes connected by metal for facial sEMG detection on levator labii superioris alaeque nasi on one side of the face with Ag/AgCl as control on the other side. (e) A laser speckle contrast imaging corresponding to (d). The blood perfusion under PTG electrodes can be measured by laser speckle contrast imaging technique (LSCI) whereas the blood perfusion under Ag/AgCl electrodes cannot be measured by LSCI. (f) Driving the robotic hand based on multi-channel facial sEMG signals extracted by PTG electrodes. Two PTG electrode pairs were adhered on zygomaticus and risorius to control the index and middle finger, respectively. Right photos showing that the robotic hand at the gesture of victory when the volunteer smiles. (g) PTG electrodes were adhered on corrugator to control the wrist of the robotic hand. Photos showing that the robotic hand rotates its twist when the volunteer extends and knits her brows.

Reviewer #2 (Remarks to the Author):

The manuscript by Zhao et al. reports an interesting approach to employ PEDOT:PSS as a transfer medium for CVD graphene, which produces a bilayer composite material termed as PTG. The process essentially eliminates the need to remove the sacrificial transfer medium of PMMA in the conventional technique. As a transparent electrode, the excellent optoelectronic performances of PTG comes from the synergistic charge transfer effects between graphene and PEDOT:PSS supported by Raman spectroscopy and GIWAXS. In addition, PTG also show decent mechanical deformability to function as conformal skin electrodes to acquire electrophysiological signals with low motion artifacts. I recommend the manuscript to be considered for publication after minor revisions regarding the following aspects:

We thank the reviewer for the positive comments of our work. Below we addressed the reviewer's questions point-to-point.

(1) The origin of the stretchability of PTG is less clear. I wonder whether it comes from certain intrinsic mechanisms or structural reasons (e.g. surface wrinkles formed during the transfer process). Additional discussions/experimental evidences should be provided. In addition, the reason for improved stretchability upon increasing the number of underlying graphene layers should be explained.

Response: We thank the reviewer for the comment. PTG exhibits a certain electrical durability upon tensile strain. This is mainly attributed to the sliding between graphene and the substrate, and the formation of percolation pathway with multi-layer graphene.

To understand this mechanism, we newly performed a systematic morphology study on graphene, pure PEDOT:PSS and PTGs (1G+PEDOT:PSS, 2G+PEDOT:PSS) on SEBS elastomer (~ 50 μm thick) before and after mechanical stretching at 20% strain by AFM. In the high magnification images, the granule-to-nanofibril transition was clearly observed on PTGs (Figure S5). Applied 20% strain on SEBS, cracks appeared on all the films, which are mostly perpendicular to the strain direction. However, arising from the underlying graphene, there are obviously less cracks and vaguer parallel wrinkles on PTGs than those on pure PEDOT:PSS (revised Figure 2e). It indicates that a smaller strain was transferred onto above PEDOT:PSS in PTG due to the large sliding between graphene and substrates. With the increases in graphene layer number, the sliding is strengthened accompanied with the formation of more percolation pathways. Multilayer graphene for stretchable electrodes has been reported in the literature of "Bao et al., Sci. Adv. 2017; 3: e1700159", which was also explained as the maintenance of percolation pathway and sliding over each other.

New morphological data of graphene, pure PEDOT:PSS and PTGs before and after tensile strain were added in the revised manuscript (Figure 2e) and supporting information (Figure S5) with corresponding mechanism understanding. Detailed revision is as below.

In the revised manuscript:

"The underlying graphene of PTG results in the better electrical stability upon tensile strain. To understand this mechanism, we performed a systematic morphology study on pure PEDOT:PSS and PTGs (1G+PEDOT:PSS, 2G+PEDOT:PSS) on SEBS elastomer (50 μm thick) before and after

mechanical stretching at 20% strain by AFM. In the high magnification images, the granule-to-nanofibril transition was clearly observed on PTGs (Figure S5). Applied 20% strain on SEBS, cracks appeared on all the films, which are mostly perpendicular to the strain direction. However, arising from the underlying graphene, there are obviously less cracks and vaguer parallel wrinkles on PTGs than those on pure PEDOT:PSS film (Figure 2e). It indicates that a smaller tensile strain was transferred onto PTG due to the large sliding between graphene and elastomer. With the increase of graphene layer number, this factor is strengthened accompanied with the formation of more percolation pathways, leading to better electrical durability at strain. This is consistent with the report for multilayer graphene as stretchable electrodes^{40,41}. Thus, to compromise the transparency, conductivity and mechano-electrical stability of PTG, our optimal condition for following epidermal electrophysiology is bi-layer graphene transferred by PEDOT:PSS at both SDS and BSL concentrations of 1.0 wt%.”

Figure 2. (e) Morphological comparison of pure PEDOT:PSS and PTGs (mono- and bi-layer graphene) upon 20% strain by AFM. The strain direction is perpendicular to the cracks and parallel to the wrinkles which are generated by Poisson effect.

Figure S5. Morphological comparison of pure PEDOT:PSS and PTGs (mono- and bi-layer graphene) by AFM.

(2) The authors claim the addition of bis(trifluoromethane) sulfonimide lithium salt into PEDOT:PSS improves the electrical conductivity and the mechanical stretchability. Although it is intuitive that ionic compound may boost the conductivity of PEDOT:PSS, the benefit of BSL salt on mechanical deformability requires additional clarifications and explanations.

Response: We thank the reviewer for the valuable suggestion. To demonstrate the benefit of BSL additives on mechanical deformability of PTG, we compared the Young’s modulus maps of PTGs without and with BSL (1.0 wt%) under AFM. The average Young’s modulus of PTG with BSL is ~ 640 kPa, whereas that of PTG without BSL is ~ 2.1 MPa (new Figure 5c, Figure S10). Also, with the addition of BSL, the crystallinity of PEDOT is getting better from the height images (Figure 4b, Figure S10) and GIWAXS data (Figure 4a). These indicate that BSL additives soften the PSS

domain, leading to the decreased Young's modulus and increased mechanical deformability of PTG.

New Young's modulus maps and AFM height images of PTGs with and without BSL additives (1.0 wt%) were added in the revised manuscript (Figure 5c) and supporting information (Figure S10) with corresponding description. Detailed revision is as below.

In the revised manuscript:

“Interfacial impedance is a critical parameter in epidermal electrophysiological measurement. Without additional gel, dry electrodes would reduce the contact impedance by forming conformal interface with skin. According to the Young's modulus map under AFM, average Young's modulus of PTG is about 640 kPa (Figure 5c, Figure S10), which matches the modulus of human stratum corneum. Transferred onto a human ring finger full of wrinkles, PTG exhibited intimate contact on it (Figure 5a). The conformability is also confirmed by SEM imaging of PTG on porcine skin (Figure 5b, Figure S11). The ultrathin PTG with low Young's modulus leads to conformal contact with wrinkled skin^{45,46}.”

Figure 5 (c) AFM Young's modulus map of PTG. They showed PTG is conformal with skin due to the small thickness and low Young's modulus.

Figure S10. Young's modulus maps and AFM height images of PTGs with (a, c) and without BSL (b, d) additives (1.0 wt%).

Bending stiffness (\overline{EI}_{sum}) of our PTG electrode is calculated as following:

$$\overline{EI} = \frac{1}{12(1-\nu^2)} \overline{E}h^3$$

$$\overline{EI}_{sum} = \alpha \overline{EI}_{electrode} + (1-\alpha) \overline{EI}_{substrate}$$

Where \overline{E} represents Young's modulus, h represents thickness, ν represents Poisson ratio, and α represents the area ratio of electrode to substrate. Together with the small Young's modulus and ultrathin nature, bending stiffness of our PTG electrode (with tattoo substrate) is ~ 0.00236 pNm. Correspondingly, according to above formular, bending stiffness of skin is ~ 0.00156 pNm. (Young's modulus of skin is ~ 150 kPa, and the thickness of stratum corneum is ~ 15 μm)¹. They match with each other very well.

(3) In order to put the optoelectronic performances of PTG into the research context, a table summarizing the state-of-the-art of graphene transparent electrodes should be provided.

Response: We thank the reviewer for the suggestion. A comparison table of the state-of-the-art of graphene-based transparent electrodes were summarized in Table S1 as following. From this table, it is observed that pure graphene structures have relatively high resistance and high transparency. With additional conductive species, the resistance can be lowered down at a compromising transparency. Graphene/AgNWs hybrid film exhibits the best combination of conductivity and transparency. However, AgNWs are not air-stable due to oxidation, which are not suitable in epidermal electrophysiology for long-time use. Because PEDOT:PSS is solution-processible, conductive, and able to form a thin film on graphene with the help of additives, we employed it as a medium to transfer graphene directly. Their synergistic enhancement in conductivity and mechano-electrical stability leads PTG as an ideal ultra-conformal skin-electrode for low-motion artifact electrophysiology.

Table S1. Comparison of the optoelectronic properties of the state-of-the-art of graphene-based transparent electrodes.

Methods	R _{sh} (Ω/sq)	Transparency	References
Solution-Processed rGO	$10^2 - 10^3$	80%	ACS Nano 2 , 463-470 (2008)
CVD graphene on Cu	2100 (1 layer) 350 (4 layers)	98% 90%	Nano Lett. 9 , 4359-4363 (2009)
CVD graphene on Ni	280	80%	Nature 457 , 706-710 (2009)
Graphene/AgNWs hybrid film	8	94%	Nano Lett. 15 , 4206-4213 (2015)
MGG	185	88%	Sci. Adv. 3 e1700159 (2017)
GEF /monolayer graphene	150	83%	ACS Materials Lett. 2 , 999-1007 (2020)
PTG	45	80%	This work

REVIEWER COMMENTS

Reviewer #1 (Remarks to the Author):

In the revision, the authors added more experiments and discussions to address the questions from the reviewer. Regarding the material properties, the comparison with the previous works is much clearer and more convincing; the understandings about the mechanistic behaviors are also deepened. However, my questions about the practical merits of the PTG electrode for working as electrophysiology dry electrodes remain largely unaddressed. Although the overall engineering target is to replace the gel electrodes as the current approach for electrophysiology measurement, since dry electrodes based on engineered PEDOT:PSS has already been reported in several works before, such as "Fully organic compliant dry electrodes self-adhesive to skin for long-term motion-robust epidermal biopotential monitoring. Nat. Commun. 2020", the comparisons in these demonstrations in Fig. 6 should be made with other types of PEDOT:PSS dry electrodes, rather than the gel electrode.

Moreover, regarding my previous question about the stable adhesion of the PTG electrode to the skin surface, the response addressed it mainly from the aspect of achieving good conformability, which is a different aspect than the physical/chemical bonding at such an interface to prevent the delamination or detachment caused by motion or skin deformations. Even for the conformability, the experimental evidence shown by the authors is just based on a freestanding PTG film with a very low thickness, which is different than its use in real electrophysiological measurements, in which deposited Ag patterns and a SEBS or paper substrate is needed, which will largely the actual thickness of the entire sheet. With the Ag on top, the stretchability of the electrode could also get much worse.

Reviewer #2 (Remarks to the Author):

The authors have carefully revised the manuscript according to my suggestions/comments. I recommend the current version of the manuscript to be considered for publication. I also encourage the authors to further polish the manuscript for clear delivery of the key ideas.

Reviewer #1 (Remarks to the Author):

In the revision, the authors added more experiments and discussions to address the questions from the reviewer. Regarding the material properties, the comparison with the previous works is much clearer and more convincing; the understandings about the mechanistic behaviors are also deepened. However, my questions about the practical merits of the PTG electrode for working as electrophysiology dry electrodes remain largely unaddressed. Although the overall engineering target is to replace the gel electrodes as the current approach for electrophysiology measurement, since dry electrodes based on engineered PEDOT:PSS has already been reported in several works before, such as "Fully organic compliant dry electrodes self-adhesive to skin for long-term motion-robust epidermal biopotential monitoring. Nat. Commun. 2020", the comparisons in these demonstrations in Fig. 6 should be made with other types of PEDOT:PSS dry electrodes, rather than the gel electrode.

Response: We thank the reviewer for the positive comments on our revised manuscript.

Because of the low sheet resistance (24 Ω /sq, 4142 S/cm), high transparency, mechano-electrical stability, and ultra-thin skin-conformal nature, we believe our PTG has the following practical merits as electrophysiological electrodes: 1) uniqueness in facial nerve diseases; 2) higher signal quality and lower motion artifact in sEMG detection; 3) long-term monitoring of the very weak EEG (electroencephalogram) signals. To compare with other PEDOT:PSS electrodes, in particular the PWS (blend of PEDOT:PSS, waterborne polyurethane and D-sorbitol) recently published in "Fully organic compliant dry electrodes self-adhesive to skin for long-term motion-robust epidermal biopotential monitoring. Nat. Commun. 2020, 11, 4683", we elaborate above merits separately as below and summarized them in Table S1.

1. Uniqueness in facial nerve diseases:

PTG is able to accurately acquire facial sEMG without detriment to laser speckle contrast imaging, as well as to control artificial limbs (Figure 6) due to its high conductivity, transparency, mechano-electrical stability, and ultra-thin skin-conformal nature. Simultaneously achieving sEMG and doing laser speckle contrast imaging will provide more accurate information for the deep understanding of pathological process and mechanism of facial nerve diseases, which will be beneficial for its diagnosis and treatment. In contrast, pure PEDOT:PSS without underlying graphene is easily fractured, become insulative, and cannot detect sEMG when facial skin is continuously in motion (Figure S24). In terms of other engineered PEDOT:PSS dry electrodes, such as PWS, the authors did not apply it on the largely deformed facial skin surface. However, deduced from its signal quality in sEMG on forearm, PWS is very likely not to be able to detect facial sEMG accurately due to its low conductivity (545 S/cm vs 4142 S/cm) and large thickness (bulk vs nm scale). In addition, PWS electrode is not compatible with laser speckle contrast imaging, as it is opaque and will completely block the laser.

2. Higher signal quality and lower motion artifact in sEMG acquisition:

To compare the sEMG signal quality, we performed the same sEMG detection on PTG with that on PWS. By holding the grip dynamometer at 3 kPa, 4 kPa and 5 kPa, the sEMG intensity on forearm gradually increased with larger gripping force, exhibiting an increasing trend of signal-to-noise ratio ($SNR_{dB} = 20\lg\left(\frac{A_{signal}}{A_{noise}}\right)$). Although our test forces are smaller than those on PWS (0.21, 0.27, 0.33GPa), the SNRs are much larger (19, 23, 26 dB vs 13, 16, 19 dB), indicating the better signal

quality of PTG over PWS (Figure R1). When finger is in flexion/extension, weak sEMG is generated. Again, PTG showed larger SNRs than PWS on the sEMG acquisition at each finger (from thumb to little finger: 21, 19, 21, 25, 20 dB vs 11, 13, 17, 11, 14 dB). We attribute the better signal quality of PTG over PWS to the higher conductivity and skin-conformability.

Figure R1. Comparison of signal quality of sEMG obtained by PTG and PWS electrodes. (a, b) SNRs of sEMG generated at different gripping force measured by PTG (a) and PWS (b) electrodes. (c, d) SNRs of sEMG generated by each finger acquired by PTG (c) and PWS (d) electrodes.

3. Long-term monitoring of the very weak EEG signals

One biggest motivation for developing dry electrodes is to reduce motion artifact coming from the displacement of gel on skin when long-term monitoring of electrophysiology. Due to the very weak signal strength, long-term monitoring of EEG is extremely challenging, in particular for the status of exercising. Normally, for EEG monitoring, the volunteer sits in a comfortable position peacefully and experiences brain stimulus, such as doing math quiz, watching videos or listening to music. As PTG dry electrodes are ultra-thin and the attachment is comfortable, we therefore applied PTG on a volunteer for long-term monitoring of EEG without taking off the electrode pairs. During the monitoring of 12 hours, the volunteer slept, exercised and recovered to calm with two PTG electrodes mounted in the frontal region (Fp1 and Fp2) of the brain and one reference electrode behind the ear. The EEG signal quality remains decent, despite that the volunteer is soaked in sweat after exercising. Figure 7a-c showed three typical EEG signals corresponding to the three statuses of brain. After Fast Fourier Transformation (FFT) treatment (Figure 7d-f), the brain waves can be divided into several parts: delta wave (0–4 Hz), theta wave (4–8 Hz), alpha wave (8–13 Hz), beta wave (13–30 Hz), and gamma wave (40 Hz or higher). To differentiate various statuses of the brain, alpha and beta waves were extracted and compared separately (Figure 7g-l). The intensity of alpha and beta waves at exercising is obviously stronger than that at sleeping and calm status. This is in accord with the cognition that exercise improves our concentration. The stable recording of EEG signals can be explained by the intimate contact of ultra-thin PTG on skin despite the extension/contraction and sweat secretion of skin. This can be verified by the stable contact impedance at 10 Hz as a function of time (Figure S25). The capability of long-term monitoring of EEG suggests that our PTG is an ideal skin electrode candidate, facilitating the future healthcare and human-machine interfacing based on accurate EEG acquisition.

Regarding the practical merits and advantages of PTG as electrophysiological electrodes, we discussed the fractured feature of PEDOT:PSS only (new Figure S24), compared sEMG detection between PTG and PWS (new Figure 5i and j, Figure S24, Table S1), and added the long-time monitoring of EEG (new Figure 7). Detailed revisions in the revised manuscript and supporting information are listed as below.

In the revised manuscript:

In the section of “Facial sEMG acquisition and facial expression-robot interaction”:

“..., and the PEDOT:PSS is too brittle and completely insulative when the facial skin is stretched in dynamic state (Figure S24).”

Figure S24. Facial sEMG measured by PEDOT:PSS only. With time going on, the movement of face is unavoidable (e.g.: smile, eating, talking...), leading PEDOT:PSS to be failed in sEMG measurement. Inset is a photo of the PEDOT:PSS dry electrode on face. After a short facial sEMG measurement, the PEDOT:PSS electrode is fractured and become insulative.

Figure 5. Electrophysiological signal detection of PTG dry electrodes. (a) A photograph of PTG conformed on a human ring finger. (b) A SEM image of cross-sectional view of PTG (top) on porcine skin (bottom). (c) AFM Young's modulus map of PTG. (d) The impedance analysis of PTG (red), pure PEDOT:PSS (blue) and Ag/AgCl (black) electrodes. (e) EOG measured by PTG (red), pure PEDOT:PSS (blue) and Ag/AgCl (black) electrodes, showing peaks and valleys corresponding to the eyelid movement. (f) sEMG baseline noise comparison of the three types of electrodes in static and dynamic status. (g) ECG measured by PTG (red), pure PEDOT:PSS (blue) and Ag/AgCl (black) electrodes without and with vibration. Characteristic P, Q, R, S and T waves are clearly identified. (h) sEMG measured by PTG (red), pure PEDOT:PSS (blue) and Ag/AgCl (black) electrodes without and with vibration, showing lower-motion artifact detected by PTG. Insets of (e, g, h) are photos of PTG electrodes for EOG, ECG, sEMG measurements. (i) sEMG measured by PTG at different gripping force, showing that sEMG intensity becomes higher at larger gripping force. (j) sEMG acquired by PTG from the finger (thumb, index, middle, ring and little fingers) motion. (k) Driving the robotic hand to show various gestures (victory, Arabic numerals of six and three, and holding a centrifuge tube) based on multi-channel sEMG signals extracted by PTG electrodes.

Five PTG electrode pairs were adhered on designated locations for acquiring sEMG signals of thumb, index, middle, ring and little fingers, and remotely driving the movement of robotic hand.

In the section of “Low-motion artifact electrophysiological monitoring”:

“Five electrode pairs were adhered on designated locations to respectively detect sEMG signals of thumb, index, middle, ring and little finger of the volunteer (Figure 5j), showing high detected quality with SNRs around 20 dB (Figure S23).”

Figure S23. The SNRs of sEMG on forearm generated with different gripping force (a) and the SNRs of sEMG around the wrist generated by each finger.

Table S1. Comparison of the practical merits of various PEDOT:PSS electrodes for electrophysiology.

Methods	Material Properties	Signal Detection	ECG	sEMG (facial sEMG)	EEG	Human-machine interfacing	Ref.
Inject-printed PEDOT:PSS	520 Ω /sq (150 nm thick, transparent), conductive at 5% strain	ECG, sEMG	Similar SNR and noise voltage vs. Ag/AgCl in static status	Similar SNR and noise voltage vs. Ag/AgCl in static status	No EEG data	No	Adv. Sci. 5 , 1700771 (2018)
PWS	545 S/cm (20 μ m thick, opaque), stably conductive at 30% strain	ECG, sEMG, EEG	Lower noise voltage vs. Ag/AgCl both in static and dynamic status	No comparison with other electrodes (no facial sEMG data)	EEG at the stimuli of sound	One-channel	Nat. Commun. 11 , 4683 (2020)
PTG	4142 S/cm (25 Ω /sq, 100 nm thick, transparent), conductive at 40% strain	ECG, EOG, sEMG at largely deformed position, EEG	Higher SNR and lower noise voltage vs. Ag/AgCl both in static and dynamic status	Higher SNR and lower noise voltage vs. Ag/AgCl both in static and dynamic status	EEG monitoring for ~12 hours	Multi-channel	This work

“**Long-time monitoring of EEG signals.** Electroencephalogram (EEG) recording is to measure the electrical activity of the brain, providing helpful information for the diagnosis or treatment of the brain disorders. Due to the very weak signal strength (μ V scale), long-time monitoring of EEG is

extremely challenging, in particular for the status of exercising. Normally, for EEG monitoring, the volunteer sits in a comfortable position peacefully and experiences brain stimulus, such as doing math quiz, watching videos or listening to music. As PTG is ultra-thin and the attachment is comfortable, we therefore applied PTG on a volunteer for long-time monitoring of EEG without taking off the electrode pairs. During the monitoring of 12 hours, the volunteer slept, exercised and recovered to calm with two PTG electrodes mounted in the frontal region (Fp1 and Fp2) of the brain and one reference electrode behind the ear⁴⁷. The EEG signal quality is decent, despite that the volunteer is at various statuses and even soaked in sweat after exercising. Figure 7a-c showed three typical EEG signals corresponding to the three statuses of brain. After Fast Fourier Transformation (FFT) treatment (Figure 7d-f), the brain waves can be divided into five parts: delta wave (0–4 Hz), theta wave (4–8 Hz), alpha wave (8–13 Hz), beta wave (13–30 Hz), and gamma wave (40 Hz or higher)^{47,48}. To differentiate various statuses of the brain, alpha and beta waves were extracted and compared separately (Figure 7g-l). The intensity of alpha and beta waves at exercising is obviously stronger than that at sleeping and calm status. This is in accord with the cognition that exercise improves our concentration. The stable recording of EEG signals can be explained by the intimate contact of ultra-thin PTG on skin regardless the extension/contraction and sweat secretion of skin. This can be verified by the stable contact impedance at 10 Hz as a function of time (Figure S25). The capability of long-time monitoring of EEG suggests that PTG is an ideal skin-electrode candidate, facilitating the future healthcare and human-machine interfacing based on accurate EEG acquisition.”

Figure 7. Long-time monitoring of EEG. (a-c) Original EEG signals, (d-f) Fast Fourier Transformed

EEG signals, (g-i) the extracted alpha waves and (j-l) beta waves when the volunteer was at status of sleeping, exercising (skipping rope) and recovering to calm state after ~ 1 hour.

Moreover, regarding my previous question about the stable adhesion of the PTG electrode to the skin surface, the response addressed it mainly from the aspect of achieving good conformability, which is a different aspect than the physical/chemical bonding at such an interface to prevent the delamination or detachment caused by motion or skin deformations. Even for the conformability, the experimental evidence shown by the authors is just based on a freestanding PTG film with a very low thickness, which is different than its use in real electrophysiological measurements, in which deposited Ag patterns and a SEBS or paper substrate is needed, which will largely the actual thickness of the entire sheet. With the Ag on top, the stretchability of the electrode could also get much worse.

Response: We thank the reviewer for the comment.

The electrophysiological signal sensor is composed of two parts: the electrodes for sensing electrophysiological signal and the connection between electrodes and the measurement kit. PTG works as the sensing electrode, and the connection between soft electrodes and the hard measurement kit is achieved by the deposited metal film referred to literatures (*Adv. Funct. Mater.* **28**, 1803279 (2018), *Adv. Sci.* **5**, 1700771 (2018)). The adhesive force between ultra-thin PTG and skin is simply Van der Waals force (*Science*. **333**, 838-843 (2011)). Due to the ultra-thin nature, PTG and skin forms good conformability, leading to stable attachment of PTG electrode to the skin surface. In our last response we did morphological observation and mechanical force measurement by AFM to illustrate the good conformability of PTG to skin. Herein, we additionally monitored the contact impedance of PTG electrode on skin as a function of time in real measurement (PTG on tattoo paper with Ag as connection) to verify the stable attachment, and explained it from the matched bending stiffness between PTG electrode and skin as following:

During long-time monitoring of EEG, the contact impedance of PTG electrodes on skin were tested. The volunteer wore PTG electrodes (PTG on tattoo paper with Ag as connection) in frontal region of her brain for 12 hours. During this period, she slept, skipped rope for 0.5 hour, and rest. The extension/contraction and sweat secretion of the skin did not affect the contact impedance obviously. 10 Hz is the center frequency for alpha wave in EEG. From the changing trend of contact impedance at 10 Hz vs time (Figure S25), it is confirmed that the attachment between PTG electrodes and skin are stable.

To interpret the stable attachment of PTG electrodes on skin during electrophysiological measurement, we introduced the study of literature (*Adv. Funct. Mater.* **28**, 1803279 (2018), *Nano Lett.* **19**, 1577–1586 (2019)) in ultra-thin electrodes by the concept of bending stiffness. Bending stiffness refers to the amount of stiffness that the subject will deflect. The smaller the stiffness, the larger the deflection will occur. According to the formula (1) and (2), bending stiffness (\overline{EI}_{sum}) of our PTG electrode can be calculated as following:

$$\overline{EI} = \frac{1}{12(1-\nu^2)} \overline{E}h^3 \quad (1)$$

$$\overline{EI}_{sum} = \alpha \overline{EI}_{electrode} + (1 - \alpha) \overline{EI}_{substrate} \quad (2)$$

Where \overline{E} represents Young's modulus, h represents thickness, ν represents Poisson ratio, and α

represents the area ratio of electrode to substrate. In this work, the thickness of PTG is about 100 nm. And in real electrophysiological measurements, we transfer PTG on decal transfer tattoo paper, and the thickness of ethylcellulose layer in decal transfer tattoo paper is about 450 nm. Together with the small Young's modulus and ultra-thin nature, bending stiffness of our PTG electrode (with tattoo substrate) is ~ 0.00236 pNm. Correspondingly, according to above formula, bending stiffness of skin is ~ 0.00156 pNm. (Young's modulus of skin is ~ 150 kPa, and the thickness of stratum corneum is ~ 15 μm). They match with each other very well. The matched bending stiffness leads to the ultra-conformability between PTG electrodes and skin. Once two subjects form ultra-conformability with each other, they tend to attach intimately just like contact lens with curved eyes (*Nat. Commun.* **9**, 2334 (2018)). The physical/chemical interaction in between the interface is just Van der Waals force. Besides, the existence of sweat may introduce capillary force, which can enhance the intimate contact and stable attachment (*ACS Nano* **15**, 2800-2811 (2021)).

Considering the PTG film is too thin, how to achieve robust connection between ultra-thin electrodes and external electrophysiological kits is difficult. Referring to literature (*Adv. Funct. Mater.* **28**, 1803279 (2018), *Adv. Sci.* **5**, 1700771 (2018), *ACS Nano.* **11**, 7634-7641 (2017)), we additionally deposit 50 nm-thick Ag patterns on tattoo to contact with PTG film. The center region for sensing the electrophysiological signals is still the PTG film, and the Ag patterns are just for connection to the measurement kits. Within the range of skin deformation, the mechano-electrical stability for Ag or Au is sufficient (*Appl. Phys. Lett.* **94**, 071902 (2009)).

Regarding the stable adhesion of PTG electrodes on skin, we monitored its contact impedance as a function of time (Figure S25), and strengthened the explanation from the matched bending stiffness (both revised manuscript and Supporting Information). Detailed revisions are listed as below.

In the revised manuscript:

“The stable recording of EEG signals can be explained by the intimate contact of ultra-thin PTG on skin regardless the extension/contraction and sweat secretion of skin (See discussion in supporting information). This can be verified by the stable contact impedance at 10 Hz as a function of time (Figure S25).”

Figure S25. (a) Typical contact impedance of PTG on skin after 12 hours. (b) Contact impedance at 10 Hz as a function of time.

In the revised manuscript:

“According to the Young's modulus map under AFM, average Young's modulus of PTG is about 640 kPa (Figure 5c, Figure S10), which matches the modulus of human stratum corneum (~ 150

kPa). Bending stiffness (\overline{EI}) refers to the amount of stiffness that the subject will deflect:

$$\overline{EI} = \frac{1}{12(1-\nu^2)} \overline{E}h^3$$

where \overline{E} represents Young's modulus, h represents thickness, ν represents Poisson ratio. The ultra-thin nature and low Young's modulus of PTG leads to a similar bending stiffness with that of skin, contributing to conformal contact with wrinkled skin (see discussion in Supporting Information)^{45,46}."

In supporting information:

"To interpret the stable attachment of PTG electrodes on skin during electrophysiological measurement, we introduced the concept of bending stiffness. Bending stiffness refers to the amount of stiffness that the subject will deflect. The smaller the stiffness, the larger the deflection will occur. According to the formula (1) and (2), bending stiffness (\overline{EI}_{sum}) of our PTG electrode can be calculated as following:

$$\overline{EI} = \frac{1}{12(1-\nu^2)} \overline{E}h^3 \quad (1)$$

$$\overline{EI}_{sum} = \alpha \overline{EI}_{electrode} + (1-\alpha) \overline{EI}_{substrate} \quad (2)$$

Where \overline{E} represents Young's modulus, h represents thickness, ν represents Poisson ratio, and α represents the area ratio of electrode to substrate. In this work, the thickness of PTG is about 100 nm. And in real electrophysiological measurements, we transfer PTG on decal transfer tattoo paper, and the thickness of ethylcellulose layer in decal transfer tattoo paper is about 450 nm. Together with the small Young's modulus and ultra-thin nature, bending stiffness of our PTG electrode (with tattoo substrate) is ~ 0.00236 pNm. Correspondingly, according to above formula, bending stiffness of skin is ~ 0.00156 pNm. (Young's modulus of skin is ~150 kPa, and the thickness of stratum corneum is ~ 15 μ m)¹. They match with each other very well. The matched bending stiffness leads to the ultra-conformability between PTG electrodes and skin. Once two subjects form ultra-conformability with each other, they tend to attach intimately. Besides, the existence of sweat may introduce capillary force, which can enhance the intimate contact and stable attachment².

REVIEWERS' COMMENTS

Reviewer #1 (Remarks to the Author):

This paper now addressed all the comments and can be recommended for acceptance.

REVIEWER COMMENTS

Reviewer #1 (Remarks to the Author):

The main innovation in this work is the use of CVD-grown graphene as the substrate for the spin-coating of PEDOT:PSS so as to enhance the morphological ordering and packing of PEDOT:PSS, and therefore its conductance. A set of systematic characterizations have been carried out to understand the changes of the PEDOT:PSS's molecular and packing ordering. Also, the transferred graphene and PEDOT:PSS bilayer exhibits stretchability up to 40% strain, which is another interesting point. The overall application direction proposed and demonstrated for this material design is for the on-body measurement of electrophysiological signals, to achieve a higher signal-to-noise ratio. Although the presented material design in this work carries some new phenomenon and achieved some interesting results, the technological advancement in reference to the state-of-the-art alternatives has been convincingly illustrated. Moreover, some results also deserve deeper understanding. Here are the comments:

We thank the reviewer for the positive comments of our work and welcome the opportunity to clarify the questions. Below we address the reviewer's comments point-by-point.

1. There has already been a number of different approaches established for enhancing the conductivity of PEDOT:PSS. Compared to these approaches, it seems like the achieved conductance is on the similar level. For example, in the work of "Wang et al., Sci. Adv. 2017;3: e1602076", the sheet resistance of the PEDOT:PSS with some ionic liquid additives reached the lowest value of 59 ohms/sq, which is not much higher than the lowest value of 45 ohms/sq in this work. As such, to illustrate the technological value of this work, the authors need to make a comprehensive comparison in all the major aspects to the reported approaches for enhancing the conductivity of PEDOT:PSS. For doing so, the sheet resistances reported in this paper needs to be converted to conductivity in the unit of S/cm.

Response: We thank the reviewer for raising the concern. In this manuscript, we aim at preparing ultra-conformal skin electrodes for achieving high-quality electrophysiological signal detection. CVD graphene is the thinnest electrode material, and PEDOT:PSS with SDS and ionic liquid is able to form a thin and continuous film (~ 100 nm) on it through spin-coating. These two conductive materials form a synergistic film, named PTG (~ 100 nm). When evaluating the conductivity of a thin film, the unit of Ω/sq is often used. As an alternative, the unit of S/cm is to express as bulk conductivity. To convert the sheet resistance (R_{sh} in Ω/sq) to conductivity (S/cm), the thickness of films (t) was carefully measured, and the conductivity can be equal to $\frac{1}{R_{\text{sh}} \times t}$. Following your suggestion, we compared our PTG with most approaches for enhancing the conductivity of PEDOT:PSS reported in recent 5 years in terms of film thickness (nm), conductivity (S/cm), and sheet resistance (Ω/sq) (new Table 1). Considering both conductivity and transparency, our PTG is superior to PEDOT:PSS as a transparent and conductive electrode particularly ultra-conformal electrode.

On the other hand, referring to the work of "Wang et al., Sci. Adv. 2017;3: e1602076", in which the

ionic liquid was as high as 50 wt%, we further increased the concentration of ionic liquid. With the addition of a much lower concentration of ionic liquid (5.0 wt%), the conductivity can reach $\sim 24 \Omega/\text{sq}$ (4142 S/cm), which is comparable to 4100 S/cm of the referred work. Such high conductivity confirms not only the enhancement effect of ionic liquid, but also the synergistic effect of graphene. We could hypothesize that in PTG system, at an equal concentration of ionic liquid to the referred work, the conductivity could be extremely high. This has been preliminary verified by $\sim 15 \Omega/\text{sq}$ for PTG at 7.0 wt% ionic liquid. However, higher concentration of ionic liquid results in non-uniform PTG film due to gelation of PEDOT:PSS solution. To balance the transparency and achieve ultrathin electrode, our optimal PTG condition is with the addition of 1.0 wt% ionic liquid.

The new experimental data for PTG at 5.0 and 7.0 wt% ionic liquid, the new comparison table of figure of merits of PEDOT:PSS, graphene and PTG films (Figure 2b, Table 1, Table S2), and corresponding discussion has been added in the revised manuscript. Detailed revision is as below. In the revised manuscript:

“At 5.0 wt% BSL concentration, the conductivity of PTG can reach $\sim 4142 \text{ S/cm}$, which exceeds the record high value (4100 S/cm) for PEDOT:PSS only film (Table S2). However, the gelation of PEDOT:PSS solution at high BSL concentration makes obstacles for the formation of uniform PEDOT:PSS carrier layers by spin-coating, leading to a thick PTG film. We also compared our PTG with most approaches recently reported for enhancing the conductivity of PEDOT:PSS in terms of film thickness (nm), conductivity (S/cm), and sheet resistance (Ω/sq) (Table 1). Considering both conductivity and sheet resistance, our PTG is superior to PEDOT:PSS as transparent and conductive electrodes, particularly as ultra-conformal electrodes.”

Figure 2. Optoelectronic performances of PTG. (b) Sheet resistance and transmittance (at $\lambda = 550 \text{ nm}$) of PTG films at various BSL (1.0 wt% SDS) concentrations when forming PEDOT:PSS carrier layer.

Table 1. Sheet resistance (R_{sh}), Thickness and Conductivity of different PEDOT:PSS film.

Methods	R_{sh} (Ω/sq)	Thickness (nm)	Conductivity (S/cm)	Application	Reference
Ionic liquid additive	59	-	3100	Stretchable circuit	Sci. Adv. 3 , e1602076 (2017)
H ₂ SO ₄ soaking	46	-	4000	Solar cell	Adv. Mater. 27 , 2317–2323 (2015)
HNO ₃ soaking	-	27	4100	Solar cell	Adv. Electron. Mater. 1 , 1500121 (2015)
Zn(TFSI) ₂ doping	-	-	4115	Solar cell	Joule 3 , 2205–2218 (2019)
MSA soaking	50	79	2540	Solar cell	ACS Appl. Mater. Interfaces 7 , 15314–15320 (2015)
Methanol/MSA soaking	43	65	3560	Solar cell	ACS Appl. Mater. Interfaces 7 , 16287–16295 (2015)
PTG	45	80	2850	Electrophysiology detection	This work
	24	100	4142		

Table S2. Comparison of the thickness and electrical conductivity of PTG and pure PEDOT:PSS films at various preparation conditions.

Sample	Preparation condition	R_{sh} (Ω/sq)	Thickness (nm)	Conductivity (S/cm)
Pure PEDOT:PSS	SDS 1.0 wt%+BSL 1.0 wt%, 4000 rpm	70.1±0.2	83.5±2.2	1620
Pure PEDOT:PSS	SDS 1.0 wt%+BSL 1.0 wt%, 3000 rpm	80.2±0.4	185.0±4.1	674
PTG (1G+PEDOT:PSS)	SDS 1.0 wt%+BSL 1.0 wt%, 3000 rpm	43.8±0.9	79.7±3.7	2850
PTG (1G+PEDOT:PSS)	SDS 1.0 wt%+BSL 5.0 wt%, 3000 rpm	24.1±0.3	100±4.8	4142
PTG (1G+PEDOT:PSS)	SDS 1.0 wt%+BSL 7.0 wt%, 3000 rpm	14.6±0.7	-	-
Graphene		600±2	~1	

2. For the claim high transconductance as an advantage from this bilayer film, it seems like this is not correlated to the interactions between PEDOT:PSS and graphene. So if it only comes from the reduced thickness of PEDOT:PSS, then there is not much uniqueness here in achieving the high transconductance.

Response: We thank the reviewer for the comments. We first compared the electrical conductivity of pure PEDOT:PSS and PTG films at the same thickness and concentration of additives (SDS+BSL). At both thicknesses of ~ 80 nm, the conductivity of PTG (~ 2850 S/cm) is ~ 2 times higher of PEDOT:PSS (1620 S/cm, new Table S2), indicating that the interaction between PEDOT:PSS and graphene plays an important role in increasing the conductivity of PEDOT:PSS.

We also noted that at the same spin-coating condition with same concentration of additives (SDS+BSL) in PEDOT:PSS solution, thickness of pure PEDOT:PSS (~ 185 nm) film is ~ 2.3 times thicker of PTG (~ 80 nm) film. The interference of wetting can be excluded, because contact angles of PEDOT:PSS solution on graphene and SiO₂/Si substrates are 74.6° and 74.5° respectively. The obviously decreased thickness of PTG is possibly due to the strong π - π interaction between PEDOT:PSS and graphene, leading to reordering of PEDOT on graphene. The reduced thickness

resulted from reordering of PEDOT was also observed in the literature (*ACS Appl. Mater. Interfaces* **10**, 29115-29126 (2018)).

Above new experimental data at various thicknesses and spin-coating condition were compared in a table (Table S2), and corresponding discussion has been added in the revised manuscript. Detailed revision is as below.

In the revised manuscript:

“It is noteworthy that at the same spin-coating condition with same concentration of additives (SDS+BSL) in PEDOT:PSS solution, PEDOT:PSS only (~ 185 nm) film is ~ 2.3 times thickness of PTG (~ 80 nm, 1G) film (Table S2). Because contact angles of PEDOT:PSS solution on graphene and SiO₂/Si substrates are similar (74.6° and 74.5°, respectively, Figure S6), the obviously decreased thickness of PTG is highly likely due to the strong π - π interaction between PEDOT:PSS and graphene, leading to reordering of PEDOT on graphene. On the other hand, at the same thickness (~ 80 nm) of PTG and pure PEDOT:PSS films, the conductivity of PTG (~ 2850 S/cm, 1G) is ~ 2 times high of PEDOT:PSS (1620 S/cm). These indicate that the synergistic enhancement between PEDOT: PSS and graphene is the main contributor to the high electrical conductivity of PTG. Detailed preparation condition and electrical property of the PTG and pure PEDOT:PSS films were summarized in Table S2.”

Table S2. Comparison of the thickness and electrical conductivity of PTG and pure PEDOT:PSS films at various preparation conditions.

Sample	Preparation condition	R _{sh} (Ω /sq)	Thickness (nm)	Conductivity (S/cm)
Pure PEDOT:PSS	SDS 1.0 wt%+BSL 1.0 wt%, 4000 rpm	70.1±0.2	83.5±2.2	1620
Pure PEDOT:PSS	SDS 1.0 wt%+BSL 1.0 wt%, 3000 rpm	80.2±0.4	185.0±4.1	674
PTG (1G+PEDOT:PSS)	SDS 1.0 wt%+BSL 1.0 wt%, 3000 rpm	43.8±0.9	79.7±3.7	2850
PTG (1G+PEDOT:PSS)	SDS 1.0 wt%+BSL 5.0 wt%, 3000 rpm	24.1±0.3	100±4.8	4142
PTG (1G+PEDOT:PSS)	SDS 1.0 wt%+BSL 7.0 wt%, 3000 rpm	14.6±0.7	-	-
Graphene		600±2	~1	

3. What is the mechanism underlying the stretchability of this bilayer film? Why the multi-layer graphene can lead to even better stretchability?

Response: We thank the reviewer for raising this concern. The underlying graphene of PTG can lead to better electrical stability upon tensile strain. To understand this mechanism, we newly performed a systematic morphology study on graphene, pure PEDOT:PSS and PTGs (1G+PEDOT:PSS, 2G+PEDOT:PSS) on SEBS elastomer (50 μ m thick) before and after mechanical stretching at 20% strain by AFM. In the high magnification images, the granule-to-nanofibril transition was clearly observed on PTGs (Figure S5). Applied 20% strain on SEBS, cracks appeared on all the films, which are mostly perpendicular to the strain direction. However, arising from the underlying graphene, there are obviously less cracks and vaguer parallel wrinkles on PTGs than those on pure PEDOT:PSS (new Figure 2e). It indicates that a smaller strain was transferred onto

above PEDOT:PSS in PTG due to the larger sliding between graphene and substrates. With the increase of graphene layer number, this factor is strengthened accompanied with the formation of more percolation pathways. Multilayer graphene for stretchable electrodes has been reported in the literature of “Bao et al., *Sci. Adv.* 2017; 3: e1700159”, which was also explained as the maintenance of percolation pathway and sliding over each other.

New morphological data of graphene, pure PEDOT:PSS and PTGs before and after tensile strain were added in the revised manuscript (Figure 2e) and supporting information (Figure S5) with corresponding mechanism understanding. Detailed revision is as below.

In the revised manuscript:

“The underlying graphene of PTG results in the better electrical stability upon tensile strain. To understand this mechanism, we performed a systematic morphology study on pure PEDOT:PSS and PTGs (1G+PEDOT:PSS, 2G+PEDOT:PSS) on SEBS elastomer (50 μm thick) before and after mechanical stretching at 20% strain by AFM. In the high magnification images, the granule-to-nanofibril transition was clearly observed on PTGs (Figure S5). Applied 20% strain on SEBS, cracks appeared on all the films, which are mostly perpendicular to the strain direction. However, arising from the underlying graphene, there are obviously less cracks and vaguer parallel wrinkles on PTGs than those on pure PEDOT:PSS film (Figure 2e). It indicates that a smaller tensile strain was transferred onto PTG due to the large sliding between graphene and elastomer. With the increase of graphene layer number, this factor is strengthened accompanied with the formation of more percolation pathways, leading to better electrical durability at strain. This is consistent with the report for multilayer graphene as stretchable electrodes^{40,41}. Thus, to compromise the transparency, conductivity and mechano-electrical stability of PTG, our optimal condition for following epidermal electrophysiology is bi-layer graphene transferred by PEDOT:PSS at both SDS and BSL concentrations of 1.0 wt%.”

Figure 2. (e) Morphological comparison of pure PEDOT:PSS and PTGs (mono- and bi-layer graphene) upon 20% strain by AFM. The strain direction is perpendicular to the cracks and parallel to the wrinkles which are generated by Poisson effect.

Figure S5. Morphological comparison of pure PEDOT:PSS and PTGs (mono- and bi-layer graphene) on SEBS by AFM.

4. For the demonstration of using the bilayer conducting film to measure the electrophysiological signals from the skin surface, the comparison was made with Ag/AgCl gel. However, this is not a really fair or meaningful comparison, as the Ag/AgCl gel is a very different material system with very different charge conducting mechanism. Since PEDOT:PSS with different designs has already been utilized in a lot of works for measuring the electrophysiological signals, the comparison should be made with a dry electrode based on another type of dry-state PEDOT.

Response: We thank the reviewer for raising this point. One motivation to develop dry electrodes is to replace commercial Ag/AgCl gel electrodes. Because the electrolytic gel may irritate the skin and will dry in a certain period, making it impossible for long-term monitoring and diagnosis. Moreover, with the gel interfacing on skin, electrode displacement with respect to skin will cause severe motion artifact. Therefore, in this work we aim at ultra-conformal dry electrodes for low-motion artifact epidermal electrophysiology, and compared the sEMG signal detection with Ag/AgCl gel electrodes.

Following your suggestion, we additionally fabricated pure PEDOT:PSS film at the same preparation condition (eg.: additives in solution and spin-coating), and compared the sEMG signal detection with our PTG. As they are both ultrathin and conformal to skin, their interfacial impedances are comparable to commercial Ag/AgCl, and at 100 Hz, the impedance of pure PEDOT:PSS ($\sim 42 \text{ k}\Omega$) is higher than PTG ($\sim 32 \text{ k}\Omega$) (Figure 5d). In sEMG detection in static and dynamic status (Figure 5h), the amplitude of baseline noise of pure PEDOT:PSS increased significantly from $20.2 \pm 1.9 \mu\text{V}$ to $41.2 \pm 3.2 \mu\text{V}$ (new Figure 5f), and the SNR (Figure S14) decreased from $18 \pm 0.5 \text{ dB}$ to $13 \pm 0.4 \text{ dB}$. On the other hand, PTG electrodes demonstrated a relatively stable baseline noise ($18.4 \pm 1.1 \mu\text{V}$ vs. $23 \pm 2.9 \mu\text{V}$) and SNR ($23 \pm 0.7 \text{ dB}$ vs. $21 \pm 0.3 \text{ dB}$). This is because without the additional graphene, pure PEDOT:PSS film is very brittle and its conductivity would be easily affected by strain. In dynamic status, pure PEDOT:PSS film would severely crack with an obviously alleviated conductivity, leading to unstable electrophysiological signal detection. In particular the situation on largely deformed skin, pure PEDOT:PSS may even fail in sEMG detection. For instance in facial sEMG detection (newly added Figure 6), because the superficial portions of face muscle are uneven, full of wrinkles, and large-shape changed, pure PEDOT:PSS and Ag/AgCl gel electrodes are not able to record sEMG or over a period.

We also compared our work with other PEDOT:PSS electrodes for electrophysiology (Table S3). One is to directly inkjet PEDOT:PSS on commercial decal transfer tattoo paper as a conformal dry

electrode for electrophysiology (*Adv. Sci.* **5**, 1700771 (2018)). The other is to blend PEDOT:PSS with water soluble PU to form adhesive film (*Nat. Commun.* **11**, 4683 (2020)). The former one has limitation in mechano-electrical stability, and the latter one is thick, opaque and not very conductive. In contrast, our PTG is highly conductive, ultrathin, relatively transparent and mechano-electrically stable, showing low-motion artifact electrophysiology with high SNR both in static and dynamic status. To strengthen the application, we optimized the measurements of ECG, EOG, and sEMG by PTG, and improved human-machine interfacing by multi-channel extraction of sEMG.

New comparisons of impedance analysis, and EOG, ECG, sEMG measurements of PTG, pure PEDOT:PSS and Ag/AgCl both in static and dynamic status (Figure 5d-h, Figure S14-15), an improved human-machine interfacing by multi-channel extraction of sEMG (Figure 5i, Figure S17-22, Video S1), a new table compared the capability of electrophysiological detection of PTG and various PEDOT:PSS (Table S3), and corresponding discussion have been added in the revised manuscript. Detailed revision is as below.

In the revised manuscript:

“PTG showed comparable interfacial impedance with PEDOT:PSS dry electrodes and commercial Ag/AgCl gel electrodes, and at 100 Hz, the impedance ($\sim 32 \text{ k}\Omega$) is lower than PEDOT:PSS ($\sim 42 \text{ k}\Omega$) and the Ag/AgCl ($\sim 45 \text{ k}\Omega$).

With low interfacial impedance, PTG was next applied in electrophysiological detection, with pure PEDOT:PSS and Ag/AgCl as control. EOG is to record eye potential and helps noninvasively assess the oculomotor function. Electrodes were adhered on the forehead and lower eyelids. Three types of electrodes showed comparable peaks and valleys with clear eye health information (Figure 5e). ECG is another important electrophysiological signal for detecting heart activity. Dry electrodes were attached on the forearm for measuring ECG signals (Figure 5g). All of the rhythmic-relevant parameters, such as P, Q, R, S, and T waves (Figure S12), can be clearly identified, which is crucial in clinical diagnosis. To record the muscle bioelectricity, sEMG was detected by adhering electrodes on the forearm (Figure 5h). In static state, the sEMG signal amplitude obtained by three kinds of electrodes were comparable, while signal-to-noise ratio (SNR) values obtained by PTG ($23 \pm 0.7 \text{ dB}$) is higher than Ag/AgCl electrodes ($19 \pm 0.5 \text{ dB}$) and PEDOT:PSS electrodes ($18 \pm 0.5 \text{ dB}$). (Figure S14). As electrolytic gel will dry and induce electrode displacement, the major challenge of Ag/AgCl electrodes in electrophysiological detection is the motion artifact. Hence, the motion artifact was evaluated here by baseline noise. We used electromechanical vibrator to mimic skin vibration in arm movement. The electromechanical vibrator was placed near the working electrodes (Figure S15). The average baseline noise of PTG in sEMG recording was just $23 \pm 2.9 \mu\text{V}$, 1.7 times smaller than $38.8 \pm 2.8 \mu\text{V}$ of Ag/AgCl and 1.8 times smaller than $41.2 \pm 3.2 \mu\text{V}$ of PEDOT:PSS in the same condition (Figure 5f). Thus, in dynamic status, PTG electrodes demonstrated a relatively stable SNR ($21 \pm 0.3 \text{ dB}$) than those of Ag/AgCl electrodes ($14 \pm 0.4 \text{ dB}$) and pure PEDOT:PSS ($13 \pm 0.4 \text{ dB}$) (Figure S14). The improved sEMG signal quality in both static and dynamic status can be attributed to the better conformability and mechano-electrical stability of PTG. Motion artifact was also evaluated in ECG (Figure 5e, f). Transited from the static to dynamic status, the baseline noise detected by PTG is relatively stable than Ag/AgCl and PEDOT:PSS electrodes. sEMG is more sensitive to the vibration compared with ECG, because the detected bandwidth (20 – 2000 Hz vs. 0.05-100 Hz) is higher, and muscle contraction comes along with large skin deformation. Although pure PEDOT:PSS was measured in above tests, it is noted that without the additional graphene, pure

PEDOT:PSS film is very brittle and easily break out upon strain. In addition, the infrared image (Figure S16) showed that a minimal level of heat was accumulated upon mounting the PTG electrodes on a volunteer's forearm. Overall, our ultra-conformal PTG exhibited less motion artifact in electrophysiological signal detection, suggesting PTG is potential for accurate human-machine interfacing.

Figure 5. Electrophysiological signals detection of PTG dry electrodes. (d) The impedance analysis of PTG (red), pure PEDOT:PSS (blue) and Ag/AgCl (black) electrodes. (e) EOG measured by PTG (red), pure PEDOT:PSS (blue) and Ag/AgCl (black) electrodes, showing peaks and valleys corresponding to the eyelid movement. (f) sEMG baseline noise comparison of the three types of electrodes in static and dynamic status. (g) ECG measured by PTG (red), pure PEDOT:PSS (blue) and Ag/AgCl (black) electrodes without and with vibration. Characteristic P, Q, R, S and T waves are clearly identified. (h) sEMG measured by PTG (red), pure PEDOT:PSS (blue) and Ag/AgCl (black) electrodes without and with vibration, showing lower-motion artifact detected by PTG. Insets of (e, g, h) are photos of PTG electrodes for EOG, ECG, sEMG. (i) Driving the robotic hand to show various gestures (victory, Arabic numerals of six and three, and holding a centrifuge tube) based on multi-channel sEMG signals extracted by PTG electrodes. Five PTG electrode pairs were adhered on designated locations for acquiring sEMG signals of thumb, index, middle, ring and little fingers of the volunteer, and remotely driving the movement of robotic hand.

The sEMG signals are generated by muscular excitation-contraction, and converting the sEMG signals to control artificial limbs is urgently expected clinically. Firstly the sEMG signals were extracted by our electrodes and then converted to Pulse Width Modulation (PWM) pulses to drive the servo motor (Figure S17-S21). Even with external interference (Figure S21), sEMG signal

extracted by PTG can still precisely control the mechanical claw due to its low-motion artifact detection. Five electrode pairs were adhered on designated locations to respectively detect sEMG signals of thumb, index, middle, ring and little finger of the volunteer, and further control the movement of corresponding fingers of the robotic hand (Figure S22). As shown in Figure 5i and Video S1, various gestures of the robotic hand, such as victory, Arabic numerals of six and three, and holding a centrifuge tube, were demonstrated. Equipped with complicated digital processing and learning algorithms, our ultra-conformal and dry PTG skin electrodes could be applicable in controlling artifact limbs with more complexity and accuracy.”

Figure S14. The baseline noise of ECG (a) and SNR of sEMG (b) measured by Ag/AgCl, PEDOT:PSS and PTG electrodes without and with vibration respectively. The signal-to-noise ratio (SNR) compares the level of signal to noise power, and is calculated as following: $SNR_{dB} = 20\lg\left(\frac{A_{signal}}{A_{noise}}\right)$, where A is root mean square amplitude.

Figure S15. Evaluation of baseline noise in sEMG detection using an electromechanical vibrator to mimic skin vibration in arm movement. The electromechanical vibrator was placed near the working electrodes at an approximate distance of 2 cm. To stably connect PTG to the electrophysiological recording instrument, evaporated Ag films were aided in the connection between soft-hard interfaces.

Figure S17. The overall functional block diagram. It includes front-end analog circuit acquisition and back-end digital signal filtering processing.

$$\begin{aligned}
 f &= \frac{1}{2\pi RC} \\
 &= \frac{1}{2 \times 3.14 \times 4.7 \times 10^4 \times 10^{-9}} \\
 &= 3000\text{Hz} \\
 \text{Gain} &= 1 + \frac{100k\Omega}{33k\Omega} = 4
 \end{aligned}$$

Figure S18. Circuit diagram of a low-pass filter (LPF).

Figure S19. Circuit diagram of a filter, 55-2500 Hz with a gain of 177.

Figure S20. Circuit diagram of the envelope mode with a gain of 22.

Figure S21. (a) Schematic illustration of PTG electrodes on the arm for sEMG detection and robotic claw controlling process. Firstly the sEMG signals were extracted by PTG electrodes and then converted to Pulse Width Modulation (PWM) pulses to drive the servo motor. (b) Demonstration of controlling a robotic claw during vibration. We knocked our arm via a pen to mimic skin vibration in dynamic state, and did not observe any interference while controlling the robotic claw.

Figure S22. Five electrode pairs were adhered on designated locations to respectively control the movement of corresponding fingers of the robotic hand.

Table S3. Comparison of the capability of electrophysiological detection of PTG with various PEDOT:PSS.

Methods	Property	Detection	Control experiments	Application	Reference
PEDOT:PSS tattoo electrode	520 Ω /sq (150 nm thick, transparent), conductive at 5% strain	ECG, sEMG	Similar SNR and noise voltage vs. Ag/AgCl in static status	No human-machine interfacing	Adv. Sci. 5 , 1700771 (2018)
PEDOT:PSS/PU/D-sorbitol hybrid film	545 S/cm (20 μ m thick, opaque), stably conductive at 30% strain	ECG, sEMG	Lower noise voltage vs. Ag/AgCl for ECG in static status	One-channel human-machine interfacing	Nat. Commun. 11 , 4683 (2020)
PTG	4142 S/cm (25 Ω /sq, 100 nm thick, transparent), conductive at 40% strain	ECG, EOG, sEMG, facial sEMG at largely deformed position	Higher SNR and lower noise voltage vs. Ag/AgCl both in static and dynamic status	Multi-channel human-machine interfacing	This work

5. One important reason for using gel as the electrode on skin is to achieve the stable adhesion to the skin. Replacing it with the bilayer film as the dry electrode, the authors didn't explain about how such stable adhesion is achieved.

Response: We thank the reviewer for raising this concern.

In our opinion, gel in Ag/AgCl helps the electrode form conformal contact with skin, whereas double side tape around the gel assists the electrode stably adhere to the skin. When the skin is rough, wrinkled or easily deformed, the commercial electrode would be easily detached, thus requiring additional media to maintain stable contact. The main challenges for using gel electrodes are the severe skin discomfort for long-time use and the motion artifact caused by the displacement. The

contact impedances of gel electrodes were monitored to be increased with time going on (new Figure S13). The degradation indicates that the contact of gel electrodes becomes unstable in a long time, which has also been reported in *Adv. Mater.* 2020, 2001496. To conquer these challenges, dry electrodes without gel are boosting for wearable healthcare.

There are basically two ways to achieve dry electrodes with stable adhesion to skin. One is to utilize intrinsically adhesive material, such as adhesive polymer or adhesive tape. The other way is to prepare ultrathin film as electrodes. For example, ultrathin Au film (~ 300 nm) was reported to be self-adhesive and conformable on skin, and can be used in biopotential recording with less-motion artifact (*Adv. Funct. Mater.* 2018, 1803279). The thickness of our PTG is ~ 80 nm, forming intimate contact on skin (new Figure 5a). To exhibit the stable adhesion on skin, we transferred PTG on porcine skin, and observed if they form conformal interface in between by SEM. The cross-sectional view clearly showed that PTG is conformable to the uneven and curved surface of the skin (new Figure 5a-b and Figure S11). Besides, the Young's modulus of PTG measured by AFM is about 640 kPa (new Figure 5c), which matches the modulus of human skin. Taken together, the ultrathin and low Young's modulus of PTG lead to conformal contact on skin, thus facilitating accurate and stable electrophysiological signal detection without gel.

New photo image of PTG on the junction of a finger (Figure 5a), SEM image of PTG on porcine skin (Figure 5b and Figure S11), AFM Young's modulus map of PTG (Figure 5c, Figure S10), the dependence of contact impedance of gel electrodes on time (Figure S13), and corresponding discussion have been added in the revised manuscript.

In the revised manuscript:

“Low-motion artifact electrophysiological monitoring. Interfacial impedance is a critical parameter in epidermal electrophysiological measurement. Without additional gel, dry electrodes would reduce the contact impedance by forming conformal interface with skin. According to the Young's modulus map under AFM, average Young's modulus of PTG is about 640 kPa (Figure 5c, Figure S10), which matches the modulus of human stratum corneum. Transferred onto a human ring finger full of wrinkles, PTG exhibited intimate contact on it (Figure 5a). The conformability is also confirmed by SEM imaging of PTG on porcine skin (Figure 5b, Figure S11). The ultrathin PTG with low Young's modulus leads to conformal contact with wrinkled skin^{45, 46}. The interfacial impedance analysis between electrodes and skin was conducted on the human arm with electrodes placed on the surface of it (Figure 5d). PTG showed comparable interfacial impedance with PEDOT:PSS dry electrodes and commercial Ag/AgCl gel electrodes, and at 100 Hz, the impedance (~ 32 k Ω) is lower than PEDOT:PSS (~ 42 k Ω) and the Ag/AgCl (~ 45 k Ω).

With low interfacial impedance, PTG was next applied in electrophysiological detection, with pure PEDOT:PSS and Ag/AgCl as control. EOG is to record eye potential and helps noninvasively assess the oculomotor function. Electrodes were adhered on the forehead and lower eyelids. Three types of electrodes showed comparable peaks and valleys with clear eye health information (Figure 5e). ECG is another important electrophysiological signal for detecting heart activity. Dry electrodes were attached on the forearm for measuring ECG signals (Figure 5g). All of the rhythmic-relevant parameters, such as P, Q, R, S, and T waves (Figure S12), can be clearly identified, which is crucial in clinical diagnosis. To record the muscle bioelectricity, sEMG was detected by adhering electrodes on the forearm (Figure 5h). In static state, the sEMG signal amplitude obtained by three kinds of

electrodes were comparable, while signal-to-noise ratio (SNR) values obtained by PTG (23 ± 0.7 dB) is higher than Ag/AgCl electrodes (19 ± 0.5 dB) and PEDOT:PSS electrodes (18 ± 0.5 dB). (Figure S14). As electrolytic gel will dry and induce electrode displacement, the major challenge of Ag/AgCl electrodes in electrophysiological detection is the motion artifact. Hence, the motion artifact was evaluated here by baseline noise. We used electromechanical vibrator to mimic skin vibration in arm movement. The electromechanical vibrator was placed near the working electrodes (Figure S15). The average baseline noise of PTG in sEMG recording was just 23 ± 2.9 μ V, 1.7 times smaller than 38.8 ± 2.8 μ V of Ag/AgCl and 1.8 times smaller than 41.2 ± 3.2 μ V of PEDOT:PSS in the same condition (Figure 5f). Thus, in dynamic status, PTG electrodes demonstrated a relatively stable SNR (21 ± 0.3 dB) than those of Ag/AgCl electrodes (14 ± 0.4 dB) and pure PEDOT:PSS (13 ± 0.4 dB) (Figure S14). The improved sEMG signal quality in both static and dynamic status can be attributed to the better conformability and mechano-electrical stability of PTG. Motion artifact was also evaluated in ECG (Figure 5e, f). Transited from the static to dynamic status, the baseline noise detected by PTG is relatively stable than Ag/AgCl and PEDOT:PSS electrodes. sEMG is more sensitive to the vibration compared with ECG, because the detected bandwidth (20 – 2000 Hz vs. 0.05-100 Hz) is higher, and muscle contraction comes along with large skin deformation. Although pure PEDOT:PSS was measured in above tests, it is noted that without the additional graphene, pure PEDOT:PSS film is very brittle and easily break out upon strain.”

Figure 5. Electrophysiological signal detection of PTG dry electrodes. (a) A photograph of PTG conformed on a human ring finger. (b) A SEM image of cross-sectional view of PTG (top) on porcine skin (bottom). (c) AFM Young’s modulus map of PTG. (d) The impedance analysis of PTG (red), pure PEDOT:PSS (blue) and Ag/AgCl (black) electrodes. (h) sEMG measured by PTG (red), pure PEDOT:PSS (blue) and Ag/AgCl (black) electrodes without and with vibration, showing lower-motion artifact detected by PTG. Inset of (h) is photo of PTG electrodes for sEMG measurements.

Figure S10. AFM height images and Young's modulus images of PTG with BSL (a, c) and without BSL (b, d).

Bending stiffness (\overline{EI}_{sum}) of our PTG electrode is calculated as following:

$$\overline{EI} = \frac{1}{12(1-\nu^2)} \overline{E}h^3$$

$$\overline{EI}_{sum} = \alpha \overline{EI}_{electrode} + (1-\alpha) \overline{EI}_{substrate}$$

Where \overline{E} represents Young's modulus, h represents thickness, ν represents Poisson ratio, and α represents the area ratio of electrode to substrate. Together with the small Young's modulus and ultrathin nature, bending stiffness of our PTG electrode (with tattoo substrate) is ~ 0.00236 pNm. Correspondingly, according to above formula, bending stiffness of skin is ~ 0.00156 pNm (Young's modulus of skin is ~ 150 kPa, and the thickness of stratum corneum is ~ 15 μm)¹. They match with each other very well.

Figure S11. SEM image of PTG (bottom) on porcine skin (top).

Figure S13. Dependence of contact impedance of Ag/AgCl gel electrodes on time.

6. The last demonstration of using the measured EMG signal to control the robotic arm doesn't really show the advantage of the improved signal-to-noise ratio obtained from the bilayer dry electrodes.

Response: We thank the reviewer for raising this concern. Following your advice, we extended PTG to the acquisition of facial sEMG and utilized the facial expression to control a robotic hand. Because face is uneven, full of wrinkles, and inevitably shape changed, it requires the facial sEMG electrodes to be highly conductive, ultra-conformal to skin as well as mechano-electrically stable. PTG indeed can accurately acquire facial sEMG signals. In contrast, gel electrodes are easily detached from the face, and the brittle PEDOT:PSS becomes completely insulative especially when the face is in motion. In addition, PTG electrodes do not affect the laser speckle contrast imaging due to the optical transparency. Thus, PTG is able to simultaneously monitor sEMG and laser speckle contrast imaging at the same location, providing more information for the diagnosis and treatment of facial nerve diseases. With the accurate monitoring of sEMG by PTG, we applied it to control the robotic hand by the multi-channel transformation of the facial muscle expression. Taken together, the high conductivity, ultra-conformability, certain mechano-electrical stability and transparency contribute PTG as an ideal electrode for the acquisition of facial sEMG without detriment to laser speckle contrast imaging, as well as being able to control artifact limbs.

A completely new Figure 6, Video S2 and corresponding description were added in the revised manuscript. Detailed revisions are listed as below.

In the revised manuscript:

“Facial sEMG acquisition and facial expression-robot interaction. EMG is an important way to evaluate facial muscle function and analyze facial muscle expression (Figure 6a). Compared to needle EMG (nEMG), sEMG is superior in painlessness, invasiveness, large facial muscle area characterization, and monitoring over time. Thus, sEMG is more practical in wearable healthcare to self-evaluate facial nerve disorder for both pre- and post-diagnosis. Considering that the superficial portions of the face muscles are uneven, full of wrinkles, and inevitably large-shape changed, facial sEMG electrodes being highly conductive, ultra-conformal as well as mechano-electrical stable are necessary. It is still a big challenge to detect facial sEMG stably and accurately both in laboratory and in clinic. To verify the capability of PTG as facial sEMG electrodes, we applied a pair of them on levator labii superioris alaeque nasi on one side of the face with Ag/AgCl as control on the other side (Figure 6d). PTG showed more stable signal quality than Ag/AgCl in facial sEMG detection (Figure 6b). This phenomenon can be attributed to the low-motion artifact electrophysiology of PTG

on skin. In contrast, Ag/AgCl gel electrodes are easily detached from the facial skin resulted from the displacement of gel, and the PEDOT:PSS is too brittle and completely insulative when the facial skin is stretched in dynamic state. With the addition of graphene underlying PEDOT:PSS, the electrical stability of PTG upon stretching is greatly enhanced (Figure 2d), facilitating the accurate sEMG signal acquisition in particular in motion of the face. Laser speckle contrast imaging technique (LSCI) is another important technique for facial nerve diseases diagnosis. It can evaluate the function of facial microcirculation, such as blood flow velocity and blood perfusion. Leveraging laser speckle contrast imaging into sEMG measurements, it will be beneficial to the deep understanding of pathological process and mechanism (Figure 6c). Commercial Ag/AgCl gel electrodes are opaque. Utilizing the relatively transparent PTG dry electrodes, simultaneous monitoring sEMG and speckle imaging at the same location can be realized, which will provide more information to the diagnosis and treatment of facial nerve palsy (Figure 6d and 6e). The blood perfusion under PTG electrodes can be measured by LSCI whereas the blood perfusion under Ag/AgCl electrodes cannot be measured by LSCI. Moreover, with the accurate monitoring of sEMG by PTG, we applied it to control the robotic hand by facial muscle expression. Three pairs of PTG were attached on zygomaticus, risorius and corrugator to control the index, middle finger, and wrist of the robotic hand respectively (Video S2). When the volunteer smiles, the robotic hand shows the gesture of victory (Figure 6f). When the volunteer extends and knits her brows, the robotic hand is able to rotate its twist correspondingly (Figure 6g). Taken together, the high conductivity, ultra-conformability, certain mechano-electrical stability and transparency contributes PTG as an ideal electrode for the acquisition of facial sEMG without detriment to laser speckle contrast imaging, as well as being able to controlling artifact limbs (Table S3).”

Figure 6. Facial sEMG acquisition and facial expression-robot interaction. (a) A photo of facial expression muscles. (b) Facial EMG measured by PTG (red) and Ag/AgCl (black) electrodes. PTG showed lower-motion artifact. (c) Schematic illustration of simultaneous monitoring sEMG and speckle imaging at the same location. (d) A photograph of PTG electrodes connected by metal for facial sEMG detection on levator labii superioris alaeque nasi on one side of the face with Ag/AgCl as control on the other side. (e) A laser speckle contrast imaging corresponding to (d), The blood perfusion under PTG electrodes can be measured by laser speckle contrast imaging technique (LSCI) whereas the blood perfusion under Ag/AgCl electrodes cannot be measured by LSCI. (f) Driving the robotic hand based on multi-channel facial sEMG signals extracted by PTG electrodes. Two PTG electrode pairs were adhered on zygomaticus and risorius to control the index and middle finger, respectively. Right photos showing that the robotic hand at the gesture of victory when the volunteer smiles. (g) PTG electrodes were adhered on corrugator to control the wrist of the robotic hand. Photos showing that the robotic hand rotates its twist when the volunteer extends and knits her brows.

Reviewer #2 (Remarks to the Author):

The manuscript by Zhao et al. reports an interesting approach to employ PEDOT:PSS as a transfer medium for CVD graphene, which produces a bilayer composite material termed as PTG. The process essentially eliminates the need to remove the sacrificial transfer medium of PMMA in the conventional technique. As a transparent electrode, the excellent optoelectronic performances of PTG comes from the synergistic charge transfer effects between graphene and PEDOT:PSS supported by Raman spectroscopy and GIWAXS. In addition, PTG also show decent mechanical deformability to function as conformal skin electrodes to acquire electrophysiological signals with low motion artifacts. I recommend the manuscript to be considered for publication after minor revisions regarding the following aspects:

We thank the reviewer for the positive comments of our work. Below we addressed the reviewer's questions point-to-point.

(1) The origin of the stretchability of PTG is less clear. I wonder whether it comes from certain intrinsic mechanisms or structural reasons (e.g. surface wrinkles formed during the transfer process). Additional discussions/experimental evidences should be provided. In addition, the reason for improved stretchability upon increasing the number of underlying graphene layers should be explained.

Response: We thank the reviewer for the comment. PTG exhibits a certain electrical durability upon tensile strain. This is mainly attributed to the sliding between graphene and the substrate, and the formation of percolation pathway with multi-layer graphene.

To understand this mechanism, we newly performed a systematic morphology study on graphene, pure PEDOT:PSS and PTGs (1G+PEDOT:PSS, 2G+PEDOT:PSS) on SEBS elastomer (~ 50 μm thick) before and after mechanical stretching at 20% strain by AFM. In the high magnification images, the granule-to-nanofibril transition was clearly observed on PTGs (Figure S5). Applied 20% strain on SEBS, cracks appeared on all the films, which are mostly perpendicular to the strain direction. However, arising from the underlying graphene, there are obviously less cracks and vaguer parallel wrinkles on PTGs than those on pure PEDOT:PSS (revised Figure 2e). It indicates that a smaller strain was transferred onto above PEDOT:PSS in PTG due to the large sliding between graphene and substrates. With the increases in graphene layer number, the sliding is strengthened accompanied with the formation of more percolation pathways. Multilayer graphene for stretchable electrodes has been reported in the literature of "Bao et al., Sci. Adv. 2017; 3: e1700159", which was also explained as the maintenance of percolation pathway and sliding over each other.

New morphological data of graphene, pure PEDOT:PSS and PTGs before and after tensile strain were added in the revised manuscript (Figure 2e) and supporting information (Figure S5) with corresponding mechanism understanding. Detailed revision is as below.

In the revised manuscript:

"The underlying graphene of PTG results in the better electrical stability upon tensile strain. To understand this mechanism, we performed a systematic morphology study on pure PEDOT:PSS and PTGs (1G+PEDOT:PSS, 2G+PEDOT:PSS) on SEBS elastomer (50 μm thick) before and after

mechanical stretching at 20% strain by AFM. In the high magnification images, the granule-to-nanofibril transition was clearly observed on PTGs (Figure S5). Applied 20% strain on SEBS, cracks appeared on all the films, which are mostly perpendicular to the strain direction. However, arising from the underlying graphene, there are obviously less cracks and vaguer parallel wrinkles on PTGs than those on pure PEDOT:PSS film (Figure 2e). It indicates that a smaller tensile strain was transferred onto PTG due to the large sliding between graphene and elastomer. With the increase of graphene layer number, this factor is strengthened accompanied with the formation of more percolation pathways, leading to better electrical durability at strain. This is consistent with the report for multilayer graphene as stretchable electrodes^{40,41}. Thus, to compromise the transparency, conductivity and mechano-electrical stability of PTG, our optimal condition for following epidermal electrophysiology is bi-layer graphene transferred by PEDOT:PSS at both SDS and BSL concentrations of 1.0 wt%.”

Figure 2. (e) Morphological comparison of pure PEDOT:PSS and PTGs (mono- and bi-layer graphene) upon 20% strain by AFM. The strain direction is perpendicular to the cracks and parallel to the wrinkles which are generated by Poisson effect.

Figure S5. Morphological comparison of pure PEDOT:PSS and PTGs (mono- and bi-layer graphene) by AFM.

(2) *The authors claim the addition of bis(trifluoromethane) sulfonimide lithium salt into PEDOT:PSS improves the electrical conductivity and the mechanical stretchability. Although it is intuitive that ionic compound may boost the conductivity of PEDOT:PSS, the benefit of BSL salt on mechanical deformability requires additional clarifications and explanations.*

Response: We thank the reviewer for the valuable suggestion. To demonstrate the benefit of BSL additives on mechanical deformability of PTG, we compared the Young’s modulus maps of PTGs without and with BSL (1.0 wt%) under AFM. The average Young’s modulus of PTG with BSL is ~ 640 kPa, whereas that of PTG without BSL is ~ 2.1 MPa (new Figure 5c, Figure S10). Also, with the addition of BSL, the crystallinity of PEDOT is getting better from the height images (Figure 4b, Figure S10) and GIWAXS data (Figure 4a). These indicate that BSL additives soften the PSS

domain, leading to the decreased Young's modulus and increased mechanical deformability of PTG.

New Young's modulus maps and AFM height images of PTGs with and without BSL additives (1.0 wt%) were added in the revised manuscript (Figure 5c) and supporting information (Figure S10) with corresponding description. Detailed revision is as below.

In the revised manuscript:

“Interfacial impedance is a critical parameter in epidermal electrophysiological measurement. Without additional gel, dry electrodes would reduce the contact impedance by forming conformal interface with skin. According to the Young's modulus map under AFM, average Young's modulus of PTG is about 640 kPa (Figure 5c, Figure S10), which matches the modulus of human stratum corneum. Transferred onto a human ring finger full of wrinkles, PTG exhibited intimate contact on it (Figure 5a). The conformability is also confirmed by SEM imaging of PTG on porcine skin (Figure 5b, Figure S11). The ultrathin PTG with low Young's modulus leads to conformal contact with wrinkled skin^{45,46}.”

Figure 5 (c) AFM Young's modulus map of PTG. They showed PTG is conformal with skin due to the small thickness and low Young's modulus.

Figure S10. Young's modulus maps and AFM height images of PTGs with (a, c) and without BSL (b, d) additives (1.0 wt%).

Bending stiffness (\overline{EI}_{sum}) of our PTG electrode is calculated as following:

$$\overline{EI} = \frac{1}{12(1-\nu^2)} \overline{E}h^3$$

$$\overline{EI}_{sum} = \alpha \overline{EI}_{electrode} + (1-\alpha) \overline{EI}_{substrate}$$

Where \overline{E} represents Young's modulus, h represents thickness, ν represents Poisson ratio, and α represents the area ratio of electrode to substrate. Together with the small Young's modulus and ultrathin nature, bending stiffness of our PTG electrode (with tattoo substrate) is ~ 0.00236 pNm. Correspondingly, according to above formular, bending stiffness of skin is ~ 0.00156 pNm. (Young's modulus of skin is ~ 150 kPa, and the thickness of stratum corneum is ~ 15 μm)¹. They match with each other very well.

(3) In order to put the optoelectronic performances of PTG into the research context, a table summarizing the state-of-the-art of graphene transparent electrodes should be provided.

Response: We thank the reviewer for the suggestion. A comparison table of the state-of-the-art of graphene-based transparent electrodes were summarized in Table S1 as following. From this table, it is observed that pure graphene structures have relatively high resistance and high transparency. With additional conductive species, the resistance can be lowered down at a compromising transparency. Graphene/AgNWs hybrid film exhibits the best combination of conductivity and transparency. However, AgNWs are not air-stable due to oxidation, which are not suitable in epidermal electrophysiology for long-time use. Because PEDOT:PSS is solution-processible, conductive, and able to form a thin film on graphene with the help of additives, we employed it as a medium to transfer graphene directly. Their synergistic enhancement in conductivity and mechano-electrical stability leads PTG as an ideal ultra-conformal skin-electrode for low-motion artifact electrophysiology.

Table S1. Comparison of the optoelectronic properties of the state-of-the-art of graphene-based transparent electrodes.

Methods	R _{sh} (Ω/sq)	Transparency	References
Solution-Processed rGO	$10^2 - 10^3$	80%	ACS Nano 2, 463-470 (2008)
CVD graphene on Cu	2100 (1 layer) 350 (4 layers)	98% 90%	Nano Lett. 9, 4359-4363 (2009)
CVD graphene on Ni	280	80%	Nature 457, 706-710 (2009)
Graphene/AgNWs hybrid film	8	94%	Nano Lett. 15, 4206-4213 (2015)
MGG	185	88%	Sci. Adv. 3 e1700159 (2017)
GEF /monolayer graphene	150	83%	ACS Materials Lett. 2, 999-1007 (2020)
PTG	45	80%	This work

Reviewer #1 (Remarks to the Author):

In the revision, the authors added more experiments and discussions to address the questions from the reviewer. Regarding the material properties, the comparison with the previous works is much clearer and more convincing; the understandings about the mechanistic behaviors are also deepened. However, my questions about the practical merits of the PTG electrode for working as electrophysiology dry electrodes remain largely unaddressed. Although the overall engineering target is to replace the gel electrodes as the current approach for electrophysiology measurement, since dry electrodes based on engineered PEDOT:PSS has already been reported in several works before, such as "Fully organic compliant dry electrodes self-adhesive to skin for long-term motion-robust epidermal biopotential monitoring. Nat. Commun. 2020", the comparisons in these demonstrations in Fig. 6 should be made with other types of PEDOT:PSS dry electrodes, rather than the gel electrode.

Response: We thank the reviewer for the positive comments on our revised manuscript.

Because of the low sheet resistance (24 Ω /sq, 4142 S/cm), high transparency, mechano-electrical stability, and ultra-thin skin-conformal nature, we believe our PTG has the following practical merits as electrophysiological electrodes: 1) uniqueness in facial nerve diseases; 2) higher signal quality and lower motion artifact in sEMG detection; 3) long-term monitoring of the very weak EEG (electroencephalogram) signals. To compare with other PEDOT:PSS electrodes, in particular the PWS (blend of PEDOT:PSS, waterborne polyurethane and D-sorbitol) recently published in "Fully organic compliant dry electrodes self-adhesive to skin for long-term motion-robust epidermal biopotential monitoring. Nat. Commun. 2020, 11, 4683", we elaborate above merits separately as below and summarized them in Table S1.

1. Uniqueness in facial nerve diseases:

PTG is able to accurately acquire facial sEMG without detriment to laser speckle contrast imaging, as well as to control artificial limbs (Figure 6) due to its high conductivity, transparency, mechano-electrical stability, and ultra-thin skin-conformal nature. Simultaneously achieving sEMG and doing laser speckle contrast imaging will provide more accurate information for the deep understanding of pathological process and mechanism of facial nerve diseases, which will be beneficial for its diagnosis and treatment. In contrast, pure PEDOT:PSS without underlying graphene is easily fractured, become insulative, and cannot detect sEMG when facial skin is continuously in motion (Figure S24). In terms of other engineered PEDOT:PSS dry electrodes, such as PWS, the authors did not apply it on the largely deformed facial skin surface. However, deduced from its signal quality in sEMG on forearm, PWS is very likely not to be able to detect facial sEMG accurately due to its low conductivity (545 S/cm vs 4142 S/cm) and large thickness (bulk vs nm scale). In addition, PWS electrode is not compatible with laser speckle contrast imaging, as it is opaque and will completely block the laser.

2. Higher signal quality and lower motion artifact in sEMG acquisition:

To compare the sEMG signal quality, we performed the same sEMG detection on PTG with that on PWS. By holding the grip dynamometer at 3 kPa, 4 kPa and 5 kPa, the sEMG intensity on forearm gradually increased with larger gripping force, exhibiting an increasing trend of signal-to-noise ratio ($SNR_{dB} = 20\lg(\frac{A_{signal}}{A_{noise}})$). Although our test forces are smaller than those on PWS (0.21, 0.27, 0.33GPa), the SNRs are much larger (19, 23, 26 dB vs 13, 16, 19 dB), indicating the better signal

quality of PTG over PWS (Figure R1). When finger is in flexion/extension, weak sEMG is generated. Again, PTG showed larger SNRs than PWS on the sEMG acquisition at each finger (from thumb to little finger: 21, 19, 21, 25, 20 dB vs 11, 13, 17, 11, 14 dB). We attribute the better signal quality of PTG over PWS to the higher conductivity and skin-conformability.

Figure R1. Comparison of signal quality of sEMG obtained by PTG and PWS electrodes. (a, b) SNRs of sEMG generated at different gripping force measured by PTG (a) and PWS (b) electrodes. (c, d) SNRs of sEMG generated by each finger acquired by PTG (c) and PWS (d) electrodes.

3. Long-term monitoring of the very weak EEG signals

One biggest motivation for developing dry electrodes is to reduce motion artifact coming from the displacement of gel on skin when long-term monitoring of electrophysiology. Due to the very weak signal strength, long-term monitoring of EEG is extremely challenging, in particular for the status of exercising. Normally, for EEG monitoring, the volunteer sits in a comfortable position peacefully and experiences brain stimulus, such as doing math quiz, watching videos or listening to music. As PTG dry electrodes are ultra-thin and the attachment is comfortable, we therefore applied PTG on a volunteer for long-term monitoring of EEG without taking off the electrode pairs. During the monitoring of 12 hours, the volunteer slept, exercised and recovered to calm with two PTG electrodes mounted in the frontal region (Fp1 and Fp2) of the brain and one reference electrode behind the ear. The EEG signal quality remains decent, despite that the volunteer is soaked in sweat after exercising. Figure 7a-c showed three typical EEG signals corresponding to the three statuses of brain. After Fast Fourier Transformation (FFT) treatment (Figure 7d-f), the brain waves can be divided into several parts: delta wave (0–4 Hz), theta wave (4–8 Hz), alpha wave (8–13 Hz), beta wave (13–30 Hz), and gamma wave (40 Hz or higher). To differentiate various statuses of the brain, alpha and beta waves were extracted and compared separately (Figure 7g-l). The intensity of alpha and beta waves at exercising is obviously stronger than that at sleeping and calm status. This is in accord with the cognition that exercise improves our concentration. The stable recording of EEG signals can be explained by the intimate contact of ultra-thin PTG on skin despite the extension/contraction and sweat secretion of skin. This can be verified by the stable contact impedance at 10 Hz as a function of time (Figure S25). The capability of long-term monitoring of EEG suggests that our PTG is an ideal skin electrode candidate, facilitating the future healthcare and human-machine interfacing based on accurate EEG acquisition.

Regarding the practical merits and advantages of PTG as electrophysiological electrodes, we discussed the fractured feature of PEDOT:PSS only (new Figure S24), compared sEMG detection between PTG and PWS (new Figure 5i and j, Figure S24, Table S1), and added the long-time monitoring of EEG (new Figure 7). Detailed revisions in the revised manuscript and supporting information are listed as below.

In the revised manuscript:

In the section of “Facial sEMG acquisition and facial expression-robot interaction”:

“..., and the PEDOT:PSS is too brittle and completely insulative when the facial skin is stretched in dynamic state (Figure S24).”

Figure S24. Facial sEMG measured by PEDOT:PSS only. With time going on, the movement of face is unavoidable (e.g.: smile, eating, talking...), leading PEDOT:PSS to be failed in sEMG measurement. Inset is a photo of the PEDOT:PSS dry electrode on face. After a short facial sEMG measurement, the PEDOT:PSS electrode is fractured and become insulative.

Figure 5. Electrophysiological signal detection of PTG dry electrodes. (a) A photograph of PTG conformed on a human ring finger. (b) A SEM image of cross-sectional view of PTG (top) on porcine skin (bottom). (c) AFM Young's modulus map of PTG. (d) The impedance analysis of PTG (red), pure PEDOT:PSS (blue) and Ag/AgCl (black) electrodes. (e) EOG measured by PTG (red), pure PEDOT:PSS (blue) and Ag/AgCl (black) electrodes, showing peaks and valleys corresponding to the eyelid movement. (f) sEMG baseline noise comparison of the three types of electrodes in static and dynamic status. (g) ECG measured by PTG (red), pure PEDOT:PSS (blue) and Ag/AgCl (black) electrodes without and with vibration. Characteristic P, Q, R, S and T waves are clearly identified. (h) sEMG measured by PTG (red), pure PEDOT:PSS (blue) and Ag/AgCl (black) electrodes without and with vibration, showing lower-motion artifact detected by PTG. Insets of (e, g, h) are photos of PTG electrodes for EOG, ECG, sEMG measurements. (i) sEMG measured by PTG at different gripping force, showing that sEMG intensity becomes higher at larger gripping force. (j) sEMG acquired by PTG from the finger (thumb, index, middle, ring and little fingers) motion. (k) Driving the robotic hand to show various gestures (victory, Arabic numerals of six and three, and holding a centrifuge tube) based on multi-channel sEMG signals extracted by PTG electrodes.

Five PTG electrode pairs were adhered on designated locations for acquiring sEMG signals of thumb, index, middle, ring and little fingers, and remotely driving the movement of robotic hand.

In the section of “Low-motion artifact electrophysiological monitoring”:

“Five electrode pairs were adhered on designated locations to respectively detect sEMG signals of thumb, index, middle, ring and little finger of the volunteer (Figure 5j), showing high detected quality with SNRs around 20 dB (Figure S23).”

Figure S23. The SNRs of sEMG on forearm generated with different gripping force (a) and the SNRs of sEMG around the wrist generated by each finger.

Table S1. Comparison of the practical merits of various PEDOT:PSS electrodes for electrophysiology.

Methods	Material Properties	Signal Detection	ECG	sEMG (facial sEMG)	EEG	Human-machine interfacing	Ref.
Inject-printed PEDOT:PSS	520 Ω /sq (150 nm thick, transparent), conductive at 5% strain	ECG, sEMG	Similar SNR and noise voltage vs. Ag/AgCl in static status	Similar SNR and noise voltage vs. Ag/AgCl in static status	No EEG data	No	Adv. Sci. 5 , 1700771 (2018)
PWS	545 S/cm (20 μ m thick, opaque), stably conductive at 30% strain	ECG, sEMG, EEG	Lower noise voltage vs. Ag/AgCl both in static and dynamic status	No comparison with other electrodes (no facial sEMG data)	EEG at the stimuli of sound	One-channel	Nat. Commun. 11 , 4683 (2020)
PTG	4142 S/cm (25 Ω /sq, 100 nm thick, transparent), conductive at 40% strain	ECG, EOG, sEMG at largely deformed position, EEG	Higher SNR and lower noise voltage vs. Ag/AgCl both in static and dynamic status	Higher SNR and lower noise voltage vs. Ag/AgCl both in static and dynamic status	EEG monitoring for ~12 hours	Multi-channel	This work

“**Long-time monitoring of EEG signals.** Electroencephalogram (EEG) recording is to measure the electrical activity of the brain, providing helpful information for the diagnosis or treatment of the brain disorders. Due to the very weak signal strength (μ V scale), long-time monitoring of EEG is

extremely challenging, in particular for the status of exercising. Normally, for EEG monitoring, the volunteer sits in a comfortable position peacefully and experiences brain stimulus, such as doing math quiz, watching videos or listening to music. As PTG is ultra-thin and the attachment is comfortable, we therefore applied PTG on a volunteer for long-time monitoring of EEG without taking off the electrode pairs. During the monitoring of 12 hours, the volunteer slept, exercised and recovered to calm with two PTG electrodes mounted in the frontal region (Fp1 and Fp2) of the brain and one reference electrode behind the ear⁴⁷. The EEG signal quality is decent, despite that the volunteer is at various statuses and even soaked in sweat after exercising. Figure 7a-c showed three typical EEG signals corresponding to the three statuses of brain. After Fast Fourier Transformation (FFT) treatment (Figure 7d-f), the brain waves can be divided into five parts: delta wave (0–4 Hz), theta wave (4–8 Hz), alpha wave (8–13 Hz), beta wave (13–30 Hz), and gamma wave (40 Hz or higher)^{47,48}. To differentiate various statuses of the brain, alpha and beta waves were extracted and compared separately (Figure 7g-l). The intensity of alpha and beta waves at exercising is obviously stronger than that at sleeping and calm status. This is in accord with the cognition that exercise improves our concentration. The stable recording of EEG signals can be explained by the intimate contact of ultra-thin PTG on skin regardless the extension/contraction and sweat secretion of skin. This can be verified by the stable contact impedance at 10 Hz as a function of time (Figure S25). The capability of long-time monitoring of EEG suggests that PTG is an ideal skin-electrode candidate, facilitating the future healthcare and human-machine interfacing based on accurate EEG acquisition.”

Figure 7. Long-time monitoring of EEG. (a-c) Original EEG signals, (d-f) Fast Fourier Transformed

EEG signals, (g-i) the extracted alpha waves and (j-l) beta waves when the volunteer was at status of sleeping, exercising (skipping rope) and recovering to calm state after ~ 1 hour.

Moreover, regarding my previous question about the stable adhesion of the PTG electrode to the skin surface, the response addressed it mainly from the aspect of achieving good conformability, which is a different aspect than the physical/chemical bonding at such an interface to prevent the delamination or detachment caused by motion or skin deformations. Even for the conformability, the experimental evidence shown by the authors is just based on a freestanding PTG film with a very low thickness, which is different than its use in real electrophysiological measurements, in which deposited Ag patterns and a SEBS or paper substrate is needed, which will largely the actual thickness of the entire sheet. With the Ag on top, the stretchability of the electrode could also get much worse.

Response: We thank the reviewer for the comment.

The electrophysiological signal sensor is composed of two parts: the electrodes for sensing electrophysiological signal and the connection between electrodes and the measurement kit. PTG works as the sensing electrode, and the connection between soft electrodes and the hard measurement kit is achieved by the deposited metal film referred to literatures (*Adv. Funct. Mater.* **28**, 1803279 (2018), *Adv. Sci.* **5**, 1700771 (2018)). The adhesive force between ultra-thin PTG and skin is simply Van der Waals force (*Science*. **333**, 838-843 (2011)). Due to the ultra-thin nature, PTG and skin forms good conformability, leading to stable attachment of PTG electrode to the skin surface. In our last response we did morphological observation and mechanical force measurement by AFM to illustrate the good conformability of PTG to skin. Herein, we additionally monitored the contact impedance of PTG electrode on skin as a function of time in real measurement (PTG on tattoo paper with Ag as connection) to verify the stable attachment, and explained it from the matched bending stiffness between PTG electrode and skin as following:

During long-time monitoring of EEG, the contact impedance of PTG electrodes on skin were tested. The volunteer wore PTG electrodes (PTG on tattoo paper with Ag as connection) in frontal region of her brain for 12 hours. During this period, she slept, skipped rope for 0.5 hour, and rest. The extension/contraction and sweat secretion of the skin did not affect the contact impedance obviously. 10 Hz is the center frequency for alpha wave in EEG. From the changing trend of contact impedance at 10 Hz vs time (Figure S25), it is confirmed that the attachment between PTG electrodes and skin are stable.

To interpret the stable attachment of PTG electrodes on skin during electrophysiological measurement, we introduced the study of literature (*Adv. Funct. Mater.* **28**, 1803279 (2018), *Nano Lett.* **19**, 1577–1586 (2019)) in ultra-thin electrodes by the concept of bending stiffness. Bending stiffness refers to the amount of stiffness that the subject will deflect. The smaller the stiffness, the larger the deflection will occur. According to the formula (1) and (2), bending stiffness (\overline{EI}_{sum}) of our PTG electrode can be calculated as following:

$$\overline{EI} = \frac{1}{12(1-\nu^2)} \overline{E} h^3 \quad (1)$$

$$\overline{EI}_{sum} = \alpha \overline{EI}_{electrode} + (1 - \alpha) \overline{EI}_{substrate} \quad (2)$$

Where \overline{E} represents Young's modulus, h represents thickness, ν represents Poisson ratio, and α

represents the area ratio of electrode to substrate. In this work, the thickness of PTG is about 100 nm. And in real electrophysiological measurements, we transfer PTG on decal transfer tattoo paper, and the thickness of ethylcellulose layer in decal transfer tattoo paper is about 450 nm. Together with the small Young's modulus and ultra-thin nature, bending stiffness of our PTG electrode (with tattoo substrate) is ~ 0.00236 pNm. Correspondingly, according to above formula, bending stiffness of skin is ~ 0.00156 pNm. (Young's modulus of skin is ~ 150 kPa, and the thickness of stratum corneum is ~ 15 μm). They match with each other very well. The matched bending stiffness leads to the ultra-conformability between PTG electrodes and skin. Once two subjects form ultra-conformability with each other, they tend to attach intimately just like contact lens with curved eyes (*Nat. Commun.* **9**, 2334 (2018)). The physical/chemical interaction in between the interface is just Van der Waals force. Besides, the existence of sweat may introduce capillary force, which can enhance the intimate contact and stable attachment (*ACS Nano* **15**, 2800-2811 (2021)).

Considering the PTG film is too thin, how to achieve robust connection between ultra-thin electrodes and external electrophysiological kits is difficult. Referring to literature (*Adv. Funct. Mater.* **28**, 1803279 (2018), *Adv. Sci.* **5**, 1700771 (2018), *ACS Nano.* **11**, 7634-7641 (2017)), we additionally deposit 50 nm-thick Ag patterns on tattoo to contact with PTG film. The center region for sensing the electrophysiological signals is still the PTG film, and the Ag patterns are just for connection to the measurement kits. Within the range of skin deformation, the mechano-electrical stability for Ag or Au is sufficient (*Appl. Phys. Lett.* **94**, 071902 (2009)).

Regarding the stable adhesion of PTG electrodes on skin, we monitored its contact impedance as a function of time (Figure S25), and strengthened the explanation from the matched bending stiffness (both revised manuscript and Supporting Information). Detailed revisions are listed as below.

In the revised manuscript:

“The stable recording of EEG signals can be explained by the intimate contact of ultra-thin PTG on skin regardless the extension/contraction and sweat secretion of skin (See discussion in supporting information). This can be verified by the stable contact impedance at 10 Hz as a function of time (Figure S25).”

Figure S25. (a) Typical contact impedance of PTG on skin after 12 hours. (b) Contact impedance at 10 Hz as a function of time.

In the revised manuscript:

“According to the Young's modulus map under AFM, average Young's modulus of PTG is about 640 kPa (Figure 5c, Figure S10), which matches the modulus of human stratum corneum (~ 150

kPa). Bending stiffness (\overline{EI}) refers to the amount of stiffness that the subject will deflect:

$$\overline{EI} = \frac{1}{12(1-\nu^2)} \overline{E}h^3$$

where \overline{E} represents Young's modulus, h represents thickness, ν represents Poisson ratio. The ultra-thin nature and low Young's modulus of PTG leads to a similar bending stiffness with that of skin, contributing to conformal contact with wrinkled skin (see discussion in Supporting Information)^{45,46}.”

In supporting information:

“To interpret the stable attachment of PTG electrodes on skin during electrophysiological measurement, we introduced the concept of bending stiffness. Bending stiffness refers to the amount of stiffness that the subject will deflect. The smaller the stiffness, the larger the deflection will occur. According to the formula (1) and (2), bending stiffness (\overline{EI}_{sum}) of our PTG electrode can be calculated as following:

$$\overline{EI} = \frac{1}{12(1-\nu^2)} \overline{E}h^3 \quad (1)$$

$$\overline{EI}_{sum} = \alpha \overline{EI}_{electrode} + (1-\alpha) \overline{EI}_{substrate} \quad (2)$$

Where \overline{E} represents Young's modulus, h represents thickness, ν represents Poisson ratio, and α represents the area ratio of electrode to substrate. In this work, the thickness of PTG is about 100 nm. And in real electrophysiological measurements, we transfer PTG on decal transfer tattoo paper, and the thickness of ethylcellulose layer in decal transfer tattoo paper is about 450 nm. Together with the small Young's modulus and ultra-thin nature, bending stiffness of our PTG electrode (with tattoo substrate) is ~ 0.00236 pNm. Correspondingly, according to above formula, bending stiffness of skin is ~ 0.00156 pNm. (Young's modulus of skin is ~150 kPa, and the thickness of stratum corneum is ~ 15 μ m)¹. They match with each other very well. The matched bending stiffness leads to the ultra-conformability between PTG electrodes and skin. Once two subjects form ultra-conformability with each other, they tend to attach intimately. Besides, the existence of sweat may introduce capillary force, which can enhance the intimate contact and stable attachment².